# Model-Free Robust Average-Reward Reinforcement Learning with Sample Complexity Analysis

**Zachary Roch** [1]  **George Atia** [1,2]  **Yue Wang** [1,2]

## Abstract

Robust reinforcement learning (RL) under the average-reward criterion is essential for long-term decision-making, particularly when the environment may differ from its training dynamics. However, most existing studies focus on model-based settings and provide only asymptotic guarantees, hindering their principled understanding and practical deployment, especially in data-limited scenarios. We aim to close this gap by proposing a model-free algorithm, **Robust Halpern Iteration (RHI)**. We first design our algorithm based on a black-box sampling oracle, which can estimate the worst-case performance accurately. We then derive the finite sample complexity of RHI under the generative model setting, assuming the sampling oracle. To concretely design such an oracle, we propose a $K$-order multi-level Monte-Carlo estimator, which is shown to have a lower bias compared to prior methods. We further instantiate our design for multiple uncertainty models, including KL and $\chi^2$ divergence sets, and show that our RHI algorithm achieves an $\varepsilon$-optimal robust policy with a sample complexity of $\tilde{\mathcal{O}}\left(\frac{SA\mathcal{H}^2}{\varepsilon^{(2+o(1))}}\right)$, where $S$, $A$ are the number of states and actions, and $\mathcal{H}$ is the robust optimal span. Our result asymptotically matches the best complexity in robust average reward RL.

## 1. Introduction

Reinforcement Learning (RL) seeks to find an optimal policy for an agent interacting with an environment to maximize a cumulative reward. While RL has achieved remarkable success in controlled settings like board games (Silver et al., 2016; Zha et al., 2021) and video games (Wei et al., 2022; Liu et al., 2022a), its deployment in real-world applications is often hindered by a significant performance drop. This issue, known as the Sim-to-Real gap (Zhao et al., 2020; Peng et al., 2018; Tobin et al., 2017), stems from mismatches between the training (simulation) and deployment (real-world) environments. In contrast to games where these environments are identical, practical scenarios are fraught with model discrepancies arising from modeling errors, environmental perturbations, or even adversarial attacks (Henderson et al., 2018; Rajeswaran et al., 2017; Zhang et al., 2018). Such mismatches can render a learned policy highly suboptimal, severely undermining the reliability of RL in practice. To address this critical reliability challenge, the framework of (distributionally) robust RL was developed (Bagnell et al., 2001; Nilim & El Ghaoui, 2003; Iyengar, 2005). Instead of assuming a single, perfectly known environment model, robust RL considers an uncertainty set of plausible transition dynamics. The objective is to find a policy that optimizes performance for the worst-case model within this set. This approach yields a policy with formal performance guarantees across all considered environmental variations, making it inherently more resilient and robust to model mismatch and enhancing its generalizability (Pinto et al., 2017; Zhang et al., 2025).

Beyond robustness, the choice of the reward criterion fundamentally shapes the RL problem. The discounted-reward criterion, while mathematically elegant and widely studied, can be myopic due to its exponential down-weighting of future rewards, potentially leading to poor long-term outcomes (Schwartz, 1993; Seijen & Sutton, 2014; Tsitsiklis & Roy, 1997; Abounadi et al., 2001). In contrast, numerous real-world applications–such as queuing control, portfolio optimization, and communication networks (Kober et al., 2013; Lu et al., 2018; Chen et al., 2022; Wu et al., 2023; Moody & Saffell, 2001; Charpentier et al., 2023; Masoudi, 2021; Li & Hai, 2024)–demand policies that are evaluated based on their long-term, steady-state performance when executed over an extended period of time. This practical necessity underscores the importance of the average-reward criterion, which does not discount the future reward and

---

[1]Department of Electrical and Computer Engineering, University of Central Florida, Orlando, Florida, USA. [2]Department of Computer Science, University of Central Florida, Orlando, Florida, USA. Correspondence to: Zachary Roch <zachary.roch@ucf.edu>, George Atia <george.atia@ucf.edu>, Yue Wang <yue.wang@ucf.edu>.

*Proceedings of the 43rd International Conference on Machine Learning*, Seoul, South Korea. PMLR 306, 2026. Copyright 2026 by the author(s).

thus captures the long-term performance (Sigaud & Buffet, 2013). In this paper, we focus on the intersection of these two needs: developing robust RL algorithms under the average-reward criterion to ensure the performance of RL systems under model mismatch.

Robust RL under the average-reward criterion, however, is more challenging than the discounted-reward setting and remains relatively understudied. The primary difficulties stem from its reliance on the limiting behavior of stochastic processes, leading to analytical and algorithmic complications. Recent work has highlighted these issues, including the non-contractive nature of the associated Bellman operator, the high dimensionality of the solution space, and the instability of standard iterative algorithms (Wang et al., 2023d; Grand-Clement et al., 2023; Wang et al., 2024e) (see Section 6 for a complete discussion). A critical gap in the literature persists: existing studies are predominantly asymptotic or planning-based, leaving the crucial finite-sample properties of data-driven robust average-reward RL largely unexplored.

A natural strategy to obtain finite-sample results is to reduce the average-reward problem to its discounted counterpart, thereby leveraging the rich literature on robust discounted-reward RL (Wang et al., 2022; Zurek & Chen, 2023). This approach is theoretically supported by the convergence of the robust discounted value function to the average-reward value function as the discount factor approaches one (Wang et al., 2023c). However, these reduction-based methods are often suboptimal (Grand-Clément & Petrik, 2023) or require additional prior knowledge (Roch et al., 2025). While other recent works have proposed direct methods, they typically rely on strong structural assumptions, such as irreducibility, which induce a contraction property (Xu et al., 2025a;b). To circumvent these limitations, in this paper, we propose a direct approach, **Robust Halpern Iteration (RHI)**, which enables a practical, model-free implementation and achieves a near-optimal sample complexity. Our contributions are summarized as follows.

**A universal algorithm with a black-box sampling oracle.** We propose the Robust Halpern Iteration (RHI), a model-free algorithm that bypasses the complexities of reduction-based approaches. Inspired by Halpern Iteration from the optimization literature (Halpern, 1967; Lieder, 2021; Lee et al., 2025), our method integrates two key technical innovations: (1) leveraging a quotient space to manage the high dimensionality of the robust Bellman equation's solution space and tackle the double unknown variables in the equation, and (2) designing a black-box sampling oracle with sufficient conditions for sample-efficient learning. We further provide a rigorous finite-sample analysis for RHI based on the sampling oracle. Our algorithm does not require any prior knowledge of problem parameters as in model-based ones (Roch et al., 2025).

**A $K$-order multi-level Monte-Carlo estimator.** To concretely design the sampling oracle, we propose a $K$-order multi-level Monte-Carlo estimator, which utilizes a weighted MLMC estimator (Liu et al., 2022b; Wang et al., 2023d; 2024g). We further show that our estimator has a much smaller bias compared to standard MLMC. Such a bias reduction further enables us to achieve a tight overall sample complexity when utilized in our RHI. Our design of the estimator is general and universal, and can be utilized for a large body of uncertainty sets defined by $f$-divergence.

**Near optimal sample complexity.** We further adapt our $K$-order MLMC estimator to two uncertainty sets: KL and $\chi^2$ models. We concretely design the sampling oracle for them and derive the overall sample complexity of our RHI. We show that to achieve an $\varepsilon$-optimal policy, RHI requires a sample complexity of $\tilde{\mathcal{O}}(SA\mathcal{H}^2\varepsilon^{-(2+o(1))})$-order. This result asymptotically matches the complexity of model-based approaches in (Roch et al., 2025; Chen et al., 2025), indicating the sample efficiency of our algorithm.

## 2. Preliminaries and Problem Formulation

**Markov decision processes (MDPs).** A discounted reward Markovian decision process (DMDP) $(\mathcal{S}, \mathcal{A}, \mathsf{P}, r, \gamma)$ is specified by: a state space $\mathcal{S}$, an action space $\mathcal{A}$, a nominal (stationary) transition kernel $\mathsf{P} = \{\mathsf{P}_s^a \in \Delta(\mathcal{S}), a \in \mathcal{A}, s \in \mathcal{S}\}$[1], where $\mathsf{P}_s^a$ is the distribution of the next state over $\mathcal{S}$ upon taking action $a$ in state $s$ (with $\mathsf{P}_{s,s'}^a$ denoting the probability of transitioning to $s'$), a reward function $r : \mathcal{S} \times \mathcal{A} \to [0, 1]$, and a discount factor $\gamma \in [0, 1)$. At each time step $t$, the agent at state $s_t$ takes an action $a_t$, the environment then transitions to the next state $s_{t+1}$ according to $\mathsf{P}_{s_t}^{a_t}$, and produces a reward signal $r_t = r(s_t, a_t)$ to the agent.

A stationary policy $\pi : \mathcal{S} \to \Delta(\mathcal{A})$ is a distribution over $\mathcal{A}$ for any given state $s$. The agent follows the policy by taking an action following the distribution $\pi(s)$. The accumulative reward of a stationary policy $\pi$ starting from $s \in \mathcal{S}$ for DMDPs is measured by the discounted value function: $V_{\gamma,\mathsf{P}}^\pi(s) \triangleq \mathbb{E}_{\pi,\mathsf{P}}\left[\sum_{t=0}^\infty \gamma^t r_t | S_0 = s\right]$.

Unlike DMDPs, average reward MDPs (AMDPs) do not discount the rewards over time and instead measure the cumulative reward by considering the behavior of the underlying Markov process under the steady-state distribution. The average reward (or the gain) of a policy $\pi$ from $s$ is

$$g_{\mathsf{P}}^\pi(s) \triangleq \liminf_{n \to \infty} \mathbb{E}_{\pi,\mathsf{P}}\left[\frac{1}{n}\sum_{t=0}^{n-1} r_t | S_0 = s\right]. \quad (1)$$

The bias or the relative value function for an AMDP is defined as the cumulative difference over time between the

---

[1] $\Delta(\mathcal{S})$: the $(|\mathcal{S}| - 1)$-dimensional probability simplex on $\mathcal{S}$.

immediate reward and the average reward:

$$h_{\mathsf{P}}^{\pi}(s) \triangleq \mathbb{E}_{\pi,\mathsf{P}} \left[ \sum_{t=0}^{\infty} (r_t - g_{\mathsf{P}}^{\pi}) | S_0 = s \right]. \qquad (2)$$

**Distributionally robust MDPs.** In distributionally robust MDPs, the transition kernel is not fixed but, instead, belongs to a designated uncertainty set denoted as $\mathcal{P}$. Following an action, the environment undergoes a transition to the next state based on an arbitrary transition kernel $\mathsf{P} \in \mathcal{P}$. In this paper, we mainly focus on the $(s,a)$-rectangular uncertainty set (Nilim & El Ghaoui, 2003; Iyengar, 2005; Wiesemann et al., 2013), where $\mathcal{P} = \bigotimes_{s,a} \mathcal{P}_s^a$, with $\mathcal{P}_s^a \subseteq \Delta(\mathcal{S})$ defined independently over all state-action pairs. In most studies, the uncertainty set is defined through some distribution divergence:

$$\mathcal{P}_s^a = \{q \in \Delta(\mathcal{S}) : D(q||\mathsf{P}_s^a) \le R\}, \qquad (3)$$

where $D$ is some distribution divergence like KL divergence or $\chi^2$ divergence, $\mathsf{P}_s^a$ is the centroid of the uncertainty set, referred to as the nominal kernel, and $R$ is the radius of the uncertainty set for the given state and action, measuring the level of uncertainties. In most studies, the nominal kernel can be viewed as the simulation, and all training data are generated under it.

Robust MDPs aim to optimize the worst-case performance over the uncertainty set. The robust DMDP $(\mathcal{S}, \mathcal{A}, \mathcal{P}, r, \gamma)$ considers the robust discounted value function of a policy $\pi$, which is the worst-case discounted value function over all possible transition kernels:

$$V_{\gamma,\mathcal{P}}^{\pi}(s) \triangleq \min_{\mathsf{P} \in \mathcal{P}} \mathbb{E}_{\pi,\mathsf{P}} \left[ \sum_{t=0}^{\infty} \gamma^t r_t | S_0 = s \right]. \qquad (4)$$

The discounted robust value functions are shown to be the unique solution to the robust discounted Bellman equation (Iyengar, 2005), where $\sigma_{s,a}(V) \triangleq \min_{\mathsf{P} \in \mathcal{P}_s^a} \mathsf{P}V$:

$$V(s) = \sum_a \pi(a|s)(r(s,a) + \gamma \sigma_{s,a}(V)). \qquad (5)$$

When the long-term performance under uncertainty is concerned, we focus on the robust AMDP $(\mathcal{S}, \mathcal{A}, \mathcal{P}, r)$. The worst-case performance is then measured by the following robust average reward:

$$g_{\mathcal{P}}^{\pi}(s) \triangleq \min_{\mathsf{P} \in \mathcal{P}} g_{\mathsf{P}}^{\pi}(s). \qquad (6)$$

The robust AMDP aims to find an optimal policy w.r.t. it: $\pi^* \triangleq \arg\max_{\pi \in \Pi} g_{\mathcal{P}}^{\pi}(s)$, for some $s \in \mathcal{S}$, and we denote the optimal robust average reward by $g_{\mathcal{P}}^* \triangleq \max_\pi g_{\mathcal{P}}^{\pi}$. Moreover, we define the optimal robust bias span for the robust

AMDP as the maximum of bias-span over all optimal policies and corresponding worst-case kernels:

$$\mathcal{H} \triangleq \max_{\mathsf{P} \in \mathcal{P}: g_{\mathsf{P}}^{\pi^*} = g_{\mathcal{P}}^{\pi^*}} \max_{\pi^* \in \Pi^*} \mathsf{sp}(h_{\mathsf{P}}^{\pi^*}) \qquad (7)$$

where $h_{\mathsf{P}}^{\pi^*}$ is the bias defined in (2) and $\mathsf{sp}(h) \triangleq \max_s h(s) - \min_s h(s)$ is the Span semi-norm. Note that $\mathcal{H}$ exists and is finite under the irreducible setting we considered (Wang et al., 2023c). Without loss of generality, we assume $\mathcal{H} \ge 1$.

**Problem formulation.** We consider the standard *generative model setting* (Panaganti & Kalathil, 2022; Shi et al., 2023; Xu et al., 2023), where the learner assumes access to a simulator to generate i.i.d. samples under any state-action pair following the nominal kernel P. Bypassing the exploration challenge, this fundamental setting provides an information-theoretic approach for practical abstraction of simulation-based applications (e.g., robotics simulators or game engines). As we will show, our algorithm in this setting is strictly model-free by foregoing the construction of a transition model; instead directly updating $Q$-values via our robust sampling oracle, R-SAMPLE, outlined in Section 5. We study the sample complexity from the nominal kernel for identifying an $\varepsilon$-optimal policy $\pi$ for the robust AMDP:

$$g_{\mathcal{P}}^{\pi^*}(s) - g_{\mathcal{P}}^{\pi}(s) \le \varepsilon. \qquad (8)$$

## 3. Robust Bellman Equations of RAMDPs

In this work, we consider robust AMDPs with compact uncertainty sets and satisfying the standard irreducible assumption (Xu et al., 2025a;b; Wang et al., 2023c;d; Roch et al., 2025).

**Assumption 3.1.** For any $s \in \mathcal{S}, a \in \mathcal{A}$, the uncertainty set $\mathcal{P}_s^a$ is a compact subset of $\Delta(\mathcal{S})$. And for any $\pi \in \Pi, \mathsf{P} \in \mathcal{P}$, the induced MDP is irreducible.

Due to the hardness of the robust average reward (see Section 4.1), this global irreducibility condition ensures our setting is well-defined and is standard for convergent algorithm design and complexity analysis in robust average-reward RL. In practice, when the nominal kernel is irreducible and the uncertainty radius is sufficiently small, this assumption holds (Xu et al., 2025a;b). Furthermore, our results directly extend to more general settings with bounded bias spans, including unichain (Wang et al., 2023c;d) and potentially weakly communicating environments (See Remark 3.3).

We then characterize structures of robust AMDPs under Assumption 3.1. Specifically, we mainly focus on the following robust Bellman equation of $(Q, g) \in \mathbb{R}^{SA} \times \mathbb{R}$:

$$Q(s,a) = r(s,a) - g + \sigma_{s,a}(Q_{\max}), \qquad (9)$$

where $\cdot_{\max} : \mathbb{R}^{SA} \to \mathbb{R}^S$ is a mapping that maps any $SA$-dimensional vector $Q$ to a $S$-dimensional vector $Q_{\max} \in \mathbb{R}^S$ with entry $Q_{\max}(s) = \max_{a \in \mathcal{A}} Q(s, a)$. This equation will play a central part in our studies.

**Lemma 3.2.** *Consider a robust AMDP satisfying Assumption 3.1. Then the robust Bellman equation in (9) is solvable. And for any solution $(Q^*, g^*)$, $g^* = g^*_{\mathcal{P}}(s)$, $\mathsf{sp}(Q^*_{\max}) \leq \mathcal{H}$, and $\mathsf{sp}(Q^*) \leq \mathcal{H} + 1 = \mathcal{O}(\mathcal{H})$. Moreover, any greedy policy w.r.t. $Q^*$ is robust optimal.*

These results are derived based on the results from (Wang et al., 2023c;d; Roch & Wang, 2025), and directly extend to the unichain setting. Denote $\mathcal{T}_{\mathcal{P},g}(Q)(s,a) \triangleq r(s,a) - g + \sigma_{s,a}(Q_{\max})$, then the robust Bellman (9) can be rewritten as $Q = \mathcal{T}_{\mathcal{P},g}(Q)$. As proved, the optimal policy $\pi^*$ can be obtained from the solution $Q^*$ to (9), thus obtaining the optimal policy is equivalent to solving the equation.

*Remark* 3.3. Lemma 3.2 and further studies could be further extended to the weakly communicating setting in (Wang & Si, 2025), where the solvability and optimality can be similarly derived. The only challenge lies in showing the Span-boundness of all of its solutions, and $\mathcal{H}$ may not be a sufficient bound. If the span of all solutions to the Bellman equation under the weakly communicating setting is upper bounded by $C$, all of our following results directly hold for weakly communicating ones by replacing $\mathcal{H}$ by $C$.

# 4. Robust Halpern Iteration (RHI)

We then design a model-free algorithm to solve (9).

## 4.1. Challenges and Hardness

As discussed in Section 2, finding the optimal policy for a robust AMDP is equivalent to solving the corresponding robust Bellman equation (9): $Q = \mathcal{T}_{\mathcal{P},g^*}(Q) = \mathcal{T}_{\mathcal{P}}(Q) - g^*_{\mathcal{P}}$, where $\mathcal{T}_{\mathcal{P}}(Q) \triangleq r + \sigma_{\mathcal{P}}(Q_{\max})$. However, solving this equation is highly challenging. Firstly, the equation has two unknown variables: $Q$ and $g^*_{\mathcal{P}}$; since $g^*_{\mathcal{P}}$ is unknown, the operator $\mathcal{T}_{\mathcal{P},g^*_{\mathcal{P}}}$ is not readily feasible. Moreover, the Bellman operator $\mathcal{T}_{\mathcal{P},g}$ is not a contraction but merely a *non-expansion* under our setting, invalidating most of the previous model-free iterative methods (Xu et al., 2025a;b; Wang et al., 2024g). Finally, the non-linear structure of $\mathcal{T}_{\mathcal{P},g}$ (compared to the linear structure of the non-robust operator) further results in a complicated solution space to the Bellman equation (Wang et al., 2023d). In the following, we address these challenges sequentially and propose our RHI algorithm.

**Curse of dual variables.** To address the issue of solving an equation with two unknown variables, we first claim that, even if we do not know the value of $g^*_{\mathcal{P}}$, we can still obtain the optimal policy through a proximal equation. Our claim is based on the following result, where we show that a

near-optimal policy can be identified by approximating the solution to the robust Bellman equation (9) w.r.t. the Span semi-norm.

**Lemma 4.1.** *Under Assumption 3.1, let $Q \in \mathbb{R}^{SA}$ and $\pi(s) \in \arg\max_{a \in \mathcal{A}} Q(s, a)$. Then, it holds that:*

$$0 \leq g^*_{\mathcal{P}} - g^\pi_{\mathcal{P}}(s) \leq \mathsf{sp}(\mathcal{T}_{\mathcal{P},g^*_{\mathcal{P}}}(Q) - Q) = \mathsf{sp}(\mathcal{T}_{\mathcal{P}}(Q) - Q).$$

The result thus implies that, to obtain the optimal policy $\pi^*$, exactly solving (9) is not necessary; instead, it suffices to find a weaker solution $Q$ such that $\mathcal{T}_{\mathcal{P},g^*_{\mathcal{P}}}(Q) - Q = ce$, for some constant $c \in \mathbb{R}$ and the all-one vector $e = (1, ..., 1) \in \mathbb{R}^{SA}$ (note that the solution $Q$ to (9) also satisfies the equation with $c = 0$). Moreover, we show that we can get rid of the unknown $g^*_{\mathcal{P}}$ and solve the proximal equation that only contains one variable:

$$\mathcal{T}_{\mathcal{P}}(Q) - Q = ce, \text{ for some } c \in \mathbb{R}. \qquad (10)$$

Noting that the span semi-norm is invariant to constant shifts, and inspired by previous studies of non-robust AMDPs (Zhang et al., 2021; Lee et al., 2025), we instead consider the embedded equation in the quotient space w.r.t. identical vectors. Namely, we define a relation between two vectors $v, w \in \mathbb{R}^{SA}$: $v \sim w$ if $v - w = ce$ for some $c$, which can be directly verified to be an equivalence relation. We thus construct the quotient space $E \triangleq \mathbb{R}^{SA}/\sim$, and the embedded equation of (10) on $E$ becomes $[\mathcal{T}_{\mathcal{P}}(Q)] = [Q]$, where $[\cdot]$ denotes the equivalence class of $\cdot$.

**Contraction-free analysis.** The second challenge is that the robust Bellman operator $\mathcal{T}_{\mathcal{P}}$ is generally not a contraction, but rather only a non-expansion, even in the quotient space $E$. Under the irreducible setting, the contraction is derived under some semi-norm, based on which convergence guarantee and complexity are further obtained (Xu et al., 2025a;b). However, these studies heavily rely on the contraction, and the resulting sample complexity inevitably depends on the unknown contraction coefficient $\gamma$ (see Table 1), leading to a sub-optimal complexity.

To address this issue and develop a contraction-free analysis, we adopt the Halpern iteration (Halpern, 1967) from the stochastic approximation area. Specifically, to solve an equation $x = T(x)$ for a non-expansion operator $T$, the Halpern iteration recursively updates the algorithms through $x_{k+1} = (1 - \beta_{k+1})x_0 + \beta_{k+1}T(x_k)$, which is a convex combination between $T(x_k)$ and the initialization $x_0$. Halpern iteration has been studied in optimization areas (Halpern, 1967; Sabach & Shtern, 2017; Lieder, 2021; Park & Ryu, 2022; Contreras & Cominetti, 2023) and more recently in non-robust RL (Lee et al., 2025; Lee & Ryu, 2025).

Based on the Halpern iteration, we can similarly develop our RHI algorithm in the quotient space as $[Q_{k+1}] = [(1 - \beta_{k+1})Q_0 + \beta_{k+1}\mathcal{T}_{\mathcal{P}}(Q_k)]$. We show in the following result that it will converge to some solution.

**Theorem 4.2.** *Consider the exact robust Halpern iteration* $[Q_{k+1}] = [(1 - \beta_{k+1})Q_0 + \beta_{k+1}\mathcal{T}_{\mathcal{P}}(Q_k)]$, *with* $\beta_k = \frac{k}{k+2}$. *Set* $\pi^k$ *to be the greedy policy w.r.t.* $Q_k$. *Then as* $k \to \infty$,

$$\mathsf{sp}(\mathcal{T}_{\mathcal{P}}(Q_k) - Q_k) \to 0, \text{ and } g_{\mathcal{P}}^* - g_{\mathcal{P}}^{\pi^k} \to 0. \quad (11)$$

It hence implies the asymptotic convergence of the exact RHI algorithm. In the next section, we further consider our algorithm under data-driven settings.

### 4.2. RHI with Black-Box Estimation

We then address the third challenge mentioned, the non-linearity of the Bellman operator, which is the major difference between the robust and non-robust cases.

The major challenge is in estimating $\mathcal{T}_{\mathcal{P}}$ from nominal samples. In the non-robust case, where the operator is linear, the plug-in estimation is unbiased; however, in the robust setting, estimating the worst case from the nominal samples is significantly harder, and the vanilla plug-in estimator is biased (Wang et al., 2023d; Liu et al., 2022b). Such a challenge is known as off-dynamic learning (Eysenbach et al., 2021; Liu & Xu, 2024; Holla, 2022).

To address this issue and design a convergent algorithm, in this section, we assume a black-box oracle, named R-SAMPLE, which can estimate the worst case from nominal data. We then utilize this oracle to design our data-driven RHI algorithm and derive the sample complexity with it. In the next section, we will further construct such an oracle and derive the overall complexity.

**Assumption 4.3** (Black-box R-SAMPLE oracle). Fix $(s, a) \in \mathcal{S} \times \mathcal{A}$. A call to R-SAMPLE$(h_k, h_{k-1}, m_k)$ returns a random increment $D_k(s, a)$ of the form

$$D_k(s, a) = \frac{1}{m_k} \sum_{j=1}^{m_k} \big(F_{s,a}(h_k; \omega_{k,j}) - F_{s,a}(h_{k-1}; \omega_{k,j})\big),$$

where $\omega_{k,j}$ are i.i.d. random seeds and $F_{s,a}(\cdot; \omega)$ is a real-valued functional on bias functions $h \in \mathbb{R}^{\mathcal{S}}$. We assume:

(1). There exists $L_{\mathrm{rs}} > 0$ such that for all $h, h' \in \mathbb{R}^{\mathcal{S}}$ with $h(s_0) = h'(s_0)$, and all $\omega$, $|F_{s,a}(h; \omega) - F_{s,a}(h'; \omega)| \leq L_{\mathrm{rs}} \mathsf{sp}(h - h')$.

(2). There exists a (deterministic) reference robust Bellman operator $\overline{\mathcal{T}} : \mathbb{R}^{SA} \to \mathbb{R}^{SA}$ such that, for every RHI iteration $k$, conditioning on the past filtration $\mathcal{F}_{k-1}$,

$$\mathbb{E}[D_k(s, a) \mid \mathcal{F}_{k-1}] = \overline{\mathcal{T}}(Q_k)(s, a) - \overline{\mathcal{T}}(Q_{k-1})(s, a).$$

(3). There exists a bias parameter $b_{\mathrm{rs}} \geq 0$ such that

$$\big|\overline{\mathcal{T}}(Q)(s, a) - \mathcal{T}_{\mathcal{P}}(Q)(s, a)\big| \leq b_{\mathrm{rs}}, \forall (s, a), Q \in \mathbb{R}^{SA}.$$

(4). Each call of $F$ at $(s, a)$ uses at most $C_{\mathrm{rs}}$ samples.

*Remark* 4.4. By Assumption 4.3, the total sample complexity required for one R-SAMPLE oracle call is $m_k C_{\mathrm{rs}}$.

We assume the black-box oracle R-SAMPLE to estimate the *difference* between two robust support functions with a bias $b_{\mathrm{rs}}$. We then incorporate the oracle in our algorithm design, and present our RHI algorithm in Algorithm 1.

---

**Algorithm 1** Robust Halpern Iteration (RHI) with black-box R-SAMPLE

---

1: **Input:** $Q_0 = 0 \in \mathbb{R}^{S \times A}$, horizon $n$, tolerance $\varepsilon > 0$, confidence $\delta \in (0, 1)$, block budget $m_k$.
2: Set $T_{-1} = r$, $h_{-1} = 0$, $\beta_0 = 0$, $c_0 = 10 \ln^2(2)$.
3: Define $\alpha \triangleq \ln\big(2SA(n + 1)/\delta\big)$.
4: **for** $k = 0, 1, \ldots, n$ **do**
5:     $c_k \triangleq 5(k + 2) \ln^2(k + 2)$, $\beta_k \triangleq k/(k + 2)$.
6:     $Q_k \triangleq (1 - \beta_k)Q_0 + \beta_k T_{k-1}$.
7:     $h_k(s) \triangleq \max_a Q_k(s, a)$ with anchoring $h_k(s_0) = 0$
8:     $d_k \triangleq h_k - h_{k-1}$.
9:     $D_k \triangleq$ R-SAMPLE$(h_k, h_{k-1}, m_k)$.
10:    $T_k \triangleq T_{k-1} + D_k$.
11: **end for**
12: Output greedy policy $\pi_n(s) \in \arg\max_a Q_n(s, a)$.

---

We now state our main result in a black-box form that isolates the dependence on the R-SAMPLE subroutine, and leave its construction for different robust uncertainty sets to the next section.

**Theorem 4.5** (RHI with a black-box R-SAMPLE oracle). *Consider a robust AMDP satisfying Assumption 3.1. Suppose R-SAMPLE satisfies Assumption 4.3 with parameters* $L_{\mathrm{rs}}, b_{\mathrm{rs}}, C_{\mathrm{rs}}$, *with* $b_{\mathrm{rs}} \leq \frac{\varepsilon}{16}$. *Then if we set* $m_k \triangleq \max\left\{1, \left\lceil \frac{\alpha \, c_k \, L_{\mathrm{rs}}^2 \, \mathsf{sp}(d_k)^2}{\varepsilon^2} \right\rceil\right\}$ *and the number of iterations as* $n \geq C \frac{\mathcal{H}}{\varepsilon}$ *for some constant* $C$, *the output policy is* $\varepsilon$-*optimal with probability at least* $1 - \delta$, *with a total sample complexity of*

$$\widetilde{\mathcal{O}}\Big(SA\, L_{\mathrm{rs}}^2\, \mathcal{H}^2\, \varepsilon^{-2}\, C_{\mathrm{rs}}\Big).$$

*Remark* 4.6. To highlight the novelty of our theoretical framework and fully explore the underlying complexities of our R-SAMPLE oracle under KL and $\chi^2$ uncertainty sets, we defer our experimental results to Appendix A.

*Remark* 4.7. Implementing our RHI algorithm does not require any prior knowledge, except that the total iteration number, $n$, depends on $\mathcal{H}$. Although it is common in sample complexity analysis to have an iteration number that depends on unknown underlying parameters like mixing time, e.g., (Li et al., 2024; 2021; Wang et al., 2024f), its concrete and practical implementations can still be challenging. To address this issue, we further modify Algorithm 1 to employ a doubling trick (Auer et al., 1995; Besson & Kaufmann, 2018; Lee et al., 2025), and propose our Parameter-Free

RHI (PF-RHI) algorithm. PF-RHI is completely independent of $\mathcal{H}$, while maintaining the same sample complexity. We defer the discussion to Appendix H.

# 5. Design of `R-SAMPLE`: $K$-Order MLMC

In this section, we design the `R-SAMPLE` oracle satisfying Assumption 4.3 for different uncertainty sets, and further derive the corresponding sample complexity.

We note that the `R-SAMPLE` oracle is designed to estimate the difference $\mathcal{T}_{\mathcal{P}}(Q_k)(s, a) - \mathcal{T}_{\mathcal{P}}(Q_{k-1})(s, a) = \sigma_{s,a}(h_k) - \sigma_{s,a}(h_{k-1})$, using only samples from the nominal kernel. In non-robust settings (Lee et al., 2025), a single sample estimator $h_k(s') - h_{k-1}(s')$ is sufficient and is an unbiased estimator; however, in robust settings, the $\sigma_{s,a}$ is a nonlinear functional of $h$ that is not directly expressible as an expectation under $P$. To address this issue, previous robust studies proposed a multi-level Monte-Carlo (MLMC) approach (Liu et al., 2022b; Wang et al., 2023d; 2024g; Xu et al., 2025b;a), which is an unbiased estimator of the support function. However, as we will discuss later, directly adopting the vanilla MLMC could lead to a high sample complexity ($C_{\mathrm{rs}}$ in Assumption 4.3). Moreover, a key observation is that, to ensure $b_{\mathrm{rs}} \leq \varepsilon$ in Assumption 4.3, we do not necessarily design an unbiased estimator; instead, we could design a sample-efficient estimator with a small bias. Based on this observation, we propose a *$K$-order multilevel Monte Carlo* estimator that satisfies Assumption 4.3 and leads to a better complexity.

## 5.1. $K$-Order MLMC

Our construction exploits a common structure of distributional uncertainty sets: for each $(s, a)$, the robust support can be written (or reduced) to *(locally) smooth functionals of finitely many moments* of $h(S')$ under the nominal kernel. Specifically, we consider scalar functionals of the form

$$G(h) = \Psi(\theta(h)), \quad \theta(h) \triangleq \mathbb{E}_{S' \sim p_{s,a}}[\varphi(h, S')], \quad (12)$$

for some feature map $\varphi(h, s') \in \mathbb{R}^d$ and $\Psi$ satisfying the below assumption.

**Assumption 5.1** (Smoothness of $\Psi$). The function $\Psi : \Theta \to \mathbb{R}$ is defined and smooth on a compact set $\Theta$ containing all possible $\theta(h)$ for $h$ with $\mathrm{sp}(h) \leq \mathcal{H}$. Specifically, for each integer $m \geq 1$, its $m$-th derivative tensor is bounded: $\sup_{x \in \Theta} \|D^m \Psi(x)\| \leq A_m B_m^m m!$, where $\| \cdot \|$ is some operator norm and $A_m, B_m < \infty$ are constants.

*Remark* 5.2. KL, total variation, and $\chi^2$ uncertainty sets all satisfy (12) and Assumption 5.1. Specifically, under the KL uncertainty set, for a fixed dual parameter $\alpha > 0$, the KL dual objective is $g_{s,a}(\alpha, h) = -\alpha \log \mu_{s,a}(\alpha, h) - \alpha \rho_{s,a}$, with $\mu_{s,a}(\alpha, h) \triangleq \mathbb{E}_{p_{s,a}}[e^{-h(S')/\alpha}]$, which fits (12) with $d = 1$, $\varphi(h, s') = e^{-h(s')/\alpha}$ and $\Psi(\mu) = -\alpha \log \mu - \alpha \rho_{s,a}$.

Its support function is then $\sigma_{s,a}^{\mathrm{KL}}(h) = \sup_{\alpha > 0} g_{s,a}(\alpha, h)$ (Iyengar, 2005). $\chi^2$ and total variation duality (Iyengar, 2005) can be similarly verified. More broadly, any $f$-divergence uncertainty set with smooth conjugate $f^*$ provides a smooth dual-objective, making this moment representation applicable to structures like Rényi divergence and the Cressie-Read family (Ghosh et al., 2026).

Given $N$ i.i.d. samples $S'_1, \ldots, S'_N \sim p_{s,a}$, we form the empirical moment as $\widehat{\theta}_N(h) \triangleq \frac{1}{N} \sum_{i=1}^{N} \varphi(h, S'_i)$. The naive, single-level plug-in estimator

$$\widehat{G}_N(h) \triangleq \Psi(\widehat{\theta}_N(h)), \quad (13)$$

incurs a bias of $\mathcal{O}(1/N)$, leading to an excessively large sample complexity. Under Assumption 5.1, we show the precise bias expansion as

$$\mathbb{E}[\widehat{G}_N(h)] = G(h) + \sum_{j=1}^{K} \frac{c_j(h)}{N^j} + f(K) N^{-(K+1)}, \quad (14)$$

with coefficients $c_j(h)$ bounded on $\mathcal{H}$ and a remainder of some scaling function $f(K) = \tilde{\mathcal{O}}(K!)$.

To eliminate the lower-order bias terms in (14), we design a $K$-order MLMC estimator by first fixing an integer $K \geq 0$ and base inner sample size $N_0 \geq 1$. Define levels $N_\ell \triangleq 2^\ell N_0$ for $\ell = 0, \ldots, K$. At a single oracle call, we draw $N_K$ samples $S'_1, \ldots, S'_{N_K} \sim p_{s,a}$ and reuse prefixes to form the single-level estimators $\widehat{G}_{N_\ell}(h)$ as in (13). By taking a linear combination, we construct our $K$-order MLMC as

$$\widetilde{G}_{N_0}^{(K)}(h) \triangleq \sum_{\ell=0}^{K} w_\ell^{(K)} \widehat{G}_{N_\ell}(h). \quad (15)$$

The weights $w_\ell^{(K)}$ are selected such that $\sum_{\ell=0}^{K} w_\ell^{(K)} = 1$, and $\sum_{\ell=0}^{K} \frac{w_\ell^{(K)}}{N_\ell^j} = 0$ is satisfied for all $\forall j \in \{1, ..., K\}$ (such weights $w_\ell^{(K)}$ exist per Lemma F.4). These weights effectively cancel the highest order bias term from each layer, providing a residual bias of $\tilde{\mathcal{O}}(K!/N_0^{K+1})$ (Proposition F.6). Then, all $1/N^j$ bias terms for $j = 1, \ldots, K$ vanish, i.e.,

$$\mathbb{E}[\widetilde{G}_{N_0}^{(K)}(h)] = G(h) + \tilde{\mathcal{O}}\left(f(K) N_0^{-(K+1)}\right). \quad (16)$$

We now show via a comparison of methods that a reduction in bias is essential for obtaining a tight sample complexity.

### 5.1.1. COMPARISON WITH PRIOR METHODS

In this section, we compare our $K$-order MLMC with previous MLMC estimators in (Liu et al., 2022b; Wang et al., 2023d; 2024g; Xu et al., 2025b;a), to illustrate our advantages. We consider the generic scalar functional in (12).

**Single-level Monte Carlo** oracle is the straightforward plug-in estimator $\widehat{G}_N(h) \triangleq \Psi\big(\widehat{\theta}_N(h)\big)$, with $\widehat{\theta}_N(h) \triangleq \frac{1}{N}\sum_{i=1}^{N}\varphi(h, S_i')$. Under our smoothness assumptions on $\Psi$ (Assumption 5.1), we show its bias has an expansion form of

$$\mathbb{E}[\widehat{G}_N(h)] = G(h) + \frac{c_1(h)}{N} + \cdots + \frac{c_K(h)}{N^K} + R_{K+1}(N, h),$$

with $|R_{K+1}(N, h)| \leq C_{K+1} N^{-(K+1)}$. Thus, the leading bias term is $b_{\mathrm{SL}}(N) \triangleq \sup_{h \in \mathcal{H}}\big|\mathbb{E}[\widehat{G}_N(h)] - G(h)\big| = \Theta\big(\frac{1}{N}\big)$. If we plug this into RHI as the inner oracle, the oracle bias requirement $b_{\mathrm{SL}}(N) \lesssim \varepsilon$ results in $C_{\mathrm{rs}} = N \gtrsim \frac{1}{\varepsilon}$. Together with Theorem 4.5 it implies a sample complexity of $\widetilde{\mathcal{O}}\big(SA\,\mathcal{H}^2\,\varepsilon^{-3}\big)$.

**Standard MLMC method** (Giles, 2008; Liu et al., 2022b) considers a sequence of estimators

$$G_\ell(h) \triangleq \widehat{G}_{N_\ell}(h), \qquad N_\ell \triangleq 2^\ell N_0.$$

Since $\mathbb{E}[G_L(h)] = \mathbb{E}[G_0(h)] + \sum_{\ell=1}^{L}\mathbb{E}[G_\ell(h) - G_{\ell-1}(h)]$, the MLMC $\widehat{G}_L^{\mathrm{MLMC}}(h)$ is constructed as

$$\frac{1}{n_0}\sum_{j=1}^{n_0}G_0^{(j)}(h) + \sum_{\ell=1}^{L}\frac{1}{n_\ell}\sum_{j=1}^{n_\ell}\Big(G_\ell^{(j)}(h) - G_{\ell-1}^{(j)}(h)\Big),$$

with appropriately coupled samples $(G_\ell^{(j)}, G_{\ell-1}^{(j)})$ and some random level $L \in [0, \infty)$ following some geometric distribution. Such MLMC estimator is unbiased (Liu et al., 2022b; Wang et al., 2023d); however, sample complexity required can be infinite (Liu et al., 2022b).

The **truncated MLMC (T-MLMC) schemes** studied in prior robust RL work (e.g., (Wang et al., 2024g; Xu et al., 2025a;b)) add a threshold $L_{\max}$ on the level value to the MLMC, so that the sample complexity for each estimator is at most $N_0 2^{L_{\max}}$. However, its bias is shown to be $\tilde{\mathcal{O}}(L_{\max}2^{-L_{\max}}) = \tilde{\mathcal{O}}(1/N)$ (Theorem 4.1 in (Wang et al., 2024g)), which similarly leads to an $\varepsilon^{-3}$-order complexity when combined with Theorem 4.5.

Thus, all of these previous MLMC estimators have an $\tilde{\mathcal{O}}(\frac{1}{N})$-bias, and lead to a high overall sample complexity. On the other hand, our $K$-order MLMC, as discussed, ensures that $\mathbb{E}[\widetilde{G}_{N_0}^{(K)}(h)] = G(h) + \tilde{\mathcal{O}}\big(\frac{f(K)}{N_0^{K+1}}\big)$ (see Proposition F.6 in Appendix for the detailed proof). By choosing the value of $K$ and balancing it with $f(K)$, our $K$-order MLMC has a much smaller bias and leads to a better sample complexity, as we will discuss in the next section.

### 5.2. Instantiations for Uncertainty Sets

We then discuss how our general construction yields a valid R-SAMPLE oracle for KL and $\chi^2$ uncertainty sets.

For KL uncertainty sets, it holds that

$$\sigma_{s,a}^{\mathrm{KL}}(h) = \sup_{\alpha > 0} g_{s,a}(\alpha, h), \tag{17}$$

$$g_{s,a}(\alpha, h) \triangleq -\alpha \log \mu_{s,a}(\alpha, h) - \alpha\rho_{s,a}. \tag{18}$$

We proceed in three steps.

Firstly, since $\alpha^*$, the optimal point of $g_{s,a}(\alpha, h)$, lies in a compact interval $[\alpha_{\min}, \alpha_{\max}]$ (Zhou et al., 2021), we discretize this range into a finite $\varepsilon_{\mathrm{grid}}$-net (Vershynin, 2018; Shi & Chi, 2024; Wang et al., 2024f) $\mathcal{A}_{s,a}$. This yields a discretized support $\sigma_{s,a}^{\mathrm{disc}}(h) \triangleq \max_{\alpha \in \mathcal{A}_{s,a}} g_{s,a}(\alpha, h)$ with an error bounded by $\tilde{\mathcal{O}}(\varepsilon_{\mathrm{grid}})$. For each fixed $\alpha \in \mathcal{A}_{s,a}$, we apply the MLMC estimator (15) to $G(h) = g_{s,a}(\alpha, h)$ as above. This yields $\widetilde{g}_{N_0}^{\mathrm{KL},(K)}(\alpha, h)$ with bias $\tilde{\mathcal{O}}(f(K)N_0^{-(K+1)})$ and span Lipschitz constant $C_w$. Finally, at a call of R-SAMPLE with bias $h$, we compute

$$\widetilde{\sigma}_{s,a}^{\mathrm{KL},(K)}(h; \omega) \triangleq \max_{\alpha \in \mathcal{A}_{s,a}} \widetilde{g}_{N_0}^{\mathrm{KL},(K)}(\alpha, h; \omega),$$

and define the normalized oracle

$$F_{s,a}^{\mathrm{KL},(K)}(h; \omega) \triangleq \widetilde{\sigma}_{s,a}^{\mathrm{KL},(K)}(h; \omega) - \widetilde{\sigma}_{s,a}^{\mathrm{KL},(K)}(0; \omega).$$

The max over a finite grid of $C_w$-Lipschitz functions is again $C_w$-Lipschitz, and the expectation of $F_{s,a}^{\mathrm{KL},(K)}$ defines a reference support $\overline{\sigma}_{s,a}^{\mathrm{KL},(K)}(h)$ whose bias w.r.t. the true $\sigma_{s,a}^{\mathrm{KL}}(h)$ is

$$b_{\mathrm{rs}}^{\mathrm{KL},(K)} = \tilde{\mathcal{O}}(\varepsilon_{\mathrm{grid}}) + \tilde{\mathcal{O}}\Big(f(K)N_0^{-(K+1)}\Big),$$

uniformly over $h \in \mathcal{H}$.

We then present our results under the KL uncertainty set.

**Theorem 5.3.** *The* R-SAMPLE *oracle constructed satisfies Assumption 4.3 with* $L_{\mathrm{rs}}^{\mathrm{KL}} = \tilde{\mathcal{O}}(1), b_{\mathrm{rs}}^{\mathrm{KL}} \leq \varepsilon/16, C_{\mathrm{rs}}^{\mathrm{KL}}(\varepsilon) = \exp\big(\tilde{\mathcal{O}}(\sqrt{\log(1/\varepsilon)})\big), by\ choosing\ K(\varepsilon) \triangleq \Big\lceil\sqrt{\frac{\log(1/\varepsilon)}{\log 2}}\Big\rceil - 1, N_0(\varepsilon, K) \triangleq \Big\lceil c_2(K+1)\,\varepsilon^{-\frac{1}{K+1}}\Big\rceil. Moreover, with*

$$N_{\mathrm{tot}}^{\mathrm{KL}}(\varepsilon) = \widetilde{\mathcal{O}}\Big(SA\mathcal{H}^2\,\varepsilon^{-2-\frac{c}{\sqrt{\log(1/\varepsilon)}}}\Big), \tag{19}$$

*RHI finds an $\varepsilon$-optimal policy with probability $1 - \delta$.*

We can similarly construct R-SAMPLE (deferred to Appendix F.4) and derive the result for $\chi^2$ sets as follows. We defer discussions on $\ell_p$-norm and contamination sets to Appendix G (Theorem G.2), and leave designs for other models as future work.

**Theorem 5.4.** *The oracle constructed satisfies Assumption 4.3 with* $L_{\mathrm{rs}}^{\chi^2} = \tilde{\mathcal{O}}(1 + \sqrt{\rho}), b_{\mathrm{rs}}^{\chi^2} \leq \varepsilon/16, C_{\mathrm{rs}}^{\chi^2}(\varepsilon) = \exp\big(\tilde{\mathcal{O}}(\sqrt{\log(1/\varepsilon)})\big). And\ with*

$$N_{\mathrm{tot}}^{\chi^2}(\varepsilon) = \widetilde{\mathcal{O}}\Big(SA\mathcal{H}^2\varepsilon^{-2-\frac{c}{\sqrt{\log(1/\varepsilon)}}}\Big), \tag{20}$$

*RHI finds an $\varepsilon$-optimal policy with probability $1 - \delta$.*

*Remark* 5.5. With our oracle design, our RHI algorithm achieves an $\varepsilon$-optimal policy with a sample complexity of $\tilde{\mathcal{O}}(SA\mathcal{H}^2\varepsilon^{(-2-o(1))})$. We emphasize that, in the asymptotic regime, the $o(1)$ penalty is strictly *sub-polynomial* and does not push the complexity towards $\varepsilon^{-3}$ in any meaningful way. Specifically, the limit of the exponent evaluates to:

$$\lim_{\varepsilon \to 0^+} \left( -2 - \frac{c}{\sqrt{\log(1/\varepsilon)}} \right).$$

As $\varepsilon \to 0^+$, the denominator $\sqrt{\log(1/\varepsilon)}$ grows to $\infty^+$, causing the lower-order penalty term to vanish. This asymptotically matches the $\tilde{\mathcal{O}}(SA\mathcal{H}^2\varepsilon^{-2})$ state-of-the-art complexity in model-based methods (Roch et al., 2025; Chen et al., 2025) (a more detailed comparison and discussion will be provided in Section 6). This *near-optimal* rate highlights the significant theoretical advantage of our $K$-order MLMC design over prior MLMC approaches, which suffer from $\tilde{\mathcal{O}}(\varepsilon^{-3})$ complexity due to the unmitigated $\mathcal{O}(1/N)$ bias.

## 6. Related Works and Comparison

### 6.1. Comparison with Prior Arts

In this section, we compare with the most related works on finite sample complexity analysis of robust average-reward RL. The comparison is summarized in Table 1.

In (Grand-Clément & Petrik, 2023; Roch et al., 2025; Chen et al., 2025), model-based methods are developed. In these works, a robust discounted reward RL with some specific discount factor (referred to as a reduction factor) is constructed, and its optimal robust policy is shown to be near-optimal under average reward. Thus, the sample complexity of robust average reward RL is then equivalent to that of the corresponding discounted RL with the reduction factor. In (Grand-Clément & Petrik, 2023), an upper bound on the reduction factor is derived as $\gamma \leq 1 - \frac{C}{S^S m^{S^2}}$, when the nominal kernels are rational, i.e., $\mathsf{P}_s^a = n_{s,a}/m_{s,a}$ with $n_{s,a}, m_{s,a} \in \mathbb{N}$, and $m$ is the smallest denominator among all kernel entries. However, coupling this bound with existing sample-complexity results for robust DMDPs yields exponential sample complexity for robust AMDPs. In (Chen et al., 2025), the reduction factor is set to a sample-number-dependent value, and the corresponding sample complexity is derived. However, their results require stronger assumptions on the AMDP structure (uniformly ergodic) and the radius of the uncertainty set (the radius has to be small), limiting the applicability. More recently, a reduction factor $\gamma = 1 - \frac{\varepsilon}{\mathcal{H}}$ has been developed in (Roch et al., 2025), and sample complexity that matches ours is derived under the unichain setting. However, this reduction factor depends on the robust optimal span $\mathcal{H}$, requiring its knowledge even before learning. In practice, access to such knowledge is infeasible, and even its estimation can be challenging and inefficient (Zurek & Chen, 2023; Tuynman et al., 2024).

Another line of work (Xu et al., 2025b;a) utilizes the truncated multi-level Monte-Carlo method to directly find the optimal policy. Under the irreducible assumption, the robust Bellman operator becomes a $\gamma$-contraction w.r.t. some seminorm. However, the sample complexity analysis depends heavily on the unknown contraction parameter $\gamma$, which can be arbitrarily close to 1 (Puterman, 2014), leading to a high complexity.

Additionally, we highlight the substantial technical differences between our work and recent MLMC-based robust RL studies. Wang et al. (2023d) proposes a model-free algorithm for robust average-reward RL, but provides only an asymptotic convergence guarantee using standard MLMC, which may result in an infinite number of samples. To bridge the gap to our finite-sample guarantees, we must balance the bias with the complexity which requires fundamentally new techniques, especially as it relates to our $K$-order MLMC. When compared to Wang et al. (2024g), this work studies robust *discounted* RL using truncated MLMC; the estimator yields a $\mathcal{O}(1/N)$ bias, which ultimately leads to a $\varepsilon^{-3}$ rate (see the discussion in Section 5.1.1). Our work not only introduces the robust average-reward framework, but our $K$-order construction achieves a $\mathcal{O}(f(K)/N_0^{K+1})$ bias which is key to obtaining our near-optimal $\varepsilon^{-(2+o(1))}$ rate. Moreover, our framework handles the double unknown variables $(Q, g^*)$ while simultaneously tackling the non-contractive nature of the operator under the average-reward criterion by employing a quotient-space formulation.

Hence, compared to these prior works, our method enjoys two major advantages: (1). We do not require *any* prior knowledge of the robust AMDP (like $\mathcal{H}$ in (Roch et al., 2025)), and our model-free algorithm is practical and implementable; (2). Our sample complexity asymptotically matches the tightest sample complexity and is better than other model-free results in (Xu et al., 2025a) (noting that $\mathcal{H} \leq t_m$, i.e., the mixing time (Wang et al., 2022; Roch et al., 2025)) as well as fundamentally new techniques from (Wang et al., 2023d) in a setting that more closely models real-world transition dynamics (Wang et al., 2024g). Thus, our RHI method represents the **state-of-the-art** in model-free robust average reward RL.

### 6.2. Other Related work

In this section, we discuss other related works on robust RL with average reward. Additional related works are discussed in Appendix B. Studies on robust RL with average reward are relatively limited. Early research focused on dynamic programming methods in robust AMDPs. These investigations, initiated by (Tewari & Bartlett, 2007) for specific finite-interval uncertainty sets, were subsequently

*Table 1.* Comparison with prior sample complexity results. $t_m$ denotes the mixing time; $\gamma < 1$ in (Xu et al., 2025a;b) is the contraction coefficient under the irreducibility assumption.

| Algorithm | AMDP Structure | Uncertainty Set | Sample Complexity |
|---|---|---|---|
| (Grand-Clément & Petrik, 2023), Model-Based | $P \in \mathbb{Q}$ | N/A | Exponential |
| (Chen et al., 2025), Model-Based | Uniformly ergodic | KL | $\tilde{\mathcal{O}}\left(\frac{SAt_m^2}{\varepsilon^2}\right)$ |
| (Roch et al., 2025), Model-Based | Unichain | TV | $\tilde{\mathcal{O}}\left(\frac{SA\mathcal{H}^2}{\varepsilon^2}\right)$ |
| (Xu et al., 2025b) (policy evaluation), Model-Free | Irreducible & aperiodic | TV | $\tilde{\mathcal{O}}\left(\frac{SAt_m^2}{(1-\gamma)^2\varepsilon^2}\right)$ |
| (Xu et al., 2025a), Model-Free | Irreducible & aperiodic | TV | $\tilde{\mathcal{O}}\left(\frac{SAt_m^2}{(1-\gamma)^2\varepsilon^2}\right)$ |
| Ours, Model-Free | Irreducible | KL/CS | $\tilde{\mathcal{O}}\left(\frac{SA\mathcal{H}^2}{\varepsilon^{2+o(1)}}\right)$ |
| Ours, Model-Free | Irreducible | Contamination | $\tilde{\mathcal{O}}\left(\frac{SA\mathcal{H}^2}{\varepsilon^2}\right)$ |

extended to more general uncertainty models in works such as (Wang et al., 2023c; Grand-Clement et al., 2023; Wang & Si, 2025). These foundational studies were instrumental in revealing the fundamental structure of robust AMDPs and illustrating their connections to robust DMDPs. As an alternative method, (Chatterjee et al., 2024; Meggendorfer et al., 2025) recently proposed a game-theoretic approach for finding the optimal policy. Building on the understanding of robust AMDP structures, the focus also extends to learning algorithms, where (Wang et al., 2023d) introduced a model-free algorithm with asymptotic convergence guarantees. However, all of these aforementioned approaches focus on asymptotic convergence only, leaving finite-sample complexity analyses largely unaddressed. Our RHI framework and accompanying analysis definitively answers this question.

# 7. Conclusion

In this paper, we studied robust reinforcement learning under the average-reward criterion. We introduced the Robust Halpern Iteration (RHI), a model-free meta-algorithm that requires no prior knowledge of the MDP parameters. We analyzed the sample complexity of RHI using a black-box sampling subroutine that can estimate the robust Bellman operator accurately, leading to the design of our $K$-order multi-level Monte-Carlo estimator. This estimator significantly reduces bias when compared to standard MLMC, allowing RHI to achieve an $\varepsilon$-optimal policy with a sample complexity of $\tilde{\mathcal{O}}(SA\mathcal{H}^2\varepsilon^{-(2+o(1))})$ for KL and $\chi^2$ divergence sets, and $\tilde{\mathcal{O}}(SA\mathcal{H}^2\varepsilon^{-2})$ for $\ell_p$-norm and contamination sets. This establishes RHI as the state-of-the-art model-free algorithm for robust average-reward RL, asymptotically matching the best known model-based bounds.

**Limitations and Future Works.** Our analysis assumes access to a generative model, isolating the fundamental statistical complexity. We leave the extensions to practical online (exploration-exploitation) (Lu et al., 2024;

Ghosh et al., 2026) or offline (dataset-driven) (Wang et al., 2024d) settings as future works. Furthermore, our $o(1)$ subpolynomial gap arises specifically from the bias-variance tradeoff inherent in estimating nonlinear functionals of the transition kernel (e.g., the KL or $\chi^2$ support functions), and completely removing it in model-free settings remains an open challenge.

# Acknowledgments

This work was supported by DARPA under Agreement No. HR0011-24-9-0427, NSF under Award CCF-2106339, and an Amazon Research Award, Fall 2025. Any opinions, findings, and conclusions or recommendations expressed in this material are those of the author(s) and do not reflect the views of DARPA, NSF or Amazon.

# Impact Statement

This paper presents work whose goal is to advance the field of Machine Learning, specifically robust reinforcement learning. There are many potential societal consequences of our work; however, since we provide a general framework, we do not feel that there is a need to highlight these.

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

# A. Experimental Results

In this section, we develop numerical experiments to validate our theoretical results.

## A.1. $K$-Order MLMC

In this section, we numerically verify that our $K$-order MLMC estimator can reduce bias compared to standard MLMC. We randomly generate a nominal distribution and implement both estimators for the worst-case among an uncertainty set centered at the nominal kernel. We plot the bias of the estimations v.s. the order of $K$ in Figure 1. As the results shown, our $K$-order MLMC has a smaller bias compared to vanilla MLMC, which validates our theoretical results.

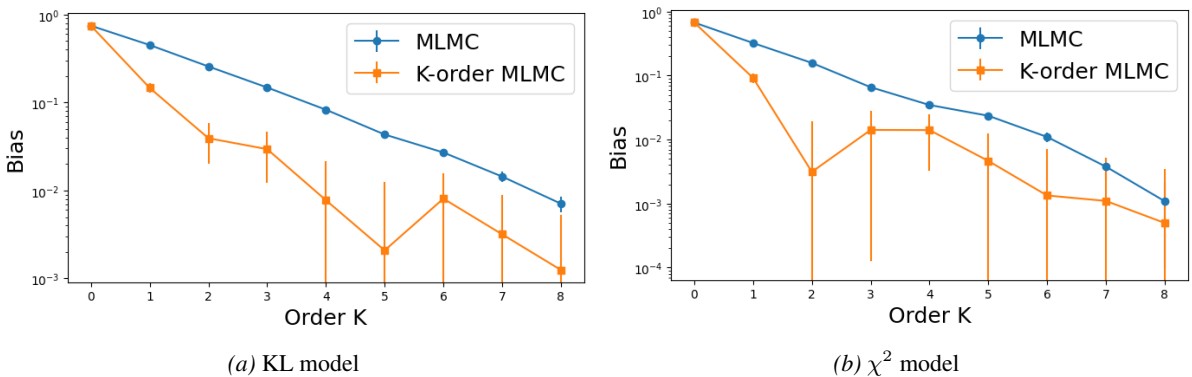

*(a)* KL model  *(b)* $\chi^2$ model

*Figure 1.* Performance of RHI.

## A.2. Convergence of RHI

We conduct experiments to validate our theoretical results and evaluate the empirical performance of RHI. We consider the Garnet problem (Archibald et al., 1995) G(20,15) with 20 states and 15 actions, where nominal transition kernels are randomly generated. We consider three uncertainty sets: the contamination model, the $\ell_\infty$-norm model (total variation), and the $\ell_2$-norm model.

After each iteration of our RHI algorithm, we derive the greedy policy based on the current Q-value estimates from RHI. The robust average reward of this derived policy is then calculated using the RRVI algorithm from (Wang et al., 2023d) and recorded. For comparison, we establish a baseline consisting of the optimal robust average reward, also computed via the RRVI algorithm. Each experimental configuration is repeated for 10 independent runs. All of our experiments require minimal compute resources and are implemented using Google Colab. We present the mean robust average reward across these runs where the shaded region in Figure 2 is the standard deviation.

As depicted in Figure 2, the experimental results demonstrate that our RHI algorithm effectively converges to the optimal robust average reward, thereby corroborating our theoretical findings.

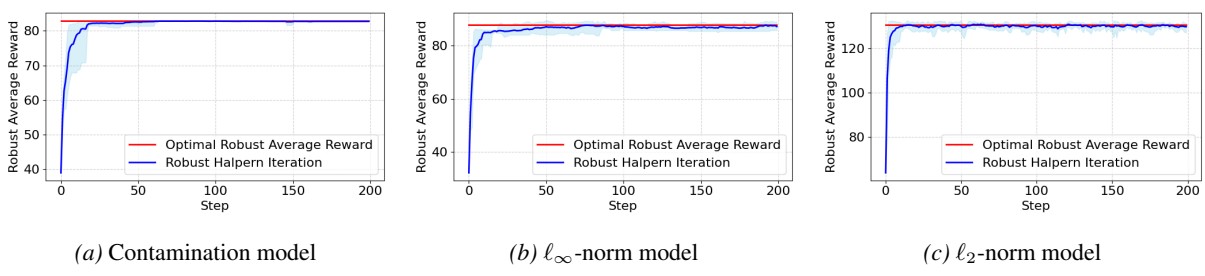

*(a)* Contamination model  *(b)* $\ell_\infty$-norm model  *(c)* $\ell_2$-norm model

*Figure 2.* Performance of RHI.

# B. Additional Related Works

**Robust RL with discounted rewards.** Robust DMDPs were first studied in foundational works such as (Iyengar, 2005; Nilim & El Ghaoui, 2003; Bagnell et al., 2001; Wiesemann et al., 2013; Lim et al., 2016). These initial investigations typically assumed a fully known uncertainty set and developed solutions based on robust DP. Since then, extensive theoretical research has been significantly adapted and extended these concepts to various learning paradigms where the uncertainty set or the nominal model might be unknown or learned from data. Prominent research directions include analyses in settings with generative models (Yang et al., 2022; Panaganti & Kalathil, 2022; Xu et al., 2023; Shi et al., 2023; Zhou et al., 2021; Wang et al., 2023b; Liang et al., 2024; Liu et al., 2022b; Wang et al., 2024c;g; Gadot et al., 2024; Kumar et al., 2023; Derman et al., 2021), investigations into offline learning from fixed datasets (Shi & Chi, 2024; Liu & Xu, 2024; Wang et al., 2024a;d), and developments within online learning frameworks involving exploration (Wang & Zou, 2021; Lu et al., 2024; Ghosh et al., 2026; He et al., 2025). A key focus across these diverse settings is often to provide rigorous finite-sample complexity guarantees or convergence rates, characterized under different assumptions regarding the structure of the uncertainty set and the nature of data access.

**Non-robust RL with average reward.** The study of non-robust AMDPs originated with foundational model-based DP techniques, such as Policy Iteration and Value Iteration, which assume a known model (Puterman, 2014; Bertsekas, 2007). Subsequently, research shifted towards model-free RL algorithms. These include adaptations of Q-learning and SARSA, like RVI Q-learning (Abounadi et al., 2001; Wan et al., 2021; Wan & Sutton, 2022), designed to learn optimal policies directly from interaction data without requiring explicit model knowledge (Dewanto et al., 2021).

Beyond asymptotic convergence, sample complexity for achieving near-optimal policies in (non-robust) AMDPs is extensively studied. A significant body of work is based on the reduction framework, which transforms the AMDP into a DMDP using a carefully chosen discount factor. However, selecting an appropriate discount factor typically requires prior knowledge of crucial MDP parameters, such as the span of the bias function (Zurek & Chen, 2023; Wang et al., 2022; Zurek & Chen, 2024; Sapronov & Yudin, 2025; Jin & Sidford, 2021) or various mixing time constants (Wang et al., 2024b; 2023a). Notable progress has been made under such assumptions, for instance, (Zurek & Chen, 2023; Sapronov & Yudin, 2025) demonstrate that if the bias span is known and used to set the reduction factor, the resulting sample complexity matches the minimax optimal rate for weakly communicating MDPs (Wang et al., 2022). Alongside reduction-based methods, direct approaches that do not involve conversion to DMDPs, but still require prior knowledge, have also been recently developed (Zhang & Xie, 2023; Li et al., 2025). Recognizing that the prerequisite of prior knowledge can be restrictive and impractical, and that estimating these parameters accurately is challenging (Tuynman et al., 2024), another line of research investigates AMDPs without prior knowledge, achieving sub-optimal sample complexity (Lee et al., 2025; Jin et al., 2024; Lee & Ryu, 2025; Tuynman et al., 2024).

Extending these diverse frameworks and insights to robust AMDPs is, however, particularly challenging. This difficulty stems from the greater complexity inherent in the robust average-reward paradigm, including issues such as the non-linearity of the robust Bellman operator and a more intricate, high-dimensional solution space for the robust Bellman equation (Wang et al., 2023d).

# C. Preliminaries and Proof Organization

**Additional notation.**

- We define a stationary policy as $\pi : \mathcal{S} \to \Delta(\mathcal{A})$, and subsequently define the finite set of all stationary policies as $\Pi$ such that $\pi \in \Pi$.

- Since the worst-case robust average reward under the time varying model is equivalent to the one under the stationary model (Wang et al., 2023c), we therefore focus on this time invariant model. For a given stationary policy, $\pi \in \Pi$, satisfying (6), we define the set of minimizing (worst-case) transition kernels as $\Omega_g^\pi \triangleq \{\mathsf{P} \in \mathcal{P} : g_\mathsf{P}^\pi = g_\mathcal{P}^\pi\}$, where $g_\mathsf{P}^\pi(s) \triangleq \liminf_{T\to\infty} \mathbb{E}_{\pi,\mathsf{P}}\left[\frac{1}{T}\sum_{n=0}^{T-1} r_t | S_0 = s\right]$.

- We use $r$ to denote the $SA$-dimensional vector, whose $(s,a)$-th entry is $r(s,a)$. We use $\mathsf{P}_{s,s'}^a$ to denote the transition probability from $s$ to $s'$ under the action $a$ of some transition kernel $\mathsf{P}$.

- Given a policy $\pi$, a reward $r$ and a transition kernel $\mathsf{P}$, we denote the induced reward and state-transition kernel by

$r_\pi \in \mathbb{R}^S$ and $\mathsf{P}_\pi \in \mathbb{R}^{S \times S}$:

$$r^\pi(s) = \sum_a \pi(a|s) r(s,a), \quad (\mathsf{P}^\pi)_{s,s'} = \sum_a \pi(a|s) \mathsf{P}^a_{s,s'}. \tag{21}$$

- For a vector $V \in \mathbb{R}^S$, we use $\mathsf{P}V$ to denote an $SA$-dimensional vector as

$$(\mathsf{P}V)_{s,a} = \mathsf{P}^a_s V. \tag{22}$$

Specifically, for $Q \in \mathbb{R}^{SA}$, $Q_{\max} \in \mathbb{R}^S$, and

$$(\mathsf{P}(Q_{\max}))_{s,a} = \mathsf{P}^a_s(Q_{\max}) = \sum_{s'} \mathsf{P}^a_{s,s'} \max_b \{Q(s',b)\}. \tag{23}$$

- For an uncertainty set $\mathcal{P}$, we denote the robust Bellman operator $\mathcal{T}_{\mathcal{P}}(Q) : \mathbb{R}^{SA} \to \mathbb{R}^{SA}$ as

$$\mathcal{T}_{\mathcal{P}}(Q)(s,a) = r(s,a) + \sigma_{s,a}(Q_{\max}). \tag{24}$$

## D. Proof of Lemma 3.2

**Lemma D.1.** *(Restatement of Lemma 3.2) Consider a robust AMDP satisfying Assumption 3.1. Then it holds that:*

*(1). The robust Bellman equation in (9) is solvable;*

*(2). For any solution $(Q^*, g^*)$, $g^* = g^*_{\mathcal{P}}(s)$, $\mathsf{sp}(Q^*_{\max}) \le \mathcal{H}$ and $\mathsf{sp}(Q^*) \le \mathcal{H} + 1$;*

*Proof.* **Proof of (1).** From Theorem 4.1 of (Roch & Wang, 2025), the value-based robust Bellman equation has a solution $(g = g^*_{\mathcal{P}}, V)$:

$$V(s) = \max_a \left( r(s,a) - g + \sigma_{s,a}(V) \right). \tag{25}$$

Then if we set $Q(s,a) = r(s,a) - g + \sigma_{s,a}(V)$, it then holds that

$$\max_a Q(s,a) = \max_a \{r(s,a) - g + \sigma_{s,a}(V)\} = V(s), \tag{26}$$

i.e., $Q_{\max} = V$. Thus $(g = g^*_{\mathcal{P}}, Q)$ defined is a solution to (9).

**Proof of (2).** The first claim is from (Wang et al., 2023d). Moreover, for any of its solution $(g^*, Q^*)$, it is shown in (Wang et al., 2023d) that $Q^*_{\max} = h^{\pi_{Q^*}}_{P_w}$, where $\pi_{Q^*}(s) = \arg\max_a Q^*(s,a)$, and $P_w$ is the worst-case kernel of $\pi_{Q^*}$, such that $g^{\pi_{Q^*}}_{P_w} = g^{\pi_{Q^*}}_{\mathcal{P}}$; And $\pi_{Q^*} \in \Pi^*$ is an optimal robust policy.

Thus

$$\mathsf{sp}(Q^*_{\max}) = \mathsf{sp}(h^{\pi_{Q^*}}_{P_w}) \le \max_{p \in \mathcal{P} : g^\pi_p = g^\pi_{\mathcal{P}}} \max_{\pi \in \Pi^*} \mathsf{sp}(h^\pi_p) = \mathcal{H}.$$

Moreover, since

$$Q^*(s,a) = r(s,a) - g^*_{\mathcal{P}} + \sigma_{s,a}(Q^*_{\max}) = r(s,a) - g^*_{\mathcal{P}} + \sigma_{s,a}(h^{\pi_{Q^*}}_{P_w}), \tag{27}$$

it holds that

$$\mathsf{sp}(Q^*) = \max_{s,s',a,a'} \{Q^*(s,a) - Q^*(s',a')\} = \max_{s,s',a,a'} \{r(s,a) - r(s',a') + \sigma_{s,a}(h^{\pi_{Q^*}}_{P_w}) - \sigma_{s',a'}(h^{\pi_{Q^*}}_{P_w})\} \tag{28}$$

$$\le 1 + \max_{s,s',a,a'} \{\sigma_{s,a}(h^{\pi_{Q^*}}_{P_w}) - \sigma_{s',a'}(h^{\pi_{Q^*}}_{P_w})\} \tag{29}$$

$$\le 1 + \mathsf{sp}(h^{\pi_{Q^*}}_{P_w}) \tag{30}$$

$$\le 1 + \mathcal{H}. \tag{31}$$

This hence completes the proof.

$\square$

# E. Analysis of RHI

## E.1. Span-Based Results

**Lemma E.1** (Span non-expansion of $T_{\mathcal{P}}$). *For any* $Q, Q' \in \mathbb{R}^{S \times A}$,

$$\mathsf{sp}\big(T_{\mathcal{P}}(Q) - T_{\mathcal{P}}(Q')\big) \leq \mathsf{sp}(Q - Q'). \tag{32}$$

*Proof.* Let $h_Q(s) \triangleq \max_a Q(s, a)$ and $h_{Q'}(s) \triangleq \max_a Q'(s, a)$. Note that

$$\mathsf{sp}\big(T_{\mathcal{P}}(Q) - T_{\mathcal{P}}(Q')\big) = \mathsf{sp}\big(\sigma_{\mathcal{P}}(h_Q) - \sigma_{\mathcal{P}}(h_{Q'})\big)$$
$$\triangleq \big(\sigma_{s,a}(h_Q) - \sigma_{s,a}(h_{Q'})\big) - \big(\sigma_{x,b}(h_Q) - \sigma_{x,b}(h_{Q'})\big). \tag{33}$$

Then it holds that

$$\sigma_{s,a}(h_Q) - \sigma_{s,a}(h_{Q'}) \leq P_{s,a} h_Q - P_{s,a} h_{Q'} = P_{s,a}(h_Q - h_{Q'}) \leq \max_s(h_Q(s) - h_{Q'}(s)), \tag{34}$$

where $P_{s,a} h_{Q'} = \sigma_{s,a}(h_{Q'})$. Moreover,

$$\max_s(h_Q(s) - h_{Q'}(s)) \triangleq h_Q(s') - h_{Q'}(s') \tag{35}$$
$$= \max_a Q(s', a) - \max_a Q'(s', a) \tag{36}$$
$$\leq \max_a\{Q(s, a) - Q'(s, a)\} \tag{37}$$
$$\leq \max_{s,a}\{Q(s, a) - Q'(s, a)\}. \tag{38}$$

On the other hand,

$$\sigma_{x,b}(h_Q) - \sigma_{x,b}(h_{Q'}) \geq q_{x,b}(h_Q - h_{Q'}), \tag{39}$$

where $q_{x,b} h_Q = \sigma_{x,b}(h_Q)$. Then

$$\sigma_{x,b}(h_Q) - \sigma_{x,b}(h_{Q'}) \geq q_{x,b}(h_Q - h_{Q'}) \tag{40}$$
$$\geq \min_s\{h_Q(s) - h_{Q'}(s)\} \tag{41}$$
$$= \min_s\{\max_a Q(s, a) - \max_a Q'(s, a)\} \tag{42}$$
$$\geq \min_{s,a}\{Q(s, a) - Q'(s, a)\}. \tag{43}$$

Thus

$$\mathsf{sp}\big(T_{\mathcal{P}}(Q) - T_{\mathcal{P}}(Q')\big) \leq \max_{s,a}\{Q(s, a) - Q'(s, a)\} - \min_{s,a}\{Q(s, a) - Q'(s, a)\} = \mathsf{sp}(Q - Q'). \tag{44}$$

$\square$

**Lemma E.2** (Restatement of Lemma 4.1). *Under Assumption 3.1, let* $Q \in \mathbb{R}^{SA}$ *and* $\pi$ *be the greedy policy w.r.t.* $Q$, *i.e.,* $\pi(s) \in \arg\max_{a \in \mathcal{A}} Q(s, a)$. *Then for every state* $s \in \mathcal{S}$, *it holds that:*

$$0 \leq g_{\mathcal{P}}^* - g_{\mathcal{P}}^\pi(s) \leq \mathsf{sp}(\mathcal{T}_{\mathcal{P}, g_{\mathcal{P}}^*}(Q) - Q) = \mathsf{sp}(\mathcal{T}_{\mathcal{P}}(Q) - Q).$$

*Proof.* Denote $h(s) \triangleq Q_{\max}(s) = \max_a Q(s, a)$. Since $\pi$ is greedy w.r.t $Q$, it follows that $h(s) = Q(s, \pi(s))$ for all $s \in \mathcal{S}$. We first denote the worst-case transition kernel of $h$ over $\mathcal{P}$ by $\mathsf{P}$, and its induced kernel by $\mathsf{P}_\pi$, i.e.,

$$(\mathsf{P}_\pi h)(s) = \min_{\mathsf{P} \in \mathcal{P}_s^{\pi(s)}} \mathbb{E}_{s' \sim \mathsf{P}}[h(s')] = \sigma_{s, \pi(s)}(h), \quad \forall s \in \mathcal{S}. \tag{45}$$

The robust average reward under Assumption 3.1, $g_{\mathcal{P}}^{\pi}$, exists and is the average reward under the worst-case kernel $\mathsf{P}_{\pi}$, thus it holds that

$$g_{\mathcal{P}}^{\pi} = g_{\mathsf{P}_{\pi}}^{\pi} = \mathsf{P}_{\pi}^{\infty} r_{\pi}, \tag{46}$$

where $r_{\pi} = r(s, \pi(s))$ and $\mathsf{P}_{\pi}^{\infty}$ is the Cesaro limit of $\mathsf{P}_{\pi}$ (Puterman, 2014). Note that it holds that $\mathsf{P}_{\pi}^{\infty} = \mathsf{P}_{\pi}^{\infty} \mathsf{P}_{\pi}$ (Puterman, 2014), thus applying this fact to (46) yields that

$$g_{\mathcal{P}}^{\pi} = \mathsf{P}_{\pi}^{\infty}(r_{\pi} + \mathsf{P}_{\pi}h - h). \tag{47}$$

We further note that the $(s', \pi(s'))$-th entry of $(\mathcal{T}_{\mathcal{P}}(Q) - Q)$ is in fact $(r_{\pi}(s') + (\mathsf{P}_{\pi}h)(s') - h(s'))$, thus it holds that

$$\min_{s' \in \mathcal{S}, a' \in \mathcal{A}} (\mathcal{T}_{\mathcal{P}}(Q) - Q)(s', a') \le (\mathcal{T}_{\mathcal{P}}(Q) - Q)(s', \pi(s)) = (r_{\pi}(s) + (\mathsf{P}_{\pi}h)(s') - h(s')),$$

and by multiplying by $\mathsf{P}_{\pi}^{\infty}$ recursively and (47) we have that

$$\min_{s' \in \mathcal{S}, a' \in \mathcal{A}} (\mathcal{T}_{\mathcal{P}_{\pi}}(Q) - Q)(s', a') \le \mathsf{P}_{\pi}^{\infty}(r_{\pi} + (\mathsf{P}_{\pi}h) - h) = g_{\mathcal{P}}^{\pi}. \tag{48}$$

On the other hand, denote the optimal robust policy as $\pi^*$ and its associated optimal average reward as $g_{\mathcal{P}}^*$. Let $\mathsf{P}_{\pi^*} \in \mathcal{P}$ be the corresponding worst-case transition kernel. Similar to equations 46-47, we have that,

$$g_{\mathcal{P}}^* = g_{\mathsf{P}_{\pi^*}}^{\pi^*} = \mathsf{P}_{\pi^*}^{\infty} r_{\pi^*} = \mathsf{P}_{\pi^*}^{\infty}(r_{\pi^*} + \mathsf{P}_{\pi^*}h - h), \tag{49}$$

We introduce an auxiliary function $h' \in \mathbb{R}^S$ as $h'(s') \triangleq Q(s', \pi^*(s'))$ for all $s' \in \mathcal{S}$. By definition of $h$ and $h'$, we have $h'(s') \le h(s')$ which implies that $-h(s') \le -h'(s')$. Substituting this in (49) implies that for all $s \in \mathcal{S}$,

$$\begin{aligned} g_{\mathcal{P}}^*(s) &= \mathsf{P}_{\pi^*}^{\infty}(r_{\pi^*}(s') + (\mathsf{P}_{\pi^*}h)(s') - h(s')) \\ &\le \mathsf{P}_{\pi^*}^{\infty}(r_{\pi^*}(s') + (\mathsf{P}_{\pi^*}h)(s') - h'(s')). \end{aligned} \tag{50}$$

Now we note that $(r_{\pi^*}(s') + (\mathsf{P}_{\pi^*}h)(s') - h'(s'))$ is exactly the $(s', \pi^*(s'))$-th entry of $(\mathcal{T}_{\mathcal{P}}(Q) - Q)$, then it holds that

$$\begin{aligned} g_{\mathcal{P}}^*(s) &= \mathsf{P}_{\pi^*}^{\infty}(r_{\pi^*}(s') + (\mathsf{P}_{\pi^*}h)(s') - h(s')) \\ &\le \mathsf{P}_{\pi^*}^{\infty}(r_{\pi^*}(s') + (\mathsf{P}_{\pi^*}h)(s') - h'(s')) \\ &\le \mathsf{P}_{\pi^*}^{\infty} \cdot \max_{s' \in \mathcal{S}, a' \in \mathcal{A}} (r(s', a') + (\mathsf{P}_{\pi^*}h)(s') - h'(s')) \\ &= \max_{s', a'} (\mathcal{T}_{\mathcal{P}}(Q) - Q)(s', a') \end{aligned} \tag{51}$$

Thus together with (48), it implies that

$$g_{\mathcal{P}}^* - g_{\mathcal{P}}^{\pi}(s) \le \max_{s', a'} (\mathcal{T}_{\mathcal{P}}(Q) - Q)(s', a') - \min_{s', a'} (\mathcal{T}_{\mathcal{P}}(Q) - Q)(s', a') = \mathsf{sp}(\mathcal{T}_{\mathcal{P}}(Q) - Q). \tag{52}$$

It hence completes the proof by noting that $\mathsf{sp}(\mathcal{T}_{\mathcal{P}, g_{\mathcal{P}}^*}(Q) - Q) = \mathsf{sp}(\mathcal{T}_{\mathcal{P}}(Q) - Q)$ since $g_{\mathcal{P}}^*$ is a constant per Lemma 3.2. $\quad\square$

### E.2. Robust Halpern Iteration and Quotient-Space View

The robust Bellman equation has two unknowns $(Q, g_{\mathcal{P}}^\star)$, and $T_{\mathcal{P}}$ is only a non-expansion (never a contraction) in $\mathsf{sp}(\cdot)$. As in the main text, we avoid this difficulty by:

- Working with the *proximal* equation

$$T_{\mathcal{P}}(Q) - Q = ce,$$

  whose solutions differ only by constant shifts, and

- Embedding into the quotient space $\mathbb{R}^{S \times A} / \sim$, where $v \sim w$ if $v - w = ce$ for some scalar $c$ and all-ones vector $e$.

In this quotient space, $[T_{\mathcal{P}}(Q)] = [Q]$ and we can apply Halpern iteration to the non-expansion $T_{\mathcal{P}}$.

**Halpern iteration.** Given a non-expansive map $T$ in a normed space and an initial point $x_0$, Halpern iteration constructs

$$x_{k+1} = (1 - \beta_{k+1})x_0 + \beta_{k+1}T(x_k), \qquad \beta_k \triangleq \frac{k}{k+2}.$$

When $T$ has a fixed point, $x_k$ converges to it under mild conditions.

In our robust setting, the *ideal* (model-based) Halpern iteration is

$$[Q_{k+1}] = \left[(1 - \beta_{k+1})Q_0 + \beta_{k+1}T_{\mathcal{P}}(Q_k)\right],$$

with $Q_0 = 0$.

In the learning setting, however, we do not know $T_{\mathcal{P}}$, and we only see samples from the nominal $P$. We thus require a sampling oracle.

### E.3. Analysis of RHI with Black-Box `R-SAMPLE`

We now formalize `R-SAMPLE` as a black-box oracle that approximates the increments of a *reference* robust operator $\overline{\mathcal{T}}$, which is itself an approximation to the true $T_{\mathcal{P}}$.

#### E.3.1. BLACK-BOX `R-SAMPLE` ASSUMPTION

We recall the assumptions in Assumption 4.3.

For each $(s, a)$, let $(\Omega, \mathcal{F}, \mathbb{P})$ be a probability space and $F_{s,a} : \mathbb{R}^{\mathcal{S}} \times \Omega \to \mathbb{R}$ a measurable map. Intuitively, $F_{s,a}(h; \omega)$ is a noisy evaluation of some reference robust support function $\overline{\sigma}_{s,a}(h)$.

**Assumption E.3** (Black-box `R-SAMPLE`). Recall Assumption 4.3:

There exist constants $L_{\mathrm{rs}} > 0$ and $b_{\mathrm{rs}} \geq 0$ such that for each $(s, a)$:

1. For all $h, h' \in \mathbb{R}^{\mathcal{S}}$ with $h(s_0) = h'(s_0)$, and all $\omega$,

$$\left|F_{s,a}(h; \omega) - F_{s,a}(h'; \omega)\right| \leq L_{\mathrm{rs}}\, \mathsf{sp}(h - h'). \tag{53}$$

2. Define

$$\overline{\sigma}_{s,a}(h) \triangleq \mathbb{E}_{\omega}\left[F_{s,a}(h; \omega)\right], \tag{54}$$

and the associated reference Bellman operator

$$\overline{\mathcal{T}}(Q)(s, a) \triangleq r(s, a) + \overline{\sigma}_{s,a}(h_Q). \tag{55}$$

At iteration $k$, `R-SAMPLE` uses i.i.d. seeds $\omega_k^{(1)}, \ldots, \omega_k^{(m_k)}$ and sets

$$Z_k^{(j)}(s, a) \triangleq F_{s,a}(h_k; \omega_k^{(j)}) - F_{s,a}(h_{k-1}; \omega_k^{(j)}), \qquad D_k(s, a) \triangleq \frac{1}{m_k}\sum_{j=1}^{m_k} Z_k^{(j)}(s, a). \tag{56}$$

Then, conditioned on the filtration $\mathcal{F}_{k-1}$ generated by all randomness up to iteration $k - 1$,

$$\mathbb{E}\left[D_k(s, a) \mid \mathcal{F}_{k-1}\right] = \overline{\mathcal{T}}(Q_k)(s, a) - \overline{\mathcal{T}}(Q_{k-1})(s, a). \tag{57}$$

3. For all $Q$,

$$\left\|\overline{\mathcal{T}}(Q) - T_{\mathcal{P}}(Q)\right\|_{\infty} \leq b_{\mathrm{rs}}. \tag{58}$$

4. Each call requires $C_{\mathrm{rs}}$ samples.

Based on the black box, we will derive the sample complexity of Algorithm 1 in terms of oracle parameters.

E.3.2. GENERIC CONCENTRATION FOR THE OPERATOR APPROXIMATION

We first show that under Assumption 4.3, R-SAMPLE yields a high-probability bound on the error $\|T_k - \overline{\mathcal{T}}(Q_k)\|_\infty$.

Fix $(s, a)$ and define

$$\mu_k(s, a) \triangleq \mathbb{E}\big[D_k(s, a) \mid \mathcal{F}_{k-1}\big] = \overline{\mathcal{T}}(Q_k)(s, a) - \overline{\mathcal{T}}(Q_{k-1})(s, a),$$

$$Y_k(s, a) \triangleq D_k(s, a) - \mu_k(s, a).$$

Let

$$X_k(s, a) \triangleq \sum_{i=0}^{k} Y_i(s, a).$$

Using initialization $T_{-1} = r$ and $F_{s,a}(0; \omega) = 0$, one checks that $X_k(s, a) = T_k(s, a) - \overline{\mathcal{T}}(Q_k)(s, a)$.

**Lemma E.4.** *Suppose Assumption 4.3 holds. Then for each $(s, a)$ and each $\lambda \in \mathbb{R}$,*

$$\mathbb{E}\big[e^{\lambda Y_k(s,a)} \mid \mathcal{F}_{k-1}\big] \le \exp\bigg(\frac{\lambda^2 L_{\mathrm{rs}}^2 \mathsf{sp}(d_k)^2}{2m_k}\bigg). \tag{59}$$

*Proof.* Conditioned on $\mathcal{F}_{k-1}$, the $Z_k^{(j)}(s, a)$ in (56) are i.i.d., with

$$\big|Z_k^{(j)}(s, a)\big| = \big|F_{s,a}(h_k; \omega_k^{(j)}) - F_{s,a}(h_{k-1}; \omega_k^{(j)})\big| \le L_{\mathrm{rs}} \, \mathsf{sp}(d_k)$$

by the Lipschitz property (53). Hence each $Z_k^{(j)}(s, a) - \mu_k(s, a)$ is centered and bounded in an interval of length at most $2L_{\mathrm{rs}}\mathsf{sp}(d_k)$.

Now $D_k(s, a)$ is the average of the $m_k$ i.i.d. $Z_k^{(j)}(s, a)$, so $Y_k(s, a) = D_k(s, a) - \mu_k(s, a)$ is also centered, with range width at most $2L_{\mathrm{rs}}\mathsf{sp}(d_k)/m_k$. By Hoeffding's lemma,

$$\mathbb{E}\big[e^{\lambda Y_k(s,a)} \mid \mathcal{F}_{k-1}\big] \le \exp\bigg(\frac{\lambda^2}{2} \frac{\big(2L_{\mathrm{rs}}\mathsf{sp}(d_k)/m_k\big)^2}{4} m_k\bigg) = \exp\bigg(\frac{\lambda^2 L_{\mathrm{rs}}^2 \mathsf{sp}(d_k)^2}{2m_k}\bigg).$$

$\square$

**Proposition E.5.** *Fix $n \in \mathbb{N}$, $\varepsilon > 0$, and $\delta \in (0, 1)$. Define*

$$\alpha \triangleq \ln \frac{2SA(n+1)}{\delta}$$

*and choose $c_k$ and $m_k$ as in Algorithm 1, with $\sum_{k=0}^{\infty} c_k^{-1} \le \frac{1}{2}$. Then under Assumption 4.3, with probability at least $1 - \delta$,*

$$\Big\|T_k - \overline{\mathcal{T}}(Q_k)\Big\|_\infty \le \varepsilon, \qquad k = 0, 1, \ldots, n. \tag{60}$$

*Proof.* Fix $(s, a)$ and $\lambda > 0$. From Lemma E.4 and the tower property,

$$\mathbb{E}[e^{\lambda X_k(s,a)}] = \mathbb{E}\big[e^{\lambda X_{k-1}(s,a)} \mathbb{E}[e^{\lambda Y_k(s,a)} \mid \mathcal{F}_{k-1}]\big] \le \mathbb{E}[e^{\lambda X_{k-1}(s,a)}] \exp\bigg(\frac{\lambda^2 L_{\mathrm{rs}}^2 \mathsf{sp}(d_k)^2}{2m_k}\bigg).$$

Inductively, using $X_0(s, a) = 0$,

$$\mathbb{E}[e^{\lambda X_k(s,a)}] \le \exp\bigg(\frac{\lambda^2 L_{\mathrm{rs}}^2}{2} \sum_{i=0}^{k} \frac{\mathsf{sp}(d_i)^2}{m_i}\bigg).$$

By the choice of $m_i$,

$$m_i \ge \alpha L_{\mathrm{rs}}^2 c_i \frac{\mathsf{sp}(d_i)^2}{\varepsilon^2} \quad \Rightarrow \quad \frac{\mathsf{sp}(d_i)^2}{m_i} \le \frac{\varepsilon^2}{\alpha L_{\mathrm{rs}}^2 c_i}.$$

Thus

$$\sum_{i=0}^{k} \frac{\mathsf{sp}(d_i)^2}{m_i} \leq \frac{\varepsilon^2}{\alpha L_{\mathrm{rs}}^2} \sum_{i=0}^{k} \frac{1}{c_i} \leq \frac{\varepsilon^2}{2\alpha L_{\mathrm{rs}}^2},$$

and hence

$$\mathbb{E}[e^{\lambda X_k(s,a)}] \leq \exp\left(\frac{\lambda^2 \varepsilon^2}{4\alpha}\right).$$

By Markov's inequality,

$$\mathbb{P}(X_k(s,a) \geq \varepsilon) \leq \exp\left(-\lambda \varepsilon + \frac{\lambda^2 \varepsilon^2}{4\alpha}\right).$$

This is minimized at $\lambda^\star = 2\alpha/\varepsilon$, giving $\mathbb{P}(X_k(s,a) \geq \varepsilon) \leq e^{-\alpha}$. The same bound holds for $\mathbb{P}(X_k(s,a) \leq -\varepsilon)$ by symmetry, so

$$\mathbb{P}(|X_k(s,a)| \geq \varepsilon) \leq 2e^{-\alpha} = \frac{\delta}{SA(n+1)}.$$

A union bound over all $(s,a)$ and $k = 0, \ldots, n$ yields (60). $\qquad\square$

**Lemma E.6.** *Under Assumption 4.3, for all Q,*

$$\left\|\overline{\mathcal{T}}(Q) - T_{\mathcal{P}}(Q)\right\|_\infty \leq b_{\mathrm{rs}}.$$

*In particular, on the event* (60)*,*

$$\left\|T_k - T_{\mathcal{P}}(Q_k)\right\|_\infty \leq \varepsilon + b_{\mathrm{rs}} =: \varepsilon_T. \tag{61}$$

*Proof.* The first inequality is just (58). Then

$$\left\|T_k - T_{\mathcal{P}}(Q_k)\right\|_\infty \leq \left\|T_k - \overline{\mathcal{T}}(Q_k)\right\|_\infty + \left\|\overline{\mathcal{T}}(Q_k) - T_{\mathcal{P}}(Q_k)\right\|_\infty \leq \varepsilon + b_{\mathrm{rs}}.$$

$\qquad\square$

For the rest of the analysis, we work under the high-probability event

$$\mathcal{E} \triangleq \left\{ \left\|T_k - T_{\mathcal{P}}(Q_k)\right\|_\infty \leq \varepsilon_T \ \forall k = 0, 1, \ldots, n \right\}, \tag{62}$$

which holds with probability at least $1 - \delta$.

### E.3.3. RHI SPAN ANALYSIS WITH GENERIC OPERATOR ERROR $\varepsilon_T$

We now bound the span distances $\mathsf{sp}(Q_k - Q^\star)$ and the Bellman residual $\mathsf{sp}(T_{\mathcal{P}}(Q_k) - Q_k)$ under $\mathcal{E}$.

### E.3.4. SPAN DISTANCE TO $Q^\star$

Let $Q^\star$ be a solution to the robust Bellman equation $Q^\star = T_{\mathcal{P}}(Q^\star)$.

**Lemma E.7.** *Under $\mathcal{E}$, for all $k = 0, 1, \ldots, n$,*

$$\mathsf{sp}(Q_k - Q^\star) \leq \mathsf{sp}(Q_0 - Q^\star) + \frac{2}{3}k\,\varepsilon_T. \tag{63}$$

*Proof.* From the RHI update,

$$Q_k = (1 - \beta_k)Q_0 + \beta_k T_{k-1}, \qquad \beta_k = \frac{k}{k+2},$$

we have

$$\mathsf{sp}(Q_k - Q^\star) \leq (1 - \beta_k)\,\mathsf{sp}(Q_0 - Q^\star) + \beta_k\,\mathsf{sp}(T_{k-1} - Q^\star).$$

Now

$$\mathsf{sp}(T_{k-1} - Q^\star) = \mathsf{sp}\big(T_{k-1} - T_{\mathcal{P}}(Q^\star)\big) \leq 2\left\|T_{k-1} - T_{\mathcal{P}}(Q_{k-1})\right\|_\infty + \mathsf{sp}(Q_{k-1} - Q^\star),$$

using $\mathsf{sp}(x) \le 2\|x\|_\infty$ and $\mathsf{sp}(T_{\mathcal{P}}(Q_{k-1}) - T_{\mathcal{P}}(Q^\star)) \le \mathsf{sp}(Q_{k-1} - Q^\star)$ from Lemma E.1. On $\mathcal{E}$, $\|T_{k-1} - T_{\mathcal{P}}(Q_{k-1})\|_\infty \le \varepsilon_T$, so

$$\mathsf{sp}(T_{k-1} - Q^\star) \le 2\varepsilon_T + \mathsf{sp}(Q_{k-1} - Q^\star).$$

Hence

$$\mathsf{sp}(Q_k - Q^\star) \le (1 - \beta_k)\,\mathsf{sp}(Q_0 - Q^\star) + \beta_k\big(2\varepsilon_T + \mathsf{sp}(Q_{k-1} - Q^\star)\big). \tag{64}$$

Define

$$\theta_k \triangleq (k+1)(k+2)\,\mathsf{sp}(Q_k - Q^\star).$$

Using $\beta_k = k/(k+2)$ and $1 - \beta_k = 2/(k+2)$, one checks that (64) implies

$$\theta_k \le \theta_0 \sum_{i=1}^{k}(i+1) + 2\varepsilon_T \sum_{i=1}^{k} i(i+1) + \theta_0.$$

Hence

$$\theta_k \le \frac{1}{2}(k+1)(k+2)\theta_0 + \frac{2}{3}\varepsilon_T k(k+1)(k+2),$$

and dividing by $(k+1)(k+2)$ yields

$$\mathsf{sp}(Q_k - Q^\star) \le \mathsf{sp}(Q_0 - Q^\star) + \frac{2}{3}k\varepsilon_T.$$

$\square$

### E.3.5. SPAN OF INCREMENTS AND BELLMAN RESIDUAL

We next bound $\mathsf{sp}(Q_k - Q_{k-1})$ and finally $\mathsf{sp}(T_{\mathcal{P}}(Q_k) - Q_k)$.

**Lemma E.8.** *Define $\rho_k \triangleq 2\,\mathsf{sp}(Q_0 - Q^\star) + \frac{2}{3}\varepsilon_T k$. Under $\mathcal{E}$, for all $k \ge 1$,*

$$\mathsf{sp}(Q_k - Q_{k-1}) \le \frac{2}{k(k+1)} \sum_{i=1}^{k} \rho_{i+2}. \tag{65}$$

*Proof.* Using

$$Q_k = \frac{2}{k+2}Q_0 + \frac{k}{k+2}T_{k-1}, \qquad Q_{k-1} = \frac{2}{k+1}Q_0 + \frac{k-1}{k+1}T_{k-2},$$

we obtain

$$Q_k - Q_{k-1} = \frac{2}{(k+1)(k+2)}(T_{k-1} - Q_0) + \frac{k-1}{k+1}(T_{k-1} - T_{k-2}).$$

Thus

$$\mathsf{sp}(Q_k - Q_{k-1}) \le \frac{2}{(k+1)(k+2)}\,\mathsf{sp}(T_{k-1} - Q_0) + \frac{k-1}{k+1}\,\mathsf{sp}(T_{k-1} - T_{k-2}). \tag{66}$$

We first bound $\mathsf{sp}(T_{k-1} - Q_0)$. Using triangle inequality, Lemma E.7 and the non-expansion of $T_{\mathcal{P}}$,

$$\mathsf{sp}(T_{k-1} - Q_0) \le \mathsf{sp}(T_{k-1} - Q^\star) + \mathsf{sp}(Q^\star - Q_0) \le \rho_{k+2}.$$

Next, note that $T_k = T_{k-1} + D_k$ and $h_k(s) = \max_a Q_k(s, a)$ imply, via the non-expansion of the 'max' operator and (56), that

$$\mathsf{sp}(T_k - T_{k-1}) = \mathsf{sp}(D_k) \le L_{\mathsf{rs}}\mathsf{sp}(h_k - h_{k-1}) \le L_{\mathsf{rs}}\mathsf{sp}(Q_k - Q_{k-1}).$$

Thus $\mathsf{sp}(T_{k-1} - T_{k-2}) \le L_{\mathsf{rs}}\mathsf{sp}(Q_{k-1} - Q_{k-2})$.

Plugging these bounds into (66), define $\tilde{\theta}_k \triangleq k(k+1)L_{\mathsf{rs}}\mathsf{sp}(Q_k - Q_{k-1})$. Then

$$\tilde{\theta}_k \le \frac{2k}{k+2}\rho_{k+2} + \tilde{\theta}_{k-1}.$$

By induction,

$$\tilde{\theta}_k \leq 2 \sum_{i=1}^{k} \rho_{i+2}.$$

Dividing by $k(k+1)$ yields (65). □

**Theorem E.9** (Restatement of Theorem 4.2). *Consider the exact robust Halpern iteration* $[Q^{k+1}] = [(1 - \beta_{k+1})Q^0 + \beta_{k+1}\mathcal{T}_{\mathcal{P}}(Q)]$, *with* $\beta_k = \frac{k}{k+2}$. *Set* $\pi^k$ *to be the greedy policy w.r.t.* $Q^k$. *Then,*

$$\text{sp}(\mathcal{T}_{\mathcal{P}}(Q^k) - Q^k) \to 0, \text{ and } g_{\mathcal{P}}^* - g^{\pi^k} \to 0, \text{ as } k \to \infty. \tag{67}$$

*Proof.* By Lemma E.2, we have that $g_{\mathcal{P}}^* - g_{\mathcal{P}}^{\pi^k} \leq \text{sp}(\mathcal{T}_{\mathcal{P}}(Q^k) - Q^k)$, thus it suffices to show that $\text{sp}(\mathcal{T}_{\mathcal{P}}(Q^k) - Q^k) \to 0$.

We derive our analysis under the event $\mathcal{E}$, that with probability at least $(1 - \delta)$, we have that $\|T^k - \mathcal{T}_{\mathcal{P}}(Q^k)\|_\infty \leq \varepsilon$ for all $(s, a) \in \mathcal{S} \times \mathcal{A}$ and for all $k = 0, 1, \ldots, n$.

For ease of reading, we drop the brackets from the equivalence class notations. Our RHI updates as $Q^k = (1 - \beta_k)Q^0 + \beta_k T^{k-1} = \frac{2}{k+2}Q^0 + \frac{k}{k+2}T^{k-1}$ in the quotient space, which implies that for each $(s, a) \in \mathcal{S} \times \mathcal{A}$, we have the following decomposition

$$\mathcal{T}_{\mathcal{P}}(Q^k) - Q^k \tag{68}$$
$$= \frac{2}{k+2} \underbrace{\left(\mathcal{T}_{\mathcal{P}}(Q^k) - Q^0\right)}_{\text{Term 1}} + \frac{k}{k+2} \underbrace{\left(\mathcal{T}_{\mathcal{P}}(Q^k) - \mathcal{T}_{\mathcal{P}}(Q^{k-1})\right)}_{\text{Term 2}} + \frac{k}{k+2} \underbrace{\left(\mathcal{T}_{\mathcal{P}}(Q^{k-1}) - T^{k-1}\right)}_{\text{Term 3}}.$$

We then bound the three terms.

**Term 1:**
Recall that $\rho_k = 2\text{sp}(Q^0 - Q^*) + \frac{2}{3}\varepsilon k$. From the invariance of $\text{sp}(\cdot)$ by additive constants, the triangle inequality, the nonexpansivity of $\mathcal{T}_{\mathcal{P}}(\cdot)$ under the span seminorm, it yields

$$\text{sp}(\mathcal{T}_{\mathcal{P}}(Q^k) - Q^0) \leq \text{sp}(Q^k - Q^*) + \text{sp}(Q^* - Q^0)$$
$$\leq \rho_k, \quad \forall(s, a) \in \mathcal{S} \times \mathcal{A}.$$

**Term 2:**
This term can be bounded through a similarly as

$$\text{sp}\left(\mathcal{T}_{\mathcal{P}}(Q^k) - \mathcal{T}_{\mathcal{P}}(Q^{k-1})\right) = \text{sp}(Q^k - Q^{k-1})$$
$$\leq \frac{2}{k(k+1)} \sum_{i=1}^{k} \rho_{i+2}, \quad \forall(s, a) \in \mathcal{S} \times \mathcal{A}.$$

**Term 3:**
From $\mathcal{E}$ we have that

$$\text{sp}(\mathcal{T}_{\mathcal{P}}(Q^{k-1}) - T^{k-1}) \leq \varepsilon, \quad \forall(s, a) \in \mathcal{S} \times \mathcal{A}.$$

We then combine all three terms in (68), and we have that

$$g_{\mathcal{P}}^* - g_{\mathcal{P}}^{\pi^k}(s)$$

$$\overset{Lemma\ 4.1}{\leq} \mathsf{sp}(\mathcal{T}_{\mathcal{P}}(Q^k) - Q^k) \tag{69}$$

$$\overset{(68)}{\leq} \frac{2}{k+2}\rho_k + \frac{k}{k+2}\left[\frac{2}{k(k+1)}\sum_{i=1}^{k}\rho_{i+2}\right] + \frac{k}{k+2}(2\varepsilon) \tag{70}$$

$$\overset{(a)}{\leq} \frac{4}{k+2}\mathsf{sp}(Q^0 - Q^*) + \frac{4\varepsilon k}{3(k+2)} + \frac{2}{(k+1)(k+2)}\left[\sum_{i=1}^{k}\left(2\mathsf{sp}(Q^0 - Q^*) + \frac{2}{3}\varepsilon(i+2)\right)\right] + 2\varepsilon \tag{71}$$

$$\overset{(b)}{\leq} \frac{4}{k+2}\mathsf{sp}(Q^0 - Q^*) + \frac{4\varepsilon k}{3(k+2)} + \frac{4k}{(k+1)(k+2)}\mathsf{sp}(Q^0 - Q^*) + \frac{2\varepsilon(k^2 + 5k)}{3(k+2)(k+1)} + 2\varepsilon \tag{72}$$

$$\leq \frac{4(1+2k)}{(k+2)(k+1)}\mathsf{sp}(Q^0 - Q^*) + \frac{4(3k^2 + 8k + 3)}{3(k+2)(k+1)}\varepsilon \tag{73}$$

$$\leq \frac{8\mathsf{sp}(Q^0 - Q^*)}{k+2} + 4\varepsilon, \tag{74}$$

where inequality $(a)$ is from the definition of $\rho_k$, inequality $(b)$ is from $\mathsf{sp}(\cdot) \leq 2\|\cdot\|_\infty$.

The proof is thus completed by letting $k \to \infty$ and $\varepsilon \to 0$. $\qquad\square$

**Theorem E.10.** *Assume $\mathcal{E}$ holds. Then for all $k \geq 0$,*

$$\mathsf{sp}\big(T_{\mathcal{P}}(Q_k) - Q_k\big) \leq \frac{8}{k+2}\mathsf{sp}(Q_0 - Q^\star) + 4\varepsilon_T. \tag{75}$$

*Consequently,*

$$g_{\mathcal{P}}^\star - g_{\mathcal{P}}^{\pi_k}(s) \leq \frac{8}{k+2}\mathsf{sp}(Q_0 - Q^\star) + 4\varepsilon_T, \qquad \forall s \in \mathcal{S}. \tag{76}$$

*Proof.* The performance bound (76) follows directly from (75) and Lemma 4.1. We focus on (75).

Recall $Q_k = \frac{2}{k+2}Q_0 + \frac{k}{k+2}T_{k-1}$. Then

$$T_{\mathcal{P}}(Q_k) - Q_k = \frac{2}{k+2}\big(T_{\mathcal{P}}(Q_k) - Q_0\big) + \frac{k}{k+2}\big(T_{\mathcal{P}}(Q_k) - T_{k-1}\big)$$

$$= \frac{2}{k+2}\big(T_{\mathcal{P}}(Q_k) - Q_0\big) + \frac{k}{k+2}\big(T_{\mathcal{P}}(Q_k) - T_{\mathcal{P}}(Q_{k-1})\big)$$

$$+ \frac{k}{k+2}\big(T_{\mathcal{P}}(Q_{k-1}) - T_{k-1}\big).$$

Taking spans,

$$\mathsf{sp}\big(T_{\mathcal{P}}(Q_k) - Q_k\big) \leq \frac{2}{k+2}\mathsf{sp}\big(T_{\mathcal{P}}(Q_k) - Q_0\big) + \frac{k}{k+2}\mathsf{sp}\big(T_{\mathcal{P}}(Q_k) - T_{\mathcal{P}}(Q_{k-1})\big)$$

$$+ \frac{k}{k+2}\mathsf{sp}\big(T_{\mathcal{P}}(Q_{k-1}) - T_{k-1}\big).$$

For the first term, by triangle inequality, Lemma E.7, and non-expansion of $T_{\mathcal{P}}$,

$$\mathsf{sp}\big(T_{\mathcal{P}}(Q_k) - Q_0\big) \leq \mathsf{sp}(Q_k - Q^\star) + \mathsf{sp}(Q^\star - Q_0) \leq 2\mathsf{sp}(Q_0 - Q^\star) + \frac{2}{3}k\varepsilon_T = \rho_{k+2}.$$

For the second term, by non-expansion of $T_{\mathcal{P}}$ and Lemma E.8,

$$\mathsf{sp}\big(T_{\mathcal{P}}(Q_k) - T_{\mathcal{P}}(Q_{k-1})\big) \leq \mathsf{sp}(Q_k - Q_{k-1}) \leq \frac{2}{k(k+1)}\sum_{i=1}^{k}\rho_{i+2}.$$

For the third term, on $\mathcal{E}$ we have $\|T_{\mathcal{P}}(Q_{k-1}) - T_{k-1}\|_\infty \le \varepsilon_T$, hence

$$\mathsf{sp}\big(T_{\mathcal{P}}(Q_{k-1}) - T_{k-1}\big) \le 2\varepsilon_T.$$

Combining and simplifying (bounding the sum of $\rho_{i+2}$ with a crude quadratic polynomial in $k$) yields

$$\mathsf{sp}\big(T_{\mathcal{P}}(Q_k) - Q_k\big) \le \frac{8}{k+2}\,\mathsf{sp}(Q_0 - Q^\star) + 4\varepsilon_T.$$

We omit the purely algebraic simplification here, which mirrors the proofs above. $\qquad\square$

### E.3.6. BOUNDING $\mathsf{sp}(d_k)$ AND $m_k$

Under $\mathcal{E}$, Lemma E.8 and the definition of $\rho_k$ give

$$\mathsf{sp}(d_k) = \mathsf{sp}(h_k - h_{k-1}) \le \mathsf{sp}(Q_k - Q_{k-1}) \le \frac{4}{k+1}\,\mathsf{sp}(Q_0 - Q^\star) + 2\varepsilon_T.$$

Thus

$$\mathsf{sp}(d_k)^2 \le \frac{32}{(k+1)^2}\,\mathsf{sp}(Q_0 - Q^\star)^2 + 8\varepsilon_T^2,$$

up to numerical constants.

Recall

$$m_k \le 1 + \alpha L_{\mathrm{rs}}^2 c_k \frac{\mathsf{sp}(d_k)^2}{\varepsilon^2},$$

with $c_k = 5(k+2)\ln^2(k+2)$. Summing over $k = 0, \ldots, n$ and approximating sums by integrals yields

$$\sum_{k=0}^{n} m_k \le \tilde{\mathcal{O}}\left(\alpha L_{\mathrm{rs}}^2 \frac{\mathsf{sp}(Q_0 - Q^\star)^2 + \varepsilon_T^2 n^2}{\varepsilon^2}\right), \tag{77}$$

where $\tilde{\mathcal{O}}$ hides logarithmic factors in $n, S, A, 1/\delta$.

### E.3.7. TOTAL SAMPLE COMPLEXITY

Let $\mathcal{H}$ be the robust optimal bias span,

$$\mathcal{H} \triangleq \max_{P \in \mathcal{P}} \mathsf{sp}\big(h_P^\star\big),$$

where $h_P^\star$ is the bias of the optimal robust policy for kernel $P$. As $\mathsf{sp}(Q^\star) \le \mathcal{H} + 1$ and hence, with $Q_0 = 0$,

$$\mathsf{sp}(Q_0 - Q^\star) \le \mathcal{H} + 1.$$

We now choose parameters so that the explicit Bellman residual bound in Theorem E.10 yields a final performance error at most $\varepsilon_{\mathrm{out}}$.

**Theorem E.11** (Generic RHI sample complexity with black-box R-SAMPLE). *Suppose Assumption 3.1 and Assumption 4.3 hold with parameters $(L_{\mathrm{rs}}, b_{\mathrm{rs}}, C_{\mathrm{rs}})$, and let $\mathcal{H}$ be the robust optimal bias span. Fix $\varepsilon_{\mathrm{out}} > 0$ and $\delta \in (0,1)$. Assume we can choose the internal R-SAMPLE design so that*

$$b_{\mathrm{rs}} \le \frac{\varepsilon_{\mathrm{out}}}{16}.$$

*Run Algorithm 1 with*

$$n \ge \frac{16\mathcal{H}}{\varepsilon_{\mathrm{out}}}, \qquad \varepsilon = \frac{\varepsilon_{\mathrm{out}}}{16},$$

*and with $\alpha = \ln\big(2SA(n+1)/\delta\big)$ and $m_k$ chosen. Then, with probability at least $1 - \delta$, the greedy policy $\pi_n$ returned satisfies*

$$g_{\mathcal{P}}^\star - g_{\mathcal{P}}^{\pi_n}(s) \le \varepsilon_{\mathrm{out}}, \qquad \forall s \in \mathcal{S},$$

*and the total number of samples drawn from the generative model is bounded by*

$$\tilde{\mathcal{O}}\left(SA\,L_{\mathrm{rs}}^2 C_{\mathrm{rs}} \frac{\mathcal{H}^2}{\varepsilon_{\mathrm{out}}^2}\right).$$

**Choice of error budgets and horizon.** Fix a target performance accuracy $\varepsilon_{\text{out}} > 0$. We choose the internal concentration tolerance $\varepsilon$ in Proposition E.5 and the oracle bias $b_{\text{rs}}$ so that

$$\varepsilon = \frac{\varepsilon_{\text{out}}}{16}, \qquad b_{\text{rs}} \leq \frac{\varepsilon_{\text{out}}}{16}. \tag{78}$$

On the high-probability event $\mathcal{E}$ in (62) we then have

$$\varepsilon_T \;\triangleq\; \varepsilon + b_{\text{rs}} \;\leq\; \frac{\varepsilon_{\text{out}}}{8}.$$

We also choose the Halpern horizon to satisfy

$$n \;\geq\; \frac{16\mathcal{H}}{\varepsilon_{\text{out}}}. \tag{79}$$

Using $Q_0 = 0$ and $\mathsf{sp}(Q^\star) \leq H$, Theorem E.10 with $k = n$ gives, on $\mathcal{E}$,

$$\mathsf{sp}(\mathcal{T}_{\mathcal{P}}(Q_n) - Q_n) \;\leq\; \frac{8}{n+2}\mathsf{sp}(Q_0 - Q^\star) + 4\varepsilon_T \;\leq\; \frac{8\mathcal{H}}{n+2} + 4\varepsilon_T.$$

With the choices (78)–(79) we have

$$\frac{8H}{n+2} \;\leq\; \frac{8H}{16H/\varepsilon_{\text{out}}} \;=\; \frac{\varepsilon_{\text{out}}}{2}, \qquad 4\varepsilon_T \;\leq\; 4 \cdot \frac{\varepsilon_{\text{out}}}{8} \;=\; \frac{\varepsilon_{\text{out}}}{2},$$

and hence

$$\mathsf{sp}(\mathcal{T}_{\mathcal{P}}(Q_n) - Q_n) \;\leq\; \varepsilon_{\text{out}}. \tag{80}$$

By Lemma E.2, this implies

$$0 \;\leq\; g_{\mathcal{P}}^\star - g_{\mathcal{P}}^{\pi_n}(s) \;\leq\; \mathsf{sp}(\mathcal{T}_{\mathcal{P}}(Q_n) - Q_n) \;\leq\; \varepsilon_{\text{out}}, \qquad \forall s \in \mathcal{S},$$

so $\pi_n$ is $\varepsilon_{\text{out}}$-optimal.

**Total number of R-SAMPLE calls.** To bound the number of oracle calls we use the generic bound in (77):

$$\sum_{k=0}^{n} m_k \;\leq\; \tilde{\mathcal{O}}\left(\alpha L_{\text{rs}}^2 \frac{\mathsf{sp}(Q_0 - Q^\star)^2 + \varepsilon_T^2 n^2}{\varepsilon^2}\right),$$

where $\tilde{\mathcal{O}}$ hides logarithmic factors in $n, S, A, 1/\delta$ and $\alpha = \ln(2SA(n+1)/\delta)$.

With $Q_0 = 0$ we have $\mathsf{sp}(Q_0 - Q^\star) \leq \mathcal{H}$, and with the choices above,

$$\varepsilon_T \leq \frac{\varepsilon_{\text{out}}}{8}, \qquad n \leq c\frac{\mathcal{H}}{\varepsilon_{\text{out}}} \quad \text{for some absolute constant } c \geq 16.$$

Therefore

$$\mathsf{sp}(Q_0 - Q^\star)^2 + \varepsilon_T^2 n^2 \;\leq\; \mathcal{H}^2 + \left(\frac{\varepsilon_{\text{out}}}{8}\right)^2 \left(c\frac{\mathcal{H}}{\varepsilon_{\text{out}}}\right)^2 \;\leq\; C_0 \mathcal{H}^2$$

for some numerical constant $C_0 > 0$. Using $\varepsilon = \varepsilon_{\text{out}}/16$ then yields

$$\sum_{k=0}^{n} m_k \;\leq\; \tilde{\mathcal{O}}\left(\alpha L_{\text{rs}}^2 \frac{\mathcal{H}^2}{\varepsilon_{\text{out}}^2}\right).$$

Each R-SAMPLE call at a given $(s, a)$ uses at most $C_{\text{rs}}$ nominal samples. Multiplying by $SA$ (one call per state–action pair at each $k$) therefore gives a total generative sample complexity

$$\tilde{\mathcal{O}}\left(SA\, L_{\text{rs}}^2 C_{\text{rs}} \frac{\mathcal{H}^2}{\varepsilon_{\text{out}}^2}\right).$$

# F. `R-SAMPLE` via Truncated Multi-Level Monte Carlo for KL and $\chi^2$ Uncertainty Sets

In this section we construct and analyze the `R-SAMPLE` subroutine for two distributionally robust uncertainty sets: KL divergence balls and $\chi^2$ divergence balls.

We proceed in three steps:

1. We state the abstract conditions required of `R-SAMPLE` for the RHI analysis to hold, and recall the resulting sample complexity.

2. We develop a general multi-level Monte Carlo (MLMC) estimator for scalar functionals of the form $G(h) = \Psi(\theta(h))$, where $\theta(h)$ is a finite-dimensional moment vector of $h$ under the nominal kernel.

3. We specialize this construction to KL and $\chi^2$ balls, and show that the corresponding `R-SAMPLE` implementations satisfy the abstract conditions with suitable Lipschitz and bias constants. We then choose the number of MLMC levels $K$ and the base sample size $N_0$ as functions of $\varepsilon$ and derive the total sample complexity.

We denote by $\mathcal{T}_{\mathcal{P}}$ the robust Bellman operator

$$\mathcal{T}_{\mathcal{P}}(Q)(s,a) \triangleq r(s,a) + \sigma_{s,a}(Q_{\max}), \qquad Q_{\max}(s) \triangleq \max_{a'} Q(s,a'),$$

where $\sigma_{s,a}(\cdot)$ is the robust support functional at $(s,a)$ associated with the uncertainty set. The RHI algorithm maintains iterates $(Q_k, T_k)$ and uses `R-SAMPLE` to estimate $\mathcal{T}_{\mathcal{P}}(Q_k) - \mathcal{T}_{\mathcal{P}}(Q_{k-1})$ at each step.

We also use the span seminorm $\mathsf{sp}(h) = \max_s h(s) - \min_s h(s)$ and define

$$\mathcal{H} \triangleq \{h \in \mathbb{R}^{\mathcal{S}} : \mathsf{sp}(h) \le \mathcal{H}\},$$

where $\mathcal{H}$ is the (unknown) optimal robust bias span.

## F.1. Abstract Conditions on `R-SAMPLE` and RHI Performance

We first recall the properties that `R-SAMPLE` must satisfy in order for the RHI analysis.

**Assumption F.1** (Abstract `R-SAMPLE` properties). For each $(s,a)$ and each pair of bias functions $h, h' \in \mathcal{H}$, `R-SAMPLE` takes as input $(h, h', m)$ and returns $D_k(s,a)$ of the form

$$D_k(s,a) = \frac{1}{m} \sum_{j=1}^{m} \big(F_{s,a}(h; \omega_j) - F_{s,a}(h'; \omega_j)\big),$$

where the $\omega_j$ are i.i.d. sample blocks drawn according to some internal distribution (depending only on $(s,a)$), and $F_{s,a}(\cdot; \omega)$ is a real-valued functional on $\mathcal{H}$. We assume:

1. **Span Lipschitz:** There exists $L_{\mathrm{rs}} > 0$ such that for all $h, h' \in \mathbb{R}^{\mathcal{S}}$ with $h(s_0) = h'(s_0)$, and all $\omega$,

   $$|F_{s,a}(h; \omega) - F_{s,a}(h'; \omega))| \le L_{\mathrm{rs}} \, \mathsf{sp}(h - h').$$

2. **Reference operator and unbiased increments:** There exists a (deterministic) robust Bellman operator $\overline{\mathcal{T}} : \mathbb{R}^{SA} \to \mathbb{R}^{SA}$ such that for every RHI iteration $k$, conditioning on the filtration $\mathcal{F}_{k-1}$ generated by past iterates and randomness,

   $$\mathbb{E}[D_k(s,a) \mid \mathcal{F}_{k-1}] = \overline{\mathcal{T}}(Q_k)(s,a) - \overline{\mathcal{T}}(Q_{k-1})(s,a),$$

   where $Q_k$ is the RHI iterate and $h_k = Q_{k,\max}$.

3. **Uniform bias between $\overline{\mathcal{T}}$ and $\mathcal{T}_{\mathcal{P}}$:** There exists $b_{\mathrm{rs}} \ge 0$ such that for all $Q \in \mathbb{R}^{SA}$,

   $$\big|\overline{\mathcal{T}}(Q)(s,a) - \mathcal{T}_{\mathcal{P}}(Q)(s,a)\big| \le b_{\mathrm{rs}}, \qquad \forall (s,a).$$

Under Assumption F.1, the RHI analysis shows that if we run RHI for $n \simeq H/\varepsilon$ iterations with appropriately chosen per-iteration sample sizes $m_k$, and if the oracle bias satisfies $b_{\mathrm{rs}} \lesssim \varepsilon$, then with probability at least $1 - \delta$ the greedy policy $\pi_n$ w.r.t. $Q_n$ is $\varepsilon$-optimal:

$$g_{\mathcal{P}}^* - g_{\mathcal{P}}^{\pi_n}(s) \le \varepsilon, \qquad \forall s \in \mathcal{S},$$

and the total number of R-SAMPLE calls is

$$N_{\mathrm{or}} = \widetilde{\mathcal{O}}\Big( S A \, \mathcal{H}^2 \, \varepsilon^{-2} \Big),$$

where $\widetilde{\mathcal{O}}$ hides logarithmic factors in $S, A, H, 1/\varepsilon$. Each R-SAMPLE call uses a certain number of nominal samples, denoted $N_{\mathrm{inner}}$, so the *total generative sample complexity* is

$$N_{\mathrm{tot}} \asymp N_{\mathrm{or}} \times N_{\mathrm{inner}}.$$

## F.2. A General Multi-Level Monte Carlo (MLMC) Scheme

We now develop a general MLMC scheme for scalar functionals of the form $G(h) = \Psi(\theta(h))$, where $\theta(h)$ is a finite-dimensional moment of $h$ under the nominal kernel. This scheme will be used as a building block for both KL and $\chi^2$ uncertainty sets.

### F.2.1. MOMENT REPRESENTATION AND SINGLE-LEVEL ESTIMATOR

Fix a state-action pair $(s, a)$ and write $p = p_{s,a}$ for the nominal next-state distribution. We consider a functional of the bias $h \in \mathcal{H}$ of the form

$$G(h) = \Psi(\theta(h)),$$

where:

- $\theta : \mathcal{H} \to \mathbb{R}^d$ is a moment map

$$\theta(h) \triangleq \mathbb{E}_{S' \sim p}[\varphi(h, S')],$$

 for some bounded feature map $\varphi : \mathcal{H} \times \mathcal{S} \to \mathbb{R}^d$.

- $\Psi : \Theta \subset \mathbb{R}^d \to \mathbb{R}$ is a smooth map on a compact set $\Theta$ containing all possible $\theta(h)$ for $h \in \mathcal{H}$.

Given $N \in \mathbb{N}$ and $h \in \mathcal{H}$, draw $S'_1, \dots, S'_N \overset{\text{i.i.d.}}{\sim} p$ and define the empirical moment

$$\widehat{\theta}_N(h) \triangleq \frac{1}{N} \sum_{i=1}^{N} \varphi(h, S'_i).$$

The single-level plug-in estimator is

$$\widehat{G}_N(h) \triangleq \Psi(\widehat{\theta}_N(h)). \tag{81}$$

In addition to Assumption 5.1, we assume:

**Assumption F.2.** There exists $L_0 > 0$ such that for all $N \ge 1$, all realizations of $S'_1, \dots, S'_N$, and all $h(s_0) = h'(s_0) \in \mathcal{H}$,

$$\left| \widehat{G}_N(h) - \widehat{G}_N(h') \right| \le L_0 \, \mathsf{sp}(h - h'). \tag{82}$$

**Bias expansion.** Under Assumption 5.1, a standard multivariate Delta-method argument yields a $K$-order bias expansion.

**Lemma F.3** (Bias expansion). *Fix an integer $K \ge 0$. Under Assumption 5.1, there exist bounded functions $c_1, \dots, c_K : \mathcal{H} \to \mathbb{R}$ and constants $C_{K+1} < \infty$ (depending on $K$ but not on $N$ or $h$) such that for all $h \in \mathcal{H}$ and $N \ge 1$,*

$$\mathbb{E}\big[\widehat{G}_N(h)\big] = G(h) + \sum_{j=1}^{K} \frac{c_j(h)}{N^j} + R_{K+1}(N, h), \qquad \left| R_{K+1}(N, h) \right| \le \frac{C_{K+1}}{N^{K+1}}. \tag{83}$$

*Proof.* Fix $h \in \mathcal{H}$ and abbreviate $\theta = \theta(h)$, $\widehat{\theta}_N = \widehat{\theta}_N(h)$, $\Delta_N = \widehat{\theta}_N - \theta$. By boundedness of $\varphi$, $\|\Delta_N\| = O_{\mathbb{P}}(1)$ and $\mathbb{E}[\Delta_N] = 0$, with

$$\mathrm{Cov}(\Delta_N) = \frac{1}{N}\Sigma(h),$$

for some covariance matrix $\Sigma(h)$ that is bounded in operator norm uniformly over $h \in \mathcal{H}$.

We now expand $\Psi$ in a multivariate Taylor series around $\theta$ to order $K+1$. Let $\nabla\Psi$ and $D^m\Psi$ denote the gradient and $m$-th order derivative tensor of $\Psi$ at $\theta$. For any $z$ in a neighborhood of $\theta$,

$$\Psi(\theta + z) = \Psi(\theta) + \sum_{m=1}^{K} \frac{1}{m!}D^m\Psi(\theta)[z^{\otimes m}] + R_{K+1}(z), \tag{84}$$

with remainder

$$R_{K+1}(z) = \frac{1}{(K+1)!}D^{K+1}\Psi(\theta + tz)[z^{\otimes(K+1)}]$$

for some $t = t(z) \in (0,1)$ (by the mean-value form of Taylor's theorem). Bounding $D^{K+1}\Psi$ on the compact $\Theta$ by some constant $M_{K+1}$ and using $\|z^{\otimes(K+1)}\| \leq \|z\|^{K+1}$, we have

$$|R_{K+1}(z)| \leq \frac{M_{K+1}}{(K+1)!}\|z\|^{K+1}. \tag{85}$$

Apply (84) with $z = \Delta_N$, then

$$\widehat{G}_N(h) = \Psi(\theta) + \sum_{m=1}^{K} \frac{1}{m!}D^m\Psi(\theta)[\Delta_N^{\otimes m}] + R_{K+1}(\Delta_N).$$

Taking expectations,

$$\mathbb{E}[\widehat{G}_N(h)] = \Psi(\theta) + \sum_{m=1}^{K} \frac{1}{m!}D^m\Psi(\theta)\big[\mathbb{E}[\Delta_N^{\otimes m}]\big] + \mathbb{E}[R_{K+1}(\Delta_N)]. \tag{86}$$

We now examine $\mathbb{E}[\Delta_N^{\otimes m}]$ as a function of $N$. Write $\Delta_N = \frac{1}{N}\sum_{i=1}^{N} Z_i$ where $Z_i = \varphi(h, S_i') - \theta$ are i.i.d. with $\mathbb{E}[Z_i] = 0$ and bounded moments of all orders. Then

$$\Delta_N^{\otimes m} = \frac{1}{N^m}\sum_{i_1,\ldots,i_m} Z_{i_1} \otimes \cdots \otimes Z_{i_m}.$$

By independence and centering, only index patterns with all indices appearing at least twice contribute to $\mathbb{E}[\Delta_N^{\otimes m}]$. Hence $\mathbb{E}[\Delta_N^{\otimes m}]$ is a linear combination of terms of the form $N^{-r}\Gamma_r(h)$ with integer $r \in \{1, 2, \ldots, \lfloor m/2 \rfloor\}$, where $\Gamma_r(h)$ are tensors independent of $N$. Therefore, for each fixed $m$ we can write

$$\mathbb{E}[\Delta_N^{\otimes m}] = \sum_{j=1}^{\lfloor m/2 \rfloor} \frac{A_{m,j}(h)}{N^j},$$

for suitable tensors $A_{m,j}(h)$ uniformly bounded in $h$.

Substituting this into (86), and regrouping terms by powers of $1/N$, we obtain

$$\mathbb{E}[\widehat{G}_N(h)] = \Psi(\theta) + \sum_{j=1}^{K} \frac{c_j(h)}{N^j} + \widetilde{R}_{K+1}(N, h),$$

for some functions $c_j(h)$ (polynomials in the tensors $A_{m,j}(h)$ and $D^m\Psi(\theta)$) and a remainder $\widetilde{R}_{K+1}(N, h)$ absorbing all contributions from $m > K$ and from $\mathbb{E}[R_{K+1}(\Delta_N)]$ and from $j > K$ in the above sums.

Since $\varphi$ is bounded and $\Psi$ has bounded derivatives, there is a constant $C'_{K+1}$ (depending on $K$ but not on $N$ or $h$) such that

$$\mathbb{E}\big[\|\Delta_N\|^{K+1}\big] \leq \frac{C'_{K+1}}{N^{K+1}}.$$

Combining with (85) and using Jensen's inequality yields

$$\big|\mathbb{E}[R_{K+1}(\Delta_N)]\big| \leq \frac{M_{K+1}C'_{K+1}}{(K+1)!}\,\frac{1}{N^{K+1}}.$$

All other leftover terms in $\widetilde{R}_{K+1}(N,h)$ are also of order $N^{-(K+1)}$ by construction. Hence

$$\big|\widetilde{R}_{K+1}(N,h)\big| \leq \frac{C_{K+1}}{N^{K+1}},$$

for some constant $C_{K+1}$ independent of $N$ and $h$, which establishes (83) with $G(h) = \Psi(\theta(h))$. $\qquad\square$

### F.2.2. K-LEVEL MLMC ESTIMATOR AND ITS WEIGHTS

Fix an integer $K \geq 0$ and a base inner sample size $N_0 \geq 1$. Define levels $N_\ell \triangleq 2^\ell N_0$ for $\ell = 0, 1, \ldots, K$. At a single oracle call (fixed $h$), draw $N_K$ i.i.d. samples $S'_1, \ldots, S'_{N_K} \sim p$ and reuse prefixes to define

$$\widehat{G}_{N_\ell}(h) \triangleq \Psi\Big(\frac{1}{N_\ell}\sum_{i=1}^{N_\ell} \varphi(h, S'_i)\Big).$$

We then form a linear combination

$$\widetilde{G}^{(K)}_{N_0}(h) \triangleq \sum_{\ell=0}^{K} w^{(K)}_\ell\,\widehat{G}_{N_\ell}(h), \tag{87}$$

with weights $w^{(K)} = (w^{(K)}_0, \ldots, w^{(K)}_K)$ chosen so that

$$\sum_{\ell=0}^{K} w^{(K)}_\ell = 1, \tag{88}$$

$$\sum_{\ell=0}^{K} \frac{w^{(K)}_\ell}{N^j_\ell} = 0, \qquad j = 1, \ldots, K. \tag{89}$$

Let $x_\ell \triangleq 2^{-\ell}$, so $N_\ell = 2^\ell N_0$ and $N^{-j}_\ell = N^{-j}_0 x^j_\ell$. Then (88)–(89) are equivalent to

$$\sum_{\ell=0}^{K} w^{(K)}_\ell x^j_\ell = \mathbb{I}\{j = 0\}, \qquad j = 0, \ldots, K.$$

**Lemma F.4** (Existence of weights)**.** *The linear system $\sum_{\ell=0}^{K} w^{(K)}_\ell x^j_\ell = \mathbb{I}\{j=0\}$, $j = 0, \ldots, K$, has a unique solution given by*

$$w^{(K)}_\ell = \prod_{\substack{m=0 \\ m\neq\ell}}^{K} \frac{-x_m}{x_\ell - x_m} = (-1)^K \prod_{\substack{m=0 \\ m\neq\ell}}^{K} \frac{1}{2^{m-\ell}-1}, \qquad \ell = 0, \ldots, K. \tag{90}$$

*Proof.* The matrix $V_{j\ell} = x^j_\ell$ is a Vandermonde matrix with distinct nodes $x_\ell$, hence invertible. The solution is $w^{(K)}_\ell = L_\ell(0)$, where $L_\ell$ is the $\ell$th Lagrange basis polynomial for the nodes $x_0, \ldots, x_K$:

$$L_\ell(x) = \prod_{\substack{m=0 \\ m\neq\ell}}^{K} \frac{x - x_m}{x_\ell - x_m}.$$

Setting $x = 0$ yields the first expression in (90); plugging $x_m = 2^{-m}$ and simplifying gives the second. $\qquad\square$

**Lemma F.5.** *There exists a universal constant $C_w < \infty$ such that for all $K \geq 0$,*

$$\sum_{\ell=0}^{K} |w_\ell^{(K)}| \leq C_w. \tag{91}$$

*Proof.* From (90),

$$|w_\ell^{(K)}| = \prod_{\substack{m=0 \\ m \neq \ell}}^{K} \frac{1}{|2^{m-\ell} - 1|} = \prod_{d=1}^{\ell} \frac{1}{1 - 2^{-d}} \cdot \prod_{d=1}^{K-\ell} \frac{1}{2^d - 1},$$

where we reindexed $d = \ell - m$ (for $m < \ell$) and $d = m - \ell$ (for $m > \ell$). Let

$$C_1 \triangleq \prod_{d=1}^{\infty} \frac{1}{1 - 2^{-d}} < \infty, \qquad a_L \triangleq \prod_{d=1}^{L} \frac{1}{2^d - 1}, \quad a_0 \triangleq 1.$$

Then $|w_\ell^{(K)}| \leq C_1 a_{K-\ell}$ and

$$\sum_{\ell=0}^{K} |w_\ell^{(K)}| \leq C_1 \sum_{\ell=0}^{K} a_{K-\ell} = C_1 \sum_{L=0}^{K} a_L \leq C_1 \sum_{L=0}^{\infty} a_L.$$

Since $2^d - 1 \geq 2^{d-1}$, we have $a_L \leq 2^{-L(L-1)/2}$, and hence $\sum_L a_L < \infty$. Let $C_2 \triangleq \sum_{L=0}^{\infty} a_L$ and set $C_w \triangleq C_1 C_2$. $\square$

**Proposition F.6** (Bias and Lipschitzness of the MLMC estimator). *Under Assumptions 5.1 and F.2, the MLMC estimator $\widetilde{G}_{N_0}^{(K)}$ defined in (87) satisfies, for all $h \in \mathcal{H}$,*

$$\left| \mathbb{E}[\widetilde{G}_{N_0}^{(K)}(h)] - G(h) \right| \leq \frac{\widetilde{C}_{K+1}}{N_0^{K+1}}, \tag{92}$$

*with constants $\widetilde{C}_{K+1} \leq C_0' \alpha'^{K+1}(K+1)!$ for some $C_0', \alpha' > 0$, and for all realizations and all $h(s_0) = h'(s_0) \in \mathcal{H}$,*

$$\left| \widetilde{G}_{N_0}^{(K)}(h) - \widetilde{G}_{N_0}^{(K)}(h') \right| \leq L_0 C_w \, \mathsf{sp}(h - h'). \tag{93}$$

*Proof.* Bias: apply Lemma F.3 to each level $N_\ell$:

$$\mathbb{E}[\widehat{G}_{N_\ell}(h)] = G(h) + \sum_{j=1}^{K} \frac{c_j(h)}{N_\ell^j} + R_{K+1}(N_\ell, h).$$

Summing with weights $w_\ell^{(K)}$ and using the constraints (88)–(89),

$$\mathbb{E}[\widetilde{G}_{N_0}^{(K)}(h)] = G(h) + \sum_{\ell=0}^{K} w_\ell^{(K)} R_{K+1}(N_\ell, h).$$

Hence

$$\left| \mathbb{E}[\widetilde{G}_{N_0}^{(K)}(h)] - G(h) \right| \leq \frac{C_{K+1}}{N_0^{K+1}} \sum_{\ell=0}^{K} |w_\ell^{(K)}| 2^{-(K+1)\ell}.$$

The sum is bounded by $C_w$ from Lemma F.5, yielding (92) with $\widetilde{C}_{K+1} \leq C_{K+1} C_w$.

Lipschitz: by (82) and the triangle inequality,

$$\left| \widetilde{G}_{N_0}^{(K)}(h) - \widetilde{G}_{N_0}^{(K)}(h') \right| \leq \sum_{\ell=0}^{K} |w_\ell^{(K)}| |\widehat{G}_{N_\ell}(h) - \widehat{G}_{N_\ell}(h')| \leq L_0 \left( \sum_{\ell=0}^{K} |w_\ell^{(K)}| \right) \mathsf{sp}(h - h'),$$

and (91) gives (93). $\square$

## F.3. KL Divergence Balls

We now instantiate the above scheme for KL divergence uncertainty sets.

### F.3.1. KL-BALL AND DUAL REPRESENTATION

For each $(s, a)$, the KL-ball around the nominal kernel $p_{s,a}$ with radius $\rho_{s,a} > 0$ is

$$\mathcal{P}^{\mathrm{KL}}_{s,a} \triangleq \Big\{ q \in \Delta(\mathcal{S}) : D_{\mathrm{KL}}(q\|p_{s,a}) \triangleq \sum_{s'} q(s') \log \frac{q(s')}{p_{s,a}(s')} \leq \rho_{s,a} \Big\}.$$

The robust support used in the Bellman operator is the worst-case expectation:

$$\sigma^{\mathrm{KL}}_{s,a}(h) \triangleq \inf_{q \in \mathcal{P}^{\mathrm{KL}}_{s,a}} \sum_{s'} q(s')h(s'), \qquad h \in \mathcal{H}.$$

By Donsker–Varadhan duality, this can be written as

$$\sigma^{\mathrm{KL}}_{s,a}(h) = \sup_{\alpha > 0} g_{s,a}(\alpha, h), \tag{94}$$

where for each $\alpha > 0$,

$$\mu_{s,a}(\alpha, h) \triangleq \mathbb{E}_{S' \sim p_{s,a}}[e^{-h(S')/\alpha}], \qquad g_{s,a}(\alpha, h) \triangleq -\alpha \log \mu_{s,a}(\alpha, h) - \alpha \rho_{s,a}.$$

So the KL support is a supremum over one-dimensional dual objectives.

### F.3.2. RANGE TRUNCATION AND GRID IN $\alpha$

First, one can show that for any $h \in \mathcal{H}$, any maximizer $\alpha^*$ of $g_{s,a}(\alpha, h)$ lies in a compact interval $[\alpha_{\min}, \alpha_{\max}]$, where the bounds depend only on $\rho_{s,a}$, $H$ and the support of $p_{s,a}$. Intuitively, as $\alpha \to \infty$, the $-\alpha \rho_{s,a}$ term drives $g_{s,a}(\alpha, h) \to -\infty$, while as $\alpha \downarrow 0$ the soft-min behavior of $\mu_{s,a}(\alpha, h)$ yields finite limits and we can bound $\alpha^*$ away from zero using first-order optimality conditions. We denote

$$\mathcal{I}_{s,a} \triangleq [\alpha_{\min}, \alpha_{\max}].$$

Next, we choose a finite grid $\mathcal{A}_{s,a} = \{\alpha_1, \dots, \alpha_M\} \subset \mathcal{I}_{s,a}$, with mesh size at most $\Delta_\alpha > 0$. The map $\alpha \mapsto g_{s,a}(\alpha, h)$ is smooth in $\alpha$ on $\mathcal{I}_{s,a}$ for each fixed $h \in \mathcal{H}$, and its derivative can be bounded uniformly in $(h, \alpha)$ using the boundedness of $h$ and $\mu_{s,a}(\alpha, h)$. Therefore there exists $L_\alpha < \infty$ such that

$$|g_{s,a}(\alpha, h) - g_{s,a}(\alpha', h)| \leq L_\alpha |\alpha - \alpha'|, \qquad \forall \alpha, \alpha' \in \mathcal{I}_{s,a}, \ h \in \mathcal{H}.$$

Choosing $\Delta_\alpha \leq \varepsilon/(32 L_\alpha)$ yields

$$\Big| \sup_{\alpha \in \mathcal{I}_{s,a}} g_{s,a}(\alpha, h) - \max_{\alpha \in \mathcal{A}_{s,a}} g_{s,a}(\alpha, h) \Big| \leq \varepsilon/32, \qquad \forall h \in \mathcal{H}.$$

Define the discretized support

$$\sigma^{\mathrm{disc}}_{s,a}(h) \triangleq \max_{\alpha \in \mathcal{A}_{s,a}} g_{s,a}(\alpha, h).$$

Combining range truncation and gridding yields

$$\big| \sigma^{\mathrm{KL}}_{s,a}(h) - \sigma^{\mathrm{disc}}_{s,a}(h) \big| \leq b_{\mathrm{range}} + b_{\mathrm{grid}} \leq \varepsilon/16, \qquad \forall h \in \mathcal{H}, \tag{95}$$

for suitable choices of $\alpha_{\min}, \alpha_{\max}$ and $\mathcal{A}_{s,a}$.

F.3.3. MOMENT REPRESENTATION AND SINGLE-LEVEL ESTIMATOR FOR FIXED $\alpha$

Fix $(s, a)$ and $\alpha \in \mathcal{A}_{s,a}$. We can write

$$g_{s,a}(\alpha, h) = \Psi_{s,a,\alpha}(\theta_{s,a,\alpha}(h)),$$

with

$$\theta_{s,a,\alpha}(h) \triangleq \mathbb{E}_{S' \sim p_{s,a}}[e^{-h(S')/\alpha}], \qquad \Psi_{s,a,\alpha}(\mu) \triangleq -\alpha \log \mu - \alpha \rho_{s,a}.$$

For $h \in \mathcal{H}$ and $\alpha \in \mathcal{I}_{s,a}$ we have $h(S') \in [\min h, \max h] \subset [-H, H]$, hence $e^{-H/\alpha_{\min}} \leq e^{-h(S')/\alpha} \leq e^{H/\alpha_{\min}}$, and $\theta_{s,a,\alpha}(h)$ lies in a compact interval $[\mu_{\min}, \mu_{\max}] \subset (0, \infty)$ independent of $h$ and $\alpha$. On this interval, $\Psi_{s,a,\alpha}$ is analytic, with

$$\frac{d^m}{d\mu^m} \Psi_{s,a,\alpha}(\mu) = (-1)^m \alpha (m-1)! \mu^{-m},$$

so $|D^m \Psi_{s,a,\alpha}(\mu)| \leq \alpha(m-1)! \mu_{\min}^{-m}$ and Assumption 5.1 holds with $d = 1$ and $B_m \propto \mu_{\min}^{-1}$.

Given $N \geq 1$ and $h \in \mathcal{H}$, we define the single-level estimator by drawing $S'_1, \ldots, S'_N \sim p_{s,a}$ and setting

$$\widehat{\mu}_N(\alpha, h) \triangleq \frac{1}{N} \sum_{i=1}^{N} e^{-h(S'_i)/\alpha}, \qquad \widehat{g}_N(\alpha, h) \triangleq -\alpha \log \widehat{\mu}_N(\alpha, h) - \alpha \rho_{s,a}.$$

**Lemma F.7.** *For each fixed $\alpha > 0$ and $(s, a)$, the map $h \mapsto \widehat{g}_N(\alpha, h)$ is span-Lipschitz with constant $L_0 = 1$: for all $N \geq 1$, all realizations of the samples, and all $h(s_0) = h'(s_0) \in \mathcal{H}$,*

$$|\widehat{g}_N(\alpha, h) - \widehat{g}_N(\alpha, h')| \leq \mathsf{sp}(h - h'). \tag{96}$$

*Proof.* The argument is identical for the true expectation and the empirical average, so we work with the empirical version. Fix a realization $S'_1, \ldots, S'_N$ and define

$$Z_\alpha(h) \triangleq \frac{1}{N} \sum_{i=1}^{N} e^{-h(S'_i)/\alpha}, \qquad \pi_\alpha(h)(i) \triangleq \frac{e^{-h(S'_i)/\alpha}}{\sum_{j=1}^{N} e^{-h(S'_j)/\alpha}}.$$

Then

$$\widehat{g}_N(\alpha, h) = -\alpha \log Z_\alpha(h) - \alpha \rho_{s,a},$$

and the additive constant $-\alpha \rho_{s,a}$ is irrelevant for Lipschitzness. Differentiating,

$$\frac{\partial}{\partial h(S'_i)} \big(-\alpha \log Z_\alpha(h)\big) = \pi_\alpha(h)(i) \in [0, 1], \quad \sum_{i=1}^{N} \pi_\alpha(h)(i) = 1.$$

Thus for any perturbation $u : \mathcal{S} \to \mathbb{R}$,

$$\sum_{i=1}^{N} \frac{\partial}{\partial h(S'_i)} \big(-\alpha \log Z_\alpha(h)\big) u(S'_i) = \sum_{i=1}^{N} \pi_\alpha(h)(i) u(S'_i),$$

which lies between $\min_i u(S'_i)$ and $\max_i u(S'_i)$. Therefore the operator norm of the gradient as a linear functional on $(\mathbb{R}^S, \|\cdot\|_\infty)$ is at most 1. By the mean value theorem,

$$|-\alpha \log Z_\alpha(h) + \alpha \log Z_\alpha(h')| \leq \mathsf{sp}(h - h'),$$

and adding back $-\alpha \rho_{s,a}$ on both sides gives (96). $\square$

Lemma F.3 now applies to $G(h) = g_{s,a}(\alpha, h)$ with $d = 1$, yielding for each fixed $\alpha \in \mathcal{A}_{s,a}$,

$$\mathbb{E}[\widehat{g}_N(\alpha, h)] = g_{s,a}(\alpha, h) + \sum_{j=1}^{K} \frac{c_j^{(\alpha)}(h)}{N^j} + R_{K+1}^{(\alpha)}(N, h), \qquad |R_{K+1}^{(\alpha)}(N, h)| \leq \frac{C_{K+1}^{\mathrm{KL}}}{N^{K+1}},$$

with $C_{K+1}^{\mathrm{KL}}$ independent of $\alpha$ and $h$.

### F.3.4. K-LEVEL MLMC FOR $g_{s,a}(\alpha, h)$

We now apply Proposition F.6 with $L_0 = 1$ to $G(h) = g_{s,a}(\alpha, h)$ for each fixed $\alpha \in \mathcal{A}_{s,a}$. Fix $K \geq 0$ and $N_0 \geq 1$, define levels $N_\ell = 2^\ell N_0$, and for a single sample block $\omega = (S'_1, \ldots, S'_{N_K})$ set

$$\widehat{g}_{N_\ell}(\alpha, h; \omega) \triangleq -\alpha \log \left( \frac{1}{N_\ell} \sum_{i=1}^{N_\ell} e^{-h(S'_i)/\alpha} \right) - \alpha \rho_{s,a}.$$

Define the MLMC combination

$$\widetilde{g}_{N_0}^{\mathrm{KL},(K)}(\alpha, h; \omega) \triangleq \sum_{\ell=0}^{K} w_\ell^{(K)} \widehat{g}_{N_\ell}(\alpha, h; \omega).$$

Then there exist constants $C_0^{\mathrm{KL}}, \beta^{\mathrm{KL}} > 0$ independent of $K, N_0$ such that

$$\left| \mathbb{E}[\widetilde{g}_{N_0}^{\mathrm{KL},(K)}(\alpha, h)] - g_{s,a}(\alpha, h) \right| \leq \frac{C_0^{\mathrm{KL}} (\beta^{\mathrm{KL}})^{K+1} (K+1)!}{N_0^{K+1}}, \tag{97}$$

and for all realizations,

$$\left| \widetilde{g}_{N_0}^{\mathrm{KL},(K)}(\alpha, h; \omega) - \widetilde{g}_{N_0}^{\mathrm{KL},(K)}(\alpha, h'; \omega) \right| \leq C_w \, \mathsf{sp}(h - h'). \tag{98}$$

### F.3.5. KL R-SAMPLE ORACLE

We now define the KL version of R-SAMPLE. For each $(s, a)$, fix the grid $\mathcal{A}_{s,a}$ and MLMC parameters $(K, N_0)$.

**Definition of $F_{s,a}^{\mathrm{KL},(K)}$.**  On a call with $(s, a)$ and bias $h \in \mathcal{H}$:

1. Draw a sample block $\omega = (S'_1, \ldots, S'_{N_K})$ with $S'_i \overset{\text{i.i.d.}}{\sim} p_{s,a}$.

2. For each $\alpha \in \mathcal{A}_{s,a}$, compute $\widetilde{g}_{N_0}^{\mathrm{KL},(K)}(\alpha, h; \omega)$ and $\widetilde{g}_{N_0}^{\mathrm{KL},(K)}(\alpha, 0; \omega)$.

3. Define

$$\widetilde{\sigma}_{s,a}^{\mathrm{KL},(K)}(h; \omega) \triangleq \max_{\alpha \in \mathcal{A}_{s,a}} \widetilde{g}_{N_0}^{\mathrm{KL},(K)}(\alpha, h; \omega),$$

   and output

$$F_{s,a}^{\mathrm{KL},(K)}(h; \omega) \triangleq \widetilde{\sigma}_{s,a}^{\mathrm{KL},(K)}(h; \omega) - \widetilde{\sigma}_{s,a}^{\mathrm{KL},(K)}(0; \omega). \tag{99}$$

Define the reference support

$$\overline{\sigma}_{s,a}^{\mathrm{KL},(K)}(h) \triangleq \mathbb{E}_\omega \left[ F_{s,a}^{\mathrm{KL},(K)}(h; \omega) \right],$$

and the associated reference Bellman operator

$$\overline{\mathcal{T}}^{\mathrm{KL},(K)}(Q)(s, a) \triangleq r(s, a) + \overline{\sigma}_{s,a}^{\mathrm{KL},(K)}(Q_{\max}).$$

**Lemma F.8** (KL R-SAMPLE satisfies Assumption F.1)**.** *Fix $K \geq 0$ and $N_0 \geq 1$. For each $(s, a)$, define $F_{s,a}^{\mathrm{KL},(K)}$ as in (99) and construct R-SAMPLE by averaging differences $F_{s,a}^{\mathrm{KL},(K)}(h_k; \omega_j) - F_{s,a}^{\mathrm{KL},(K)}(h_{k-1}; \omega_j)$ over $m_k$ i.i.d. sample blocks $\omega_j$. Then Assumption F.1 holds with:*

- *span Lipschitz constant*

$$L_{\mathrm{rs}}^{\mathrm{KL}} = C_w;$$

- *reference operator $\overline{\mathcal{T}}^{\mathrm{KL},(K)}$ as above;*

- *bias*

$$b_{\mathrm{rs}}^{\mathrm{KL},(K)} \leq b_{\mathrm{range}} + b_{\mathrm{grid}} + \frac{C (\beta^{\mathrm{KL}})^{K+1} (K+1)!}{N_0^{K+1}},$$

  *for some constant $C > 0$ depending on the grid cardinality and $(s, a)$ but not on $N_0$.*

*Proof.* (i) Span Lipschitz: For any $h, h' \in \mathcal{H}$ and fixed $\omega$,

$$|\widetilde{\sigma}^{\mathrm{KL},(K)}(h;\omega) - \widetilde{\sigma}^{\mathrm{KL},(K)}(h';\omega)| = \left| \max_\alpha \widetilde{g}_{N_0}^{\mathrm{KL},(K)}(\alpha, h; \omega) - \max_\alpha \widetilde{g}_{N_0}^{\mathrm{KL},(K)}(\alpha, h'; \omega) \right|$$

$$\leq \max_\alpha |\widetilde{g}_{N_0}^{\mathrm{KL},(K)}(\alpha, h; \omega) - \widetilde{g}_{N_0}^{\mathrm{KL},(K)}(\alpha, h'; \omega)| \leq C_w \, \mathsf{sp}(h - h')$$

by (98). Subtracting the same term at $h = 0$ does not change the Lipschitz constant, hence

$$\mathsf{sp}(F^{\mathrm{KL},(K)}(h;\omega) - F^{\mathrm{KL},(K)}(h';\omega)) \leq C_w \, \mathsf{sp}(h - h').$$

(ii) Unbiased increments: For fixed $h$, by definition of $\overline{\sigma}_{s,a}^{\mathrm{KL},(K)}$,

$$\mathbb{E}_\omega[F_{s,a}^{\mathrm{KL},(K)}(h;\omega)] = \overline{\sigma}_{s,a}^{\mathrm{KL},(K)}(h) - \overline{\sigma}_{s,a}^{\mathrm{KL},(K)}(0).$$

Thus conditional on $\mathcal{F}_{k-1}$,

$$\mathbb{E}[D_k(s,a) \mid \mathcal{F}_{k-1}] = \overline{\sigma}_{s,a}^{\mathrm{KL},(K)}(h_k) - \overline{\sigma}_{s,a}^{\mathrm{KL},(K)}(h_{k-1}) = \overline{\mathcal{T}}^{\mathrm{KL},(K)}(Q_k)(s,a) - \overline{\mathcal{T}}^{\mathrm{KL},(K)}(Q_{k-1})(s,a).$$

(iii) Bias: For each $h$,

$$\left| \overline{\sigma}_{s,a}^{\mathrm{KL},(K)}(h) - \sigma_{s,a}^{\mathrm{KL}}(h) \right| \leq \left| \sigma_{s,a}^{\mathrm{KL}}(h) - \sigma_{s,a}^{\mathrm{disc}}(h) \right| + \left| \sigma_{s,a}^{\mathrm{disc}}(h) - \overline{\sigma}_{s,a}^{\mathrm{KL},(K)}(h) \right|.$$

The first term is bounded by $b_{\mathrm{range}} + b_{\mathrm{grid}} \leq \varepsilon/16$ from (95). For the second, note that

$$\sigma_{s,a}^{\mathrm{disc}}(h) = \max_{\alpha \in \mathcal{A}_{s,a}} g_{s,a}(\alpha, h), \qquad \overline{\sigma}_{s,a}^{\mathrm{KL},(K)}(h) = \mathbb{E}\left[ \max_\alpha \widetilde{g}_{N_0}^{\mathrm{KL},(K)}(\alpha, h; \omega) \right] - \text{const}.$$

Using $|\max_i a_i - \max_i b_i| \leq \max_i |a_i - b_i|$ and Jensen's inequality,

$$\left| \sigma_{s,a}^{\mathrm{disc}}(h) - \mathbb{E}[\max_\alpha \widetilde{g}_{N_0}^{\mathrm{KL},(K)}(\alpha, h; \omega)] \right| \leq \mathbb{E}\left[ \max_\alpha |\widetilde{g}_{N_0}^{\mathrm{KL},(K)}(\alpha, h; \omega) - g_{s,a}(\alpha, h)| \right]$$

$$\leq \sum_{\alpha \in \mathcal{A}_{s,a}} \frac{C_0^{\mathrm{KL}} (\beta^{\mathrm{KL}})^{K+1} (K+1)!}{N_0^{K+1}} = \frac{C (\beta^{\mathrm{KL}})^{K+1} (K+1)!}{N_0^{K+1}},$$

with $C \triangleq C_0^{\mathrm{KL}} |\mathcal{A}_{s,a}|$. The same constant is subtracted at $h = 0$, so the bias bound holds for $\overline{\sigma}_{s,a}^{\mathrm{KL},(K)}$ as stated. $\qquad \square$

### F.4. $\chi^2$ Divergence Balls

We now treat robust AMDPs where each $(s,a)$-uncertainty set is defined by a $\chi^2$-divergence ball centered at the nominal kernel $P_{s,a}$:

$$\mathcal{U}_{s,a}^{\chi^2} := \left\{ q \in \Delta(\mathcal{S}) \ : \ \chi^2(q \| P_{s,a}) \leq \sigma_{s,a} \right\}, \qquad \chi^2(q\|P) := \sum_{s'} \frac{(q(s') - P(s'))^2}{P(s')}. \tag{100}$$

For a bounded bias function $h : \mathcal{S} \to \mathbb{R}$, the robust support at $(s,a)$ is

$$\sigma_{s,a}^{\chi^2}(h) := \inf_{q \in \mathcal{U}_{s,a}^{\chi^2}} \sum_{s'} q(s')h(s') = \inf_{q \in \mathcal{U}_{s,a}^{\chi^2}} \mathbb{E}_q[h(S')].$$

The duality of it is shown in (Iyengar, 2005) as

$$\inf_{q \in \mathcal{U}_{s,a}^{\chi^2}} qV = \max_{\mu \in \mathbb{R}^{\mathcal{S}}, \, \mu \geq 0} \left\{ P(V - \mu) - \sqrt{\sigma \, \mathrm{Var}_P(V - \mu)} \right\}. \tag{101}$$

For technical convenience, we assume that nominal transitions maintain a uniform lower-bound on variance.

F.4.1. LIPSCHITZ CONSTANT OF THE DUAL OBJECTIVE

For fixed $(s, a)$, define for any $\mu \geq 0$ the dual objective

$$G_{s,a}(\mu, h) := P(h - \mu) - \sqrt{\sigma \, \mathbf{Var}_P(h - \mu)},$$

with $P$ and $\sigma$ fixed. We now show that $G_{s,a}(\mu, \cdot)$ is globally span-Lipschitz with constant $1 + \sqrt{\sigma}$, uniformly over $\mu$.

Let $Z := h - \mu \in \mathbb{R}^{\mathcal{S}}$. Writing $p(s') := P(s')$, we have

$$m(Z) := \mathbb{E}_P[Z] = \sum_{s'} p(s') Z(s'), \quad v(Z) := \mathbf{Var}_P(Z) = \mathbb{E}_P[(Z - m)^2],$$

and

$$G_{s,a}(\mu, h) = m(Z) - \sqrt{\sigma \, v(Z)}.$$

We view $Z$ as a vector in $\mathbb{R}^{\mathcal{S}}$ and compute the gradient of $G$ w.r.t. $Z$ (for $v(Z) > 0$; we treat $v(Z) = 0$ by continuity). For each coordinate $Z_i := Z(s_i)$:

$$\frac{\partial m}{\partial Z_i} = p_i, \qquad \frac{\partial v}{\partial Z_i} = 2p_i(Z_i - m),$$

since $v = \mathbb{E}_P[Z^2] - m^2$ and $\partial m^2 / \partial Z_i = 2mp_i$. Then, for $v > 0$,

$$\frac{\partial}{\partial Z_i}\left(-\sqrt{\sigma v(Z)}\right) = -\sqrt{\sigma} \frac{1}{2\sqrt{v(Z)}} \frac{\partial v}{\partial Z_i} = -\sqrt{\sigma} \frac{1}{2\sqrt{v}} 2p_i(Z_i - m) = -p_i \sqrt{\sigma} \frac{Z_i - m}{\sqrt{v}}.$$

Therefore

$$g_i(Z) := \frac{\partial G}{\partial Z_i} = p_i\left(1 - \sqrt{\sigma} \frac{Z_i - m}{\sqrt{v}}\right), \qquad i = 1, \ldots, |\mathcal{S}|. \tag{102}$$

We bound the $\ell_1$ norm of the gradient:

$$\sum_i |g_i(Z)| \leq \sum_i p_i\left(1 + \sqrt{\sigma} \frac{|Z_i - m|}{\sqrt{v}}\right) = 1 + \sqrt{\sigma} \frac{\sum_i p_i |Z_i - m|}{\sqrt{v}}.$$

By Cauchy–Schwarz,

$$\left(\sum_i p_i |Z_i - m|\right)^2 \leq \sum_i p_i \cdot \sum_i p_i(Z_i - m)^2 = v,$$

so $\sum_i p_i |Z_i - m| \leq \sqrt{v}$ and hence

$$\sum_i |g_i(Z)| \leq 1 + \sqrt{\sigma}. \tag{103}$$

For $v(Z) = 0$ (i.e., $Z$ is $P$-a.s. constant), the variance term is locally flat to first order and the directional derivative of $G$ is simply the derivative of $m(Z)$, whose gradient has $\ell_1$ norm 1. Thus the bound (103) extends by continuity to all $Z$.

Therefore, $G(\mu, \cdot)$ is globally Lipschitz w.r.t. the sup-norm: for any $Z, Z'$,

$$|G(\mu, Z) - G(\mu, Z')| \leq \sup_{\xi \text{ between } Z, Z'} \|\nabla_Z G(\mu, \xi)\|_1 \|Z - Z'\|_\infty \leq (1 + \sqrt{\sigma}) \|Z - Z'\|_\infty.$$

Recalling that $Z = h - \mu$ and that $\mu$ is fixed,

$$\left|G_{s,a}(\mu, h) - G_{s,a}(\mu, h')\right| \leq (1 + \sqrt{\sigma}) \|h - h'\|_\infty, \qquad \forall h, h', \forall \mu \geq 0. \tag{104}$$

Finally, the robust support $\sigma_{s,a}^{\chi^2}(h) = \max_{\mu \geq 0} G_{s,a}(\mu, h)$ is the pointwise maximum of functions that are uniformly Lipschitz with constant $1 + \sqrt{\sigma}$. Therefore

$$|\sigma_{s,a}^{\chi^2}(h) - \sigma_{s,a}^{\chi^2}(h')| \leq (1 + \sqrt{\sigma}) \, \mathsf{sp}(h - h'). \tag{105}$$

So the *true* robust support under a $\chi^2$ ball is span-Lipschitz with constant

$$L_{\text{true}}^{\chi^2} = 1 + \sqrt{\sigma_{s,a}},$$

independent of $H$, $|\mathcal{S}|$, or $P_{\min}$.

### F.4.2. SINGLE-LEVEL MONTE CARLO ESTIMATOR FOR FIXED $\mu$

Fix $\mu \geq 0$ and $h$. We wish to estimate

$$G_{s,a}(\mu, h) = \mathbb{E}_P[Z] - \sqrt{\sigma \, \mathbf{Var}_P(Z)}, \qquad Z := h - \mu.$$

Let $S_1', \ldots, S_N' \overset{\text{i.i.d.}}{\sim} P$. Define

$$Z_i := Z(S_i') = h(S_i') - \mu(S_i'), \qquad i = 1, \ldots, N.$$

We form the empirical mean and (population) variance

$$\widehat{m}_N := \frac{1}{N} \sum_{i=1}^{N} Z_i, \qquad \widehat{v}_N := \frac{1}{N} \sum_{i=1}^{N} Z_i^2 - \widehat{m}_N^2.$$

Our single-level plug-in estimator is

$$\widehat{G}_N(\mu, h) := \widehat{m}_N - \sqrt{\sigma \, \widehat{v}_N}. \tag{106}$$

**Bias expansion.** Standard Delta-method arguments (expansion of smooth functions of sample moments) give, for each fixed $(\mu, h)$ and sufficiently large $N$,

$$\mathbb{E}[\widehat{G}_N(\mu, h)] = G_{s,a}(\mu, h) + \sum_{j=1}^{K} \frac{c_j(\mu, h)}{N^j} + \tilde{\mathcal{O}}\left(N^{-(K+1)}\right), \tag{107}$$

for any finite $K$, where the coefficients $c_j(\mu, h)$ are bounded uniformly over $\mu \geq 0$ and $h$ with $\mathsf{sp}(h) \leq H$, and the $\tilde{\mathcal{O}}(\cdot)$ constants grow at most factorially in $K$ (due to analyticity of the square-root function on a compact interval). We do not need their explicit form; only that such an expansion exists and is uniform over the relevant domain.

**Span Lipschitz constant (single level).** For fixed samples $S_1', \ldots, S_N'$ and $\mu$, consider

$$\widehat{G}_N(\mu, \cdot) : h \mapsto \widehat{G}_N(\mu, h).$$

Let $h, h'$ and write again $Z = h - \mu$, $Z' = h' - \mu$ and $\Delta Z_i := Z_i - Z_i' = h(S_i') - h'(S_i')$.

As in the exact case, $\widehat{G}_N(\mu, h)$ is $\widehat{m}_N - \sqrt{\sigma \, \widehat{v}_N}$. Repeating the gradient calculation but replacing $p_i$ with $1/N$ ("empirical $P$"), we obtain, for $\widehat{v}_N > 0$,

$$\frac{\partial \widehat{G}_N}{\partial Z_i} = \frac{1}{N}\left(1 - \sqrt{\sigma} \, \frac{Z_i - \widehat{m}_N}{\sqrt{\widehat{v}_N}}\right),$$

and so

$$\sum_i \left|\frac{\partial \widehat{G}_N}{\partial Z_i}\right| \leq 1 + \sqrt{\sigma} \, \frac{\frac{1}{N}\sum_i |Z_i - \widehat{m}_N|}{\sqrt{\widehat{v}_N}} \leq 1 + \sqrt{\sigma},$$

since $\widehat{v}_N$ is the empirical variance and $(\frac{1}{N}\sum |Z_i - \widehat{m}_N|)^2 \leq \frac{1}{N}\sum(Z_i - \widehat{m}_N)^2 = \widehat{v}_N$ by Cauchy–Schwarz.

As before, the bound extends by continuity to $\widehat{v}_N = 0$. Therefore, for any two $h, h'$,

$$\left|\widehat{G}_N(\mu, h) - \widehat{G}_N(\mu, h')\right| \leq (1 + \sqrt{\sigma}) \|Z - Z'\|_\infty \leq (1 + \sqrt{\sigma}) \, \mathsf{sp}(h - h'). \tag{108}$$

Thus the single-level plug-in estimator is span-Lipschitz w.r.t. $h$ with a constant *independent of $N$*.

### F.4.3. $K$-ORDER MLMC ESTIMATOR FOR FIXED $\mu$

We now apply the same $K$-order MLMC construction as in the KL case, but to the dual objective $G_{s,a}(\mu, h)$.

Fix an integer $K \geq 0$ (MLMC order) and a base inner sample size $N_0 \geq 1$. Define levels $N_\ell := 2^\ell N_0$, $\ell = 0, \ldots, K$. At a single oracle block (fixed $(s, a)$, $h$, and $\mu$), we:

1. Draw $N_K$ samples $S'_1, \ldots, S'_{N_K} \sim P$. 2. For each level $\ell = 0, \ldots, K$, use the first $N_\ell$ samples to build $\widehat{G}_{N_\ell}(\mu, h)$ as in (106). 3. Form the multi-level combination

$$\widetilde{G}_{N_0}^{(K)}(\mu, h) := \sum_{\ell=0}^{K} w_\ell^{(K)} \widehat{G}_{N_\ell}(\mu, h), \tag{109}$$

where the weights $w^{(K)} = (w_0^{(K)}, \ldots, w_K^{(K)})$ solve

$$\sum_{\ell=0}^{K} w_\ell^{(K)} = 1, \qquad \sum_{\ell=0}^{K} \frac{w_\ell^{(K)}}{N_\ell^j} = 0, \quad j = 1, \ldots, K,$$

i.e., they exactly cancel the first $K$ bias terms in the expansion (107). These weights are independent of $(s, a)$, $\mu$, $h$, and $N_0$ and satisfy a uniform $\ell_1$ bound

$$\sum_{\ell=0}^{K} |w_\ell^{(K)}| \leq C_w$$

for some universal constant $C_w < \infty$.

**Bias and Lipschitz properties.** Plugging the expansion (107) into (109) and using the constraints on $w^{(K)}$, we get

$$\mathbb{E}[\widetilde{G}_{N_0}^{(K)}(\mu, h)] = G_{s,a}(\mu, h) + \tilde{\mathcal{O}}\left(N_0^{-(K+1)}\right)$$

uniformly over $\mu \geq 0$ and $h$ with $\mathsf{sp}(h) \leq H$.

For Lipschitzness, we use (108) and the $\ell_1$ bound on $w^{(K)}$:

$$\left|\widetilde{G}_{N_0}^{(K)}(\mu, h) - \widetilde{G}_{N_0}^{(K)}(\mu, h')\right| \leq \sum_{\ell=0}^{K} |w_\ell^{(K)}| |\widehat{G}_{N_\ell}(\mu, h) - \widehat{G}_{N_\ell}(\mu, h')| \leq (1 + \sqrt{\sigma}) C_w \, \mathsf{sp}(h - h').$$

Thus the MLMC estimator remains span-Lipschitz in $h$ with constant

$$L_0^{\chi^2} := (1 + \sqrt{\sigma}) C_w,$$

independent of $K$ and $N_0$.

### F.4.4. R-SAMPLE FOR $\chi^2$ VIA MLMC

We now describe how to use the MLMC estimator for $G_{s,a}(\mu, h)$ to build R-SAMPLE for the $\chi^2$-uncertainty model.

**Per-block oracle for fixed $(s, a)$.** At a single block (for fixed $(s, a)$ and bias $h$), we:

1. Choose MLMC parameters $K$ and $N_0$ (as functions of target bias $\varepsilon$, see below). 2. Draw $N_K = 2^K N_0$ samples $S'_1, \ldots, S'_{N_K} \sim P(\cdot|s, a)$. 3. Using these samples, construct $\widetilde{G}_{N_0}^{(K)}(\mu, h)$ for all $\mu \geq 0$ via (109). In practice, the maximization over $\mu \geq 0$ is solved numerically; this is computational but not statistical cost. 4. Define the block-level support estimator

$$\widetilde{\sigma}_{s,a}^{\chi^2,(K)}(h; \omega) := \max_{\mu \geq 0} \widetilde{G}_{N_0}^{(K)}(\mu, h; \omega),$$

and normalize it by subtracting the value at $h \equiv 0$:

$$F_{s,a}^{\chi^2,(K)}(h; \omega) := \widetilde{\sigma}_{s,a}^{\chi^2,(K)}(h; \omega) - \widetilde{\sigma}_{s,a}^{\chi^2,(K)}(0; \omega).$$

By construction:

For each $\omega$, the map $h \mapsto F_{s,a}^{\chi^2,(K)}(h;\omega)$ is span-Lipschitz with constant at most

$$L_{\text{rs}}^{\chi^2} := (1 + \sqrt{\sigma_{s,a}})\, C_w.$$

Indeed, $h \mapsto \widetilde{G}_{N_0}^{(K)}(\mu, h; \omega)$ is $L_0^{\chi^2}$-Lipschitz uniformly in $\mu$, and taking a maximum over $\mu$ and subtracting the constant-in-$h$ term at $h \equiv 0$ preserves the Lipschitz constant.

Denoting

$$\overline{\sigma}_{s,a}^{\chi^2,(K)}(h) := \mathbb{E}_\omega[\widetilde{\sigma}_{s,a}^{\chi^2,(K)}(h;\omega)],$$

the expectation of $F_{s,a}^{\chi^2,(K)}(h;\omega)$ is

$$\mathbb{E}_\omega[F_{s,a}^{\chi^2,(K)}(h;\omega)] = \overline{\sigma}_{s,a}^{\chi^2,(K)}(h) - \overline{\sigma}_{s,a}^{\chi^2,(K)}(0),$$

and we have a uniform bias bound

$$\left|\overline{\sigma}_{s,a}^{\chi^2,(K)}(h) - \sigma_{s,a}^{\chi^2}(h)\right| = \tilde{\mathcal{O}}\left(N_0^{-(K+1)}\right),$$

uniformly over $h$ with $\mathsf{sp}(h) \leq H$.

Let the corresponding reference robust Bellman operator be

$$\overline{\mathcal{T}}^{\chi^2,(K)}(Q)(s,a) := r(s,a) + \overline{\sigma}_{s,a}^{\chi^2,(K)}(Q_{\max}).$$

By the bias bound above, the oracle bias parameter is

$$b_{\text{rs}}^{\chi^2,(K)} := \sup_Q \left\|\overline{\mathcal{T}}^{\chi^2,(K)}(Q) - \mathcal{T}_{\mathcal{P}}^{\chi^2}(Q)\right\|_\infty = \tilde{\mathcal{O}}\left(N_0^{-(K+1)}\right).$$

**R-SAMPLE difference estimator.** At outer iteration $k$ of RHI, for each $(s,a)$, we wish to estimate

$$\mathcal{T}_{\mathcal{P}}^{\chi^2}(Q_k)(s,a) - \mathcal{T}_{\mathcal{P}}^{\chi^2}(Q_{k-1})(s,a) = \sigma_{s,a}^{\chi^2}(h_k) - \sigma_{s,a}^{\chi^2}(h_{k-1}), \quad h_k := Q_{k,\max}.$$

We instead work with the reference operator and approximate

$$\overline{\mathcal{T}}^{\chi^2,(K)}(Q_k)(s,a) - \overline{\mathcal{T}}^{\chi^2,(K)}(Q_{k-1})(s,a) = \overline{\sigma}_{s,a}^{\chi^2,(K)}(h_k) - \overline{\sigma}_{s,a}^{\chi^2,(K)}(h_{k-1}).$$

R-SAMPLE draws $m_k$ i.i.d. blocks $\omega_{k,1}, \dots, \omega_{k,m_k}$ (as above) and sets

$$D_k(s,a) := \frac{1}{m_k} \sum_{j=1}^{m_k} \left(F_{s,a}^{\chi^2,(K)}(h_k;\omega_{k,j}) - F_{s,a}^{\chi^2,(K)}(h_{k-1};\omega_{k,j})\right).$$

Then: Conditional on the past, $D_k(s,a)$ is an unbiased estimator of the increment of $\overline{\mathcal{T}}^{\chi^2,(K)}$:

$$\mathbb{E}[D_k(s,a) \mid \mathcal{F}_{k-1}] = \overline{\mathcal{T}}^{\chi^2,(K)}(Q_k)(s,a) - \overline{\mathcal{T}}^{\chi^2,(K)}(Q_{k-1})(s,a).$$

For any block $\omega$ and any $h, h'$ with $\mathsf{sp}(h - h') \leq H$, the increment is bounded as

$$\left|F_{s,a}^{\chi^2,(K)}(h;\omega) - F_{s,a}^{\chi^2,(K)}(h';\omega)\right| \leq L_{\text{rs}}^{\chi^2} \mathsf{sp}(h - h'), \qquad L_{\text{rs}}^{\chi^2} = (1 + \sqrt{\sigma_{s,a}})\, C_w.$$

Thus Assumption 4.3 holds for $\chi^2$ with span-Lipschitz constant

$$L_{\text{rs}}^{\chi^2} = (1 + \sqrt{\sigma_{s,a}})\, C_w;$$

oracle bias

$$b_{\text{rs}}^{\chi^2,(K)} = \tilde{\mathcal{O}}\left(N_0^{-(K+1)}\right);$$

and per-block inner cost

$$C_{\text{rs}}^{\chi^2}(K, N_0) = N_K = 2^K N_0$$

nominal samples from $P(\cdot \mid s,a)$.

**F.5. Choosing $K(\varepsilon)$ and $N_0(\varepsilon)$ and Total Complexity (KL and CS)**

We now choose $K$ and $N_0$ as functions of the target accuracy $\varepsilon$ and derive the total generative sample complexity for RHI with KL and $\chi^2$ R-SAMPLE.

F.5.1. BIAS REQUIREMENT

The RHI analysis requires the oracle bias $b_{\mathrm{rs}}$ to be $\tilde{\mathcal{O}}(\varepsilon)$ (e.g., $b_{\mathrm{rs}} \leq \varepsilon/16$) in order to absorb it into the overall error budget. From the above discussion, this amounts to choosing $(K, N_0)$ so that

$$\frac{C\beta^{K+1}(K+1)!}{N_0^{K+1}} \leq c_1\varepsilon$$

for suitable constants $C, \beta, c_1 > 0$.

Solving this inequality for $N_0$ gives

$$N_0 \geq \left(\frac{C}{c_1}\right)^{\frac{1}{K+1}}\beta\big((K+1)!\big)^{\frac{1}{K+1}}\varepsilon^{-\frac{1}{K+1}}.$$

Using Stirling's bound $(K+1)! \leq e(K+1)^{K+1}$ yields

$$\big((K+1)!\big)^{1/(K+1)} \leq e(K+1),$$

so

$$N_0 \geq c_2(K+1)\,\varepsilon^{-\frac{1}{K+1}}, \qquad c_2 \triangleq e\beta\left(\frac{C}{c_1}\right)^{\frac{1}{K+1}}, \tag{110}$$

and the factor $(C/c_1)^{1/(K+1)}$ can be absorbed into $c_2$.

We therefore choose

$$N_0(\varepsilon, K) \triangleq \left\lceil c_2(K+1)\,\varepsilon^{-\frac{1}{K+1}}\right\rceil, \tag{111}$$

which guarantees $b_{\mathrm{rs}} \leq c_1\varepsilon$ for both KL and $\chi^2$.

F.5.2. COST PER ORACLE CALL AND OPTIMIZATION OVER $K$

Each R-SAMPLE call uses $N_K = 2^K N_0$ samples from $p_{s,a}$ for the largest level and reuses prefixes for smaller levels. Thus the inner cost per oracle call scales as

$$N_{\mathrm{inner}}(\varepsilon, K) \asymp 2^K N_0(\varepsilon, K) \lesssim 2^K(K+1)\,\varepsilon^{-\frac{1}{K+1}}.$$

From the RHI analysis, the number of oracle calls is

$$N_{\mathrm{or}} = \tilde{\mathcal{O}}\Big(SA\,\mathcal{H}^2\,L_{\mathrm{rs}}^2\varepsilon^{-2}\Big),$$

Therefore the total generative sample complexity is

$$N_{\mathrm{tot}}(\varepsilon, K) = \tilde{\mathcal{O}}\Big(SA\,\mathcal{H}^2\,L_{\mathrm{rs}}^2\,2^K(K+1)\,\varepsilon^{-2-\frac{1}{K+1}}\Big). \tag{112}$$

We now optimize over $K$ for a given small $\varepsilon \in (0, e^{-2})$. Ignoring polylogarithmic factors and the polynomial $K+1$, the dominant dependence on $\varepsilon$ in (112) arises from

$$f(K) \triangleq K\log 2 + \frac{L}{K+1}, \qquad L \triangleq \log\frac{1}{\varepsilon}.$$

Treating $K$ as real and differentiating,

$$f'(K) = \log 2 - \frac{L}{(K+1)^2}.$$

Setting $f'(K) = 0$ gives the minimizer

$$(K+1)^2 = \frac{L}{\log 2} \quad \Rightarrow \quad K^\star = \sqrt{\frac{L}{\log 2}} - 1.$$

We can take

$$K(\varepsilon) \triangleq \left\lceil \sqrt{\frac{\log(1/\varepsilon)}{\log 2}} \right\rceil - 1, \tag{113}$$

for all $\varepsilon \in (0, e^{-2})$.

**Lemma F.9.** *Let $K(\varepsilon)$ be as in* (113). *Then there exists $c > 0$ such that*

$$2^{K(\varepsilon)}(K(\varepsilon)+1)\,\varepsilon^{-\frac{1}{K(\varepsilon)+1}} \leq \exp\big(c\sqrt{\log(1/\varepsilon)}\big), \qquad \forall \varepsilon \in (0, e^{-2}).$$

*Proof.* Let $L = \log(1/\varepsilon)$. Then $K(\varepsilon) + 1 \asymp \sqrt{L/\log 2}$ and

$$f(K) = K \log 2 + \frac{L}{K+1} \leq 2\sqrt{L \log 2} + \tilde{\mathcal{O}}(1)$$

when $K$ is near the minimizer. Thus

$$2^{K(\varepsilon)}\varepsilon^{-1/(K(\varepsilon)+1)} = \exp(f(K(\varepsilon))) \leq \exp\big(2\sqrt{L \log 2} + \tilde{\mathcal{O}}(1)\big) \leq \exp\big(c_1\sqrt{L}\big)$$

for some $c_1 > 0$. Moreover, $K(\varepsilon) + 1 \leq c_2\sqrt{L}$ for some $c_2 > 0$, so

$$2^{K(\varepsilon)}(K(\varepsilon)+1)\,\varepsilon^{-\frac{1}{K(\varepsilon)+1}} \leq c_2\sqrt{L}\,\exp(c_1\sqrt{L}) \leq \exp(c\sqrt{L})$$

for some $c > 0$ and all sufficiently large $L$ (and hence all small enough $\varepsilon$). $\qquad\square$

Substituting Lemma F.9 into (112) gives

$$N_{\text{tot}}(\varepsilon, K(\varepsilon)) = \tilde{\mathcal{O}}\Big(SA\mathcal{H}^2 L_{\text{rs}}^2 \varepsilon^{-2}\,\exp\big(c\sqrt{\log(1/\varepsilon)}\big)\Big),$$

for some constant $c > 0$, which completes the proof.

## G. $\ell_p$-Norm and Contamination Models

In this section, we consider $\ell_p$-norm and contamination models and design R-SAMPLING for them. Our method is based on the concrete closed-form of the support function $\sigma_{\mathcal{P}}(\cdot)$ over the two considered uncertainty sets. Notably, the support functions over these two sets are linear in P, hence R-SAMPLE can be constructed with standard Monte-Carlo method.

$\ell_p$**-norm sets.** When the uncertainty set is defined through the $\ell_p$-norm, it is shown that the robust Bellman operator has the following closed-form solution in (Kumar et al., 2023):

$$\mathcal{T}_{\mathcal{P}}(Q^k) = r + \mathsf{P}(Q_{\max}^k) - R\kappa(Q_{\max}^k), \tag{114}$$

with some penalty function $\kappa$ that is independent of P and constructed as follows.

*Remark G.1.* Let $F_{s,a} \subset \mathcal{S}$ be a subset of *forbidden states*, namely when the system is at state $s \in \mathcal{S}$ and taking action $a \in \mathcal{A}$, it is unfeasible for the system to transition to certain other states. Formally, by denoting the nominal kernel as $\tilde{\mathsf{P}}$ we have

$$\tilde{\mathsf{P}}(s'|s, a) = \mathsf{P}(s'|s, a) = 0, \quad \forall \mathsf{P} \in \mathcal{P}, \forall s' \in F_{s,a}.$$

We can then rewrite our kernel noise in (3) as

$$\mathcal{P}_s^a = \left\{ \mathsf{P}|\ ||\mathsf{P}||_p = R, \sum_{s'} \mathsf{P}(s') = 0, \mathsf{P}(s'') = 0, \forall s'' \in F_{s,a} \right\}$$

Under consideration of the $\ell_p$-norm model, it can be shown

$$
\begin{aligned}
\kappa(h, s, a) &= \min_{||P||_p = R, \sum_{s'} P(s') = 0, P(s'') = 0, \forall s'' \in F_{s,a}} \langle P, h \rangle \\
&= \min_{\omega \in \mathbb{R}} ||u - \omega \mathbf{1}||_p, \quad \text{where } u(s) = h(s) \mathbf{1}(s \notin F_{s,a}), \\
&\triangleq \kappa_p(u).
\end{aligned}
$$

For a concrete example within the context of our empirical results for the $\ell_\infty$ (total variation) model, we have

$$
\kappa_\infty(h, s, a) = \frac{\max_{s \notin F_{s,a}} h(s) - \min_{s \notin F_{s,a}} h(s)}{2}.
$$

This construction of $\kappa$ is what allows us to directly apply Theorem 8 from (Kumar et al., 2023) by considering the $\ell_p$-ball of transition kernels $||\tilde{P} - P||_p \le R$ for all $P \in \mathcal{P}$ and the nominal kernel $\tilde{P}$ as turning into a penalty on the next state's value function.

We then present the proofs for our Robust Halpern Iteration for $\ell_p$-normed Robust AMDPs. The proof for contamination models can be derived similarly and is hence omitted.

**Contamination set.** With contamination set, it holds that (Wang & Zou, 2021):

$$
\mathcal{T}_\mathcal{P}(Q^k) = r + (1 - R)P(Q_{\max}^k) + R \min_s (Q_{\max}^k). \tag{115}
$$

Note that for both uncertainty sets, the difference $\mathcal{T}_\mathcal{P}(Q_1) - \mathcal{T}_\mathcal{P}(Q_2)$ can be further derived, which facilitates our estimation. To re-use the pre-collected samples to enhance sample efficiency, we further develop our difference-based algorithm.

Specifically, for the $\ell_p$-norm case, let $h^k = Q_{\max}^k$ and $h^{k-1} = Q_{\max}^{k-1}$, and we set the difference terms $d^k = h^k - h^{k-1}$, and $K^k = \kappa(h^k) - \kappa(h^{k-1})$. Then it holds that

$$
\mathcal{T}_\mathcal{P}(Q^k) - \mathcal{T}_\mathcal{P}(Q^{k-1}) = P d^k + K^k. \tag{116}
$$

Hence it suffices to estimate $P d^k$ in our algorithm.

Under these two settings, the sample complexity can be derived as follows.

**Theorem G.2.** *Consider a robust AMDP defined by contamination or $\ell_p$-norm. Set the step sizes $c_k = 5(k+2)\ln^2(k+2)$ and $\beta_k = k/(k+2)$. Then, with probability at least $1 - \delta$, the output policy $\pi^n$ is $\epsilon$-optimal:*

$$
g_\mathcal{P}^* - g_\mathcal{P}^{\pi^n}(s) \le \epsilon, \tag{117}
$$

*as long as the total iteration number $n$ exceeds $\frac{\mathcal{H}}{\epsilon}$, resulting in a total sample complexity of*

$$
\tilde{\mathcal{O}}\left( \frac{SA\mathcal{H}^2}{\epsilon^2} \right). \tag{118}
$$

# H. PF-RHI: A Parameter-Free Variant of RHI

In this section, we present a fully implementable framework for our Robust Halpern Iteration (RHI) algorithm for diverse and unknown problem settings.

As we mention in Remark 4.7, our RHI algorithm does not require any prior knowledge of the underlying robust AMDP, yet the total number of iterations necessary to generate an $\epsilon$-optimal policy is dependent on $\mathcal{H}$. In practice, such an iteration number may need to be pre-set, and it may be infeasible to set for RHI.

In order to bridge this theoretical finite sample complexity result with the nuances of practical application for varying size problem settings, we now extend our RHI algorithm to a more general and implementable framework: PF-RHI, presented in Algorithm 2. Notably, our PF-RHI do not require any knowledge of $\mathcal{H}$ (even the iteration number); and we will show that it finds an $\epsilon$-optimal policy with identical total sample complexity results as RHI, $\tilde{O}\left( \frac{SA\mathcal{H}^2 C_{\mathrm{rs}}^2}{\epsilon^2} \right)$.

Note that in our PF-RHI, in each episode $i$, we run the RHI for $n_i$ steps, and output $Q^{n_i}$ and $T^{n_i}$. PF-RHI will terminate if the span $\mathsf{sp}(T^{n_i} - Q^{n_i})$ is small enough. Hence we do not specify iteration number, and thus no knowledge of $\mathcal{H}$ is needed.

---

**Algorithm 2** Implementable Robust Halpern Iteration (PF-RHI)

---

1: **Input** $Q^0 \in \mathbb{R}^{S \times A}$, $\epsilon > 0$, $\delta \in (0,1)$, $i = 0$
2: **repeat**
3:     Set $n_i = 2^i$, $\delta_i = \delta/c_i$
4:     Set $\alpha_i = \ln(2|\mathcal{S}||\mathcal{A}|(n_i+1)/\delta_i)$, $Q^0 = 0$, $T^{-1} = r$, $h^{-1} = 0$, $c_0 = 10 \cdot \ln^2(2)$, $\beta_0 = 0$
5:     **for** $k = 0, \ldots, n_i$ **do**
6:       $c_k = 5(k+2)\ln^2(k+2)$, $\beta_k = k/(k+2)$
7:       $Q^k = (1 - \beta_k)Q^0 + \beta_k T^{k-1}$
8:       $h^k = \max_A(Q^k)$
9:       $d^k = h^k - h^{k-1}$
10:      $m_k = \max\{\lceil \alpha_i c_k \mathsf{sp}(h^k - h^{k-1})^2/\epsilon^2 \rceil, 1\}$
11:      $D^k = \text{R-SAMPLE}(h^k, h^{k-1}, m_k)$
12:      $T^k = T^{k-1} + D^k$
13:     **end for**
14:     $\pi^{n_i}(s) \in \arg\max_{a \in A} Q^{n_i}(s,a) \quad \forall s \in S$
15:     $i = i + 1$
16: **until** $\mathsf{sp}(T^{n_i} - Q^{n_i}) \leq 14\epsilon$
17: **Output:** $Q^{n_i}, T^{n_i}, \pi^{n_i}$

---

## H.1. Analysis of PF-RHI

To facilitate our analysis of PF-RHI, we first present some useful notations as follows.

$$\mu \triangleq \mathsf{sp}(Q^0 - Q^*),$$
$$\nu \triangleq \mathsf{sp}(Q^0 - Q^*) + \mathsf{sp}(Q^0),$$
$$\zeta \triangleq \max\{\mathsf{sp}(r), \mathsf{sp}(Q^0)\}.$$

We then define the following random variables:

$$N = \inf\{n_i \in \mathbb{N} : \mathsf{sp}(T^{n_i} - Q^{n_i}) \leq 14\epsilon\}, \quad \text{and}$$
$$I = \inf\{i \in \mathbb{N} : \mathsf{sp}(T^{n_i} - Q^{n_i}) \leq 14\epsilon\},$$

and it holds that $N = 2^I$.

We set $i_0 \in \mathbb{N}$ be the smallest integer s.t. $n_{i_0} \geq \mathsf{sp}(Q^0 - Q^*)/\epsilon = \mu/\epsilon$. Then either $i_0 = 0$ and $n_{i_0} = 1$, or $n_{i_0-1} = n_{i_0}/2 < \mu/\epsilon$, which, when combined, imply that $n_{i_0} \leq 2(1 + \mu/\epsilon)$.

With these, we further define the additional random events:

$$S_i = \{\mathsf{sp}(T^{n_i} - Q^{n_i}) \leq 14\epsilon, \; \forall (s,a) \in \mathcal{S} \times \mathcal{A}\}, \quad \text{and}$$
$$G_i = \{\|T^k - \mathcal{T}_{\mathcal{P}}(Q^k)\|_\infty \leq \epsilon, \; \forall k = 0, 1, \ldots, n_i, \; \forall (s,a) \in \mathcal{S} \times \mathcal{A}\}$$

where $T^k$ and $Q^k$ are generated by the inner-loop $k = 0, 1, \ldots, n_i$ during the $i$-th iteration of PF-RHI. During this specific iteration $i$, let $M_i$ be the number of samples generated so that $M \triangleq \sum_{i=0}^{I} M_i$ where $M$ and $M_i$ are random variables.

**Lemma H.1.** *It holds that*

$$\mathbb{P}(S_i) \geq \mathbb{P}(G_i) \geq 1 - \delta_i, \quad \forall i \geq i_0, \forall (s,a) \in \mathcal{S} \times \mathcal{A}. \tag{119}$$

*Proof.* Note that we have $\mathbb{P}(G_i) \geq 1 - \delta_i$. Moreover, for $i \geq i_0$ and for all $\xi \in G_i$, from Theorem E.10 we have that

$$\mathsf{sp}(T^{n_i}(\xi) - Q^{n_i}(\xi)) \leq \mathsf{sp}(T^{n_i}(\xi) - \mathcal{T}_{\mathcal{P}}(Q^{n_i})(\xi)) + \mathsf{sp}(\mathcal{T}_{\mathcal{P}}(Q^{n_i})(\xi) - Q^{n_i}(\xi)) \tag{120}$$

$$\leq 2\epsilon + \frac{8\mathsf{sp}(Q^0 - Q^*)}{n_i + 2} + 4\epsilon \tag{121}$$

$$\leq 14\epsilon, \tag{122}$$

thus $G_i \subseteq S_i$, which completes the proof. $\square$

**Proposition H.2.** *It holds that*

$$\mathbb{E}[N] \le 2(1 + \mu/\epsilon)/(1 - \delta).$$

*Namely, $N$ is finite almost surely and PF-RHI($Q^0, \epsilon, \delta, i = 0$) stops with probability 1 after a finite number of iterations.*

*Proof.* In each iteration $i$, in PF-RHI($Q^0, \epsilon, \delta, i = 0$), we reinitialize $Q^0 = 0$ prior to the inner for loop where $k = 0, 1, \ldots, n_i$. This implies that the events $\{S_i : i \in \mathbb{N}\}$ are mutually independent. Thus,

$$\mathbb{P}(I = i) = \mathbb{P}\Big(\bigcap_{j=0}^{i-1} S_j^c \cap S_i\Big) = \prod_{j=0}^{i-1} \mathbb{P}(S_j^c) \cdot \mathbb{P}(S_i).$$

Now from Lemma H.1, it holds that $\mathbb{P}(S_i^c) \le \mathbb{P}(G_i^c) \le \delta_i$ for all $i \ge i_0$, which implies that $\mathbb{P}(I = i) \le \prod_{j=i_0}^{i-1} \delta_j$.

Moreover, by definition of $c$, we have that $2 \sum_{i=0}^{\infty} c_i^{-1} \le 1$, thus $\delta_j = \delta/c_j \le \delta/2$, implying that $\mathbb{P}(I = i) \le (\delta/2)^{i-i_0}$. Using this and the fact that $n_i = n_{i_0} 2^{i-i_0}$, it holds that

$$\mathbb{E}[N] = \sum_{i=0}^{\infty} n_i \mathbb{P}(N = n_i)$$

$$\le n_{i_0} + \sum_{i=i_0+1}^{\infty} n_{i_0} 2^{i-i_0} \mathbb{P}(I = i)$$

$$\le n_{i_0}\Big(1 + \sum_{i=i_0+1}^{\infty} \delta^{i-i_0}\Big).$$

The proof is then completed by the bound of $n_{i_0} \le 2(1 + \mu/\epsilon)$, which implies that $\mathbb{E}[N] \le 2(1 + \mu/\epsilon)/(1 - \delta)$. $\qquad\square$

**Theorem H.3.** *Let $c_k = 5(k + 2) \ln^2(k + 2)$ and $\beta_k = k/(k + 2)$ hold. Let $n_i = N$ so that $(Q^N, T^N, \pi^N)$ is returned by PF-RHI($Q^0, \epsilon, \delta, i = 0$). Then with probability of at least $(1 - \delta)$, we have for all $s \in \mathcal{S}$,*

$$g_{\mathcal{P}}^* - g_{\mathcal{P}}^{\pi^N}(s) \le \mathsf{sp}(\mathcal{T}_{\mathcal{P}}(Q^N) - Q^N) \le 16\epsilon,$$

*Which obtains a sample and time complexity of $\tilde{\mathcal{O}}\big(\hat{L}|\mathcal{S}||\mathcal{A}|(\nu^2/\epsilon^2 + 1)\big)$, with $\hat{L} = \ln\big(4|\mathcal{S}||\mathcal{A}|(1 + \mu/\epsilon)/\delta\big) \log^4(2(1 + \mu/\epsilon))$.*

*Proof.* As Lemma E.2 implies that $0 \le g_{\mathcal{P}}^* - g_{\mathcal{P}}^{\pi^N} \le \mathsf{sp}(\mathcal{T}_{\mathcal{P}}(Q^N) - Q^N)$.

We first define $A = \{I \le i_0\}$ and $B = \bigcap_{i=0}^{\infty} G_i$, where $G_i = \{\|T^k - \mathcal{T}_{\mathcal{P}}(Q^k)\|_\infty \le \epsilon, \forall k = 0, 1, \ldots, n_i, \forall(s, a) \in \mathcal{S} \times \mathcal{A}$. We claim that, $\mathbb{P}(A \cap B) \ge (1 - \delta)$. To prove this, since $\mathbb{P}(G_i^c) \le \delta_i$. Thus

$$\mathbb{P}(B^c) = \mathbb{P}\Big(\bigcup_{i=1}^{\infty} G_i^c\Big) \le \sum_{i=1}^{\infty} \delta/c_i \le \delta/2. \tag{123}$$

Then Lemma H.1 implies that $\mathbb{P}(A) \ge \mathbb{P}(S_{i_0}) \ge \mathbb{P}(G_{i_0}) \ge 1 - \delta_{i_0} \ge 1 - \delta/2$, thus combining this with (123) implies $\mathbb{P}(A^c \cup B^c) \le \delta$, which proves our claim.

We then use the definitions of $N$, $I$, and $G_i$, which imply that

$$\mathsf{sp}(\mathcal{T}_{\mathcal{P}}(Q^N) - Q^N) \le \mathsf{sp}(\mathcal{T}_{\mathcal{P}}(Q^N) - T^N) + \mathsf{sp}(T^N - Q^N)$$

$$\le 2\epsilon + 14\epsilon$$

$$= 16\epsilon.$$

It further implies the total sample complexity, $M \triangleq \sum_{i=0}^{I} M_i$ can be bounded as

$$M \le \sum_{i=0}^{i_0} M_i \tag{124}$$

$$= |\mathcal{S}||\mathcal{A}| \sum_{i=0}^{i_0} \tilde{\mathcal{O}}\Big(\alpha_i \mathsf{sp}(Q^0)^2/\epsilon^2 + \alpha_i \ln^3(n_i + 2)\mathsf{sp}(Q^0 - Q^*)^2/\epsilon^2 + \alpha_i n_i^2 \ln^2(n_i + 2)\Big),$$

where $\alpha_i = \ln(2|\mathcal{S}||\mathcal{A}|(n_i + 1)/\delta_i)$ is the parameter defined at iteration $i$ (prior to the inner for loop) of PF-RHI. Moreover, since $n_{i_0}^2 \le 4(1 + \mu/\epsilon)^2 = \tilde{\mathcal{O}}\big(\mathsf{sp}(Q^0 - Q^*)^2/\epsilon^2 + 1\big)$, we have that

$$M \le |\mathcal{S}||\mathcal{A}|(i_0 + 1)\tilde{\mathcal{O}}\big(\alpha_{i_0}\mathsf{sp}(Q^0)^2/\epsilon^2 + \alpha_{i_0}\log^3(n_{i_0} + 2)\mathsf{sp}(Q^0 - Q^*)^2/\epsilon^2 + \alpha_{i_0}n_{i_0}^2\log^2(n_{i_0} + 2)\big)$$

$$\le |\mathcal{S}||\mathcal{A}|\alpha_{i_0}\log^4(n_{i_0} + 2)\,\tilde{\mathcal{O}}\big(\mathsf{sp}(Q^0)^2/\epsilon^2 + 2\mathsf{sp}(Q^0 - Q^*)^2/\epsilon^2 + 1\big)$$

$$\le \hat{L}|\mathcal{S}||\mathcal{A}|\tilde{\mathcal{O}}(\nu^2/\epsilon^2 + 1),$$

which completes the proof. $\qquad\square$

**Corollary H.4.** *Let $n_i = N$. Then with probability of at least $(1 - \delta)$, for all $s \in \mathcal{S}$, it holds that*

$$g_{\mathcal{P}}^* - g_{\mathcal{P}}^{\pi^N}(s) \le \mathsf{sp}(\mathcal{T}_{\mathcal{P}}(Q^N) - Q^N) \le \epsilon.$$

*This results in a sample and time complexity of $\tilde{\mathcal{O}}(\tilde{L}|\mathcal{S}||\mathcal{A}|\mathcal{H}^2/\epsilon^2) = \tilde{\mathcal{O}}(SA\mathcal{H}^2/\epsilon^2)$, where we define $\tilde{L} = \ln(2|\mathcal{S}||\mathcal{A}|\mathcal{H}/(\epsilon\delta))\ln^4(\mathcal{H}/\epsilon)$ and $\tilde{\mathcal{O}}(\cdot)$ hides logarithmic terms.*

*Proof.* Note that

$$\mathsf{sp}(Q^0 - Q^*) = \mathsf{sp}(Q^*) = \mathsf{sp}(r + \mathsf{P}h^*) \le \mathsf{sp}(r) + \mathsf{sp}(h^*),$$

which is due to $Q^0 = 0$ at each iteration $i$ and the nonexpansivity of the map $Q \mapsto \max_{\mathcal{A}}(Q) = h$. Moreover, since $2(1 + \mu/\epsilon) = \tilde{\mathcal{O}}\big(\mathsf{sp}(h)^2/\epsilon\big)$, combining with Theorem H.3, the result follows by verifying the definition of $\tilde{L}$. $\qquad\square$

We then derive the results under expectations.

**Lemma H.5.** *For an arbitrary fixed iteration $i \in \mathbb{N}$ of PF-RHI($Q^0, \epsilon, \delta, i = 0$), let $M_i = |\mathcal{S}||\mathcal{A}| \sum_{j=0}^{n_i} m_j$ be the number of samples obtained during iteration $i$. We have*

$$M_i \le |\mathcal{S}||\mathcal{A}|\tilde{\mathcal{O}}\big(n_i + (\zeta/\epsilon)^2\alpha_i n_i^2\log^2(n_i + 2)\big),$$

*where $\alpha_i = \ln(2|\mathcal{S}||\mathcal{A}|(n_i + 1)/\delta_i)$.*

*Proof.* By using induction, for $k = 0$ we have by initialization $d^0 = \max_{\mathcal{A}}(Q^0)$ and $T^{-1} = r$. For $k \ge 0$, it holds that

$$\mathsf{sp}(d^k) \le \mathsf{sp}(Q^k - Q^{k-1})$$

$$\le \frac{2}{(k+1)(k+2)}\mathsf{sp}(T^{k-1} - Q^0) + \frac{k-1}{k+1}\mathsf{sp}(d^{k-1})$$

$$\le \frac{2}{(k+1)(k+2)}\big((k+1)\zeta + \zeta\big) + \frac{k-1}{k+1}\zeta$$

$$= \zeta.$$

This implies that

$$\mathsf{sp}(T^k) \le \mathsf{sp}(T^{k-1}) + \mathsf{sp}(D^k)$$

$$\le (k+1)\zeta + \mathsf{sp}(d^k)$$

$$\le (k+2)\zeta.$$

Thus for a fixed $i \in \mathbb{N}$ in PF-RHI, we can bound $M_i$ as

$$M_i \le |\mathcal{S}||\mathcal{A}|\Big((n_i + 1) + (\alpha_i/\epsilon^2)\sum_{j=0}^{n_i} c_j\mathsf{sp}(d^j)^2\Big)$$

$$\le |\mathcal{S}||\mathcal{A}|\Big((n_i + 1) + 5(\zeta/\epsilon)^2\alpha_i\sum_{j=0}^{n_i}(j+2)\ln^2(j+2)\Big)$$

$$= |\mathcal{S}||\mathcal{A}|\tilde{\mathcal{O}}\big(n_i + (\zeta/\epsilon)^2\alpha_i n_i^2\log^2(n_i + 2)\big),$$

which completes the proof. $\qquad\square$

**Theorem H.6.** *Assume that the robust-AMDP satisfies Assumption 3.1, and that the sequences $c_k = 5(k+2)\ln^2(k+2)$ and $\beta_k = k/(k+2)$ hold. Let $n_i = N$ so that $(Q^N, T^N, \pi^N)$ is the output of PF-RHI$(Q^0, \epsilon, \delta, i = 0)$. Then for every $s \in \mathcal{S}$ we have,*

$$\mathbb{E}\big[g_{\mathcal{P}}^* - g_{\mathcal{P}}^{\pi^N}(s)\big] \leq 16\epsilon + \delta\mathsf{sp}(r),$$

*which yields an expected sample and time complexity of*

$$\tilde{\mathcal{O}}\big(|\mathcal{S}||\mathcal{A}|(\nu^2/\epsilon^2 + 1 + \delta(1 + \mu/\epsilon)^2(1 + (\zeta/\epsilon)^2))\big).$$

*Proof.* We start our proof similar to Theorem H.3 by considering the events $A = \{I \leq i_0\}$ and $B = \bigcap_{i=1}^{\infty} G_i$. From Theorem H.3, under $A \cap B$, for every $s \in \mathcal{S}$ it holds that $g_{\mathcal{P}}^* - g_{\mathcal{P}}^{\pi^n}(s) \leq 16\epsilon$ with probability $\mathbb{P}(A \cap B) \geq 1 - \delta$.

On the other hand, under $(A \cap B)^c$, we have the trivial bound of $g_{\mathcal{P}}^* - g_{\mathcal{P}}^{\pi^n}(s) \leq \mathsf{sp}(r)$, $\forall s \in \mathcal{S}$.

Hence the two cases together imply that

$$\mathbb{E}[g_{\mathcal{P}}^* - g_{\mathcal{P}}^{\pi^n}(s)] \leq 16\epsilon + \delta\mathsf{sp}(r), \quad \forall s \in \mathcal{S}.$$

Similar to Theorem H.3, we wish to estimate the sample complexity like $M = \sum_{i=0}^{I} M_i$ for each iteration $i$ of PF-RHI. We accomplish this by considering the infinite disjoint union of all indexes $i > i_0$, or more formally $A^c = \bigsqcup_{i=i_0+1}^{\infty}\{I = i\}$ which yields

$$\mathbb{E}[M] = \underbrace{\mathbb{E}[M|A \cap B]\mathbb{P}(A \cap B)}_{\text{Term 1}} + \underbrace{\mathbb{E}[M|A \cap B^c]\mathbb{P}(A \cap B^c)}_{\text{Term 2}} + \underbrace{\sum_{i=i_0+1}^{\infty} \mathbb{E}[M|I = i]\mathbb{P}(I = i)}_{\text{Term 3}}.$$

**Term 1:**
We use the result derived from the proof of Theorem H.3 on the event $(A \cap B)$ and the fact that $\mathbb{P}(A \cap B) \leq 1$. By defining $\hat{L} = \ln\big(4|\mathcal{S}||\mathcal{A}|(1 + \mu/\epsilon)/\delta\big)$, we have that

$$\mathbb{E}[M|A \cap B]\mathbb{P}(A \cap B) = \tilde{\mathcal{O}}\big(\hat{L}|\mathcal{S}||\mathcal{A}|(\nu^2/\epsilon^2 + 1)\big). \tag{125}$$

**Term 2:**
We can combine the result in Lemma H.5 with $\mathbb{P}(A \cap B^c) \leq \mathbb{P}(B^c) \leq \delta$, and $n_{i_0} \leq 2(1 + \mu/\epsilon)$ to obtain the following result:

$$
\begin{aligned}
\mathbb{E}[M|A \cap B^c]\mathbb{P}(A \cap B^c) &\leq \delta|\mathcal{S}||\mathcal{A}|\sum_{i=0}^{i_0} \tilde{\mathcal{O}}\big(n_i + (\zeta/\epsilon)^2\alpha_i n_i^2 \log^2(n_i + 2)\big) \\
&\leq \delta|\mathcal{S}||\mathcal{A}|\tilde{\mathcal{O}}\big(n_{i_0} + (\zeta/\epsilon)^2\alpha_{i_0} n_{i_0}^2 \log^3(n_{i_0} + 2)\big) \\
&\leq \delta|\mathcal{S}||\mathcal{A}|\tilde{\mathcal{O}}\big(n_{i_0} + \hat{L}(\zeta/\epsilon)^2 n_{i_0}^2\big).
\end{aligned}
\tag{126}
$$

The final inequality holds by using the definition of $\hat{L}$ and that $\alpha_{i_0}\log^3(n_{i_0} + 2) \leq \tilde{\mathcal{O}}(\hat{L})$.

**Term 3:**
To bound this term, we can again employ the result of Lemma H.5 along with defining $Z \triangleq \sum_{i=i_0+1}^{\infty} \mathbb{E}[M|I = i]\mathbb{P}(I = i)$ to have that

$$
\begin{aligned}
Z &\leq |\mathcal{S}||\mathcal{A}|\sum_{i=i_0+1}^{\infty} \tilde{\mathcal{O}}\big(n_i + (\zeta/\epsilon)^2\alpha_i n_i^2 \log^2(n_i + 2)\big)\mathbb{P}(I = i) \\
&\leq |\mathcal{S}||\mathcal{A}|\sum_{i=i_0+1}^{\infty} \tilde{\mathcal{O}}\big(n_i + \hat{L}(\zeta/\epsilon)^2 i^3 n_i^2\big)\mathbb{P}(I = i),
\end{aligned}
$$

where the final inequality follows from using the re-initializations of $n_i$, $\delta_i$, and $\alpha_i$ in PF-RHI to obtain $\alpha_i = \tilde{\mathcal{O}}(\hat{L} + i) \leq \hat{L}\tilde{\mathcal{O}}(i)$, where $\log\big((n_i + 1)c_i\big) = \tilde{\mathcal{O}}(i)$, and likewise $\log^2(n_i + 2) = \tilde{\mathcal{O}}(i^2)$. With this in place, recall that $n_i = n_{i_0}2^{i-i_0}$.

From Proposition H.2, for $i \geq i_0 + 1$ we have that $\mathbb{P}(I = i) \leq \prod_{j=i_0}^{i-1} \delta_j \leq \tilde{\mathcal{O}}\left(\delta \prod_{j=i_0}^{i-1} \frac{1}{j+2}\right)$. Therefore, we can denote the following

$$S_1 \triangleq \sum_{i=i_0+1}^{\infty} 2^{i-i_0} \prod_{j=i_0}^{i-1} \frac{1}{j+2}, \tag{127}$$

$$S_2 \triangleq \sum_{i=i_0+1}^{\infty} 2^{2(i-i_0)} i^3 \prod_{j=i_0}^{i-1} \frac{1}{j+2}, \tag{128}$$

which allows us to show that

$$Z \leq \delta |\mathcal{S}||\mathcal{A}| \tilde{\mathcal{O}}\left(S_1 n_{i_0} + S_2 \hat{L}(\zeta/\epsilon)^2 n_{i_0}^2\right).$$

However, we can calculate (127) and (128) using their incomplete Gamma functions like,

$$S_1 = e^2 2^{-(i_0+1)} [\Gamma(i_0 + 2) - \Gamma(i_0 + 2, 2)]$$
$$\leq \frac{(e^2 - 3)}{2} \tag{129}$$
$$S_2 = 84 + 4i_0(i_0 + 5) + 67e^4 2^{-2(i_0+1)} [\Gamma(i_0 + 2) - \Gamma(i_0 + 2, 4)]$$
$$= \tilde{\mathcal{O}}\left((i_0 + 1)^2\right). \tag{130}$$

With (129) and (130), we can finally bound $Z$ as

$$Z \leq \delta |\mathcal{S}||\mathcal{A}| \tilde{\mathcal{O}}\left(n_{i_0} + \hat{L}(\zeta/\epsilon)^2 n_{i_0}^2 (i_0 + 1)^2\right). \tag{131}$$

We can then find the total expected value of the sample complexity by combining (125), (126), and (131) by rearranging similar order terms and disregarding the logarithmic terms to obtain:

$$\mathbb{E}[M] \leq |\mathcal{S}||\mathcal{A}| \tilde{\mathcal{O}}\left(\hat{L}(\nu^2/\epsilon^2 + 1) + \delta n_{i_0} + \delta \hat{L}(\zeta/\epsilon)^2 n_{i_0}^2 (i_0 + 1)^2\right)$$
$$= |\mathcal{S}||\mathcal{A}| \tilde{\mathcal{O}}\left((\nu^2/\epsilon^2 + 1) + \delta(1 + \mu/\epsilon)^2 (1 + (\zeta/\epsilon)^2)\right),$$

which completes the proof. $\square$

**Corollary H.7.** *Assume that the robust-AMDP satisfies Assumption 3.1, that the sequences $c_k = 5(k+2)\ln^2(k+2)$ and $\beta_k = k/(k+2)$ hold, $r(s,a) \in [0,1]$ $\forall(s,a) \in \mathcal{S} \times \mathcal{A}$, and $\mathcal{H} \geq 1$. Let $n_i = N$ such that $N \geq \mathcal{H}/\epsilon$ so that $(Q^N, T^N, \pi^N)$ is returned by PF-RHI$(Q^0, \epsilon/17, \delta, i=0)$ with $Q^0 = 0$, $\epsilon \leq 1$, and $\delta = \epsilon^2/17$. We have for every $s \in \mathcal{S}$,*

$$\mathbb{E}[g_{\mathcal{P}}^* - g_{\mathcal{P}}^{\pi^N}(s)] \leq \epsilon,$$

*where we obtain an expected sample complexity of $\tilde{\mathcal{O}}\left(|\mathcal{S}||\mathcal{A}|\mathcal{H}^2 C_{\mathrm{rs}}^2/\epsilon^2\right)$.*

*Proof.* The proof is directly derived by applying the value of $\delta$ in Theorem H.6. $\square$

