# OpenReview forum: "Model-Free Robust Average-Reward Reinforcement Learning with Sample Complexity Analysis"
_ICML.cc/2026/Conference — ICML 2026 regular_

### Official Review · Reviewer_mjJf · 2026-03-09

**Soundness:** 3
**Presentation:** 3
**Significance:** 3
**Originality:** 3
**Overall Recommendation:** 4
**Confidence:** 3

**Summary:**

This paper studies model-free distributionally robust reinforcement learning under the average-reward criterion. The authors propose a new algorithm, Robust Halpern Iteration (RHI), which reformulates the robust average-reward Bellman equation in a quotient space and applies Halpern iteration to compute the solution without relying on a contraction coefficient. To estimate the robust Bellman operator from nominal samples, the paper introduces a black-box oracle abstraction and instantiates it using a higher-order multi-level Monte Carlo (MLMC) estimator that reduces bias compared with standard plug-in estimators. For KL and $\chi^2$ uncertainty sets, the authors derive finite-sample guarantees with complexity roughly $ \widetilde O\!\left(\frac{SAH^2}{\varepsilon^{2+o(1)}}\right) $,
which asymptotically matches the best known model-based results for robust average-reward RL while operating in the model-free generative-model setting.

**Compliance With Llm Reviewing Policy:**

Affirmed.

**Final Justification:**

The authors' rebuttal clarified the novelty of the work relative to recent literature. Specifically, the shift from asymptotic analysis to finite-sample complexity and the design of a higher-order MLMC estimator to achieve near-optimal rates are non-trivial advancements. Furthermore, the inclusion of a parameter-free variant in the appendix effectively addresses practical concerns regarding the unknown robust optimal span. Given the authors' commitment to improving the presentation of core assumptions and discussing the method's limitations in the final manuscript, I maintain my positive recommendation.

**Key Questions For Authors:**

1. **Relationship to closely related work.**
   The paper builds on recent work on model-free robust average-reward RL and MLMC-based robust RL. It would be helpful if the authors could more explicitly summarize the main technical differences between the current algorithm and these prior methods.

2. **Role of the span parameter $H$.**
   Algorithm 1 requires a horizon parameter satisfying
   $$
   n \ge C\frac{H}{\varepsilon}.
   $$
   Could the authors clarify how sensitive the algorithm is to this parameter and whether the parameter-free variant discussed later fully removes this requirement in practice?

3. **Generality of the MLMC estimator.**
   The higher-order MLMC estimator relies on smoothness assumptions for the function $\Psi$. Could the authors elaborate on which uncertainty sets satisfy these assumptions and whether additional cases could be handled with similar techniques?

4. **Extension to other uncertainty sets.**
   The main results focus on KL and $\chi^2$ uncertainty sets. Are there additional uncertainty models where the same framework can be applied with similar guarantees?

**Limitations:**

The paper briefly discusses limitations but the discussion could be expanded slightly. In particular, it would be useful to comment on the gap between the generative-model assumption used for the analysis and more realistic data-collection settings.

**Strengths And Weaknesses:**

### **Strengths**

1. Addresses an important theoretical question.
   The paper studies *robust average-reward reinforcement learning* in the model-free setting, which is a meaningful and technically challenging problem. While prior work has studied either asymptotic convergence in the model-free robust average-reward setting or finite-sample guarantees in other robust RL formulations, combining both aspects is nontrivial and well motivated.

2. Interesting algorithmic formulation.
   The use of a quotient-space reformulation together with Halpern iteration is conceptually appealing. In particular, the reformulation avoids directly estimating the unknown gain $g^\star$ and instead solves the equation
   $$
   T_P(Q) - Q = ce,
   $$
   which isolates the policy-relevant component of the solution and helps avoid dependence on an unknown contraction parameter.

3. Careful estimator design.
   The proposed **$K$-order MLMC estimator** is a technically interesting component of the paper. By canceling leading bias terms in the estimator expansion
   $$
   \mathbb{E}[\widehat G_N(h)] =
   G(h) + \sum_{j=1}^{K}\frac{c_j(h)}{N^j} + O(N^{-(K+1)}),
   $$
   the approach reduces bias relative to standard plug-in estimators and improves the final complexity dependence.

4. Strong theoretical guarantees.
   For KL and $\chi^2$ uncertainty sets, the resulting sample complexity is shown to be approximately
   $$
   \widetilde O\!\left(\frac{SAH^2}{\varepsilon^{2+o(1)}}\right),
   $$
   which matches the best known model-based rates up to logarithmic factors. Achieving this rate in the *model-free robust average-reward setting* is a meaningful theoretical result if the analysis holds.

---

### **Weaknesses**

1. Incremental novelty relative to recent work.
   The paper is closely related to two recent lines of work:
   - *Model-Free Robust Average-Reward Reinforcement Learning* (ICML 2023), which studies the same robust average-reward setting but focuses on asymptotic convergence, and
   - *Model-Free Robust Reinforcement Learning with Sample Complexity Analysis* (UAI 2024), which develops MLMC-based techniques for finite-sample robust RL.

   The current paper can be viewed as combining ideas from these directions—extending finite-sample MLMC techniques to the robust average-reward setting and introducing additional algorithmic tools (e.g., Halpern iteration). While this is a meaningful technical contribution, the distinction from these closely related works could be explained more explicitly.

2. Some assumptions are introduced somewhat implicitly.
   In particular, the smoothness conditions on $\Psi$ and the scope of uncertainty sets covered by the expansion could be stated more explicitly up front.

---

### **Overall assessment in one sentence**

Overall, the paper presents a technically interesting and well-motivated theoretical contribution that extends recent advances in model-free robust RL to the average-reward setting with finite-sample guarantees, though the novelty relative to very recent related work could be clarified more explicitly.

---

> ### Author Rebuttal · Authors · 2026-03-26
>
> We sincerely thank the reviewer for the overall positive assessment and detailed review. We address each point below.
>
> **W1 \& Q1: Compared to [Wang et al., 2023d] and [Wang et al., 2024f].**
> We appreciate this observation and would like to clarify the substantial technical differences:
>
>  vs [Wang et al., 2023d]: This work proposes a model-free algorithm for robust average-reward RL but only provides an asymptotic convergence guarantee, and the convergence analysis is based on ODE and stochastic approximation techniques. In contrast, our paper provides the finite-sample complexity. Bridging the gap from asymptotic to finite-sample analysis requires fundamentally new techniques, especially in the finite-sample estimation oracle: in  [Wang et al., 2023d], standard MLMC estimator is employed. However, the sample complexity of MLMC can be infinite per step (as we mentioned in Sec 5.1.1), and we need to design a novel K-order MLMC to balance the bias and the sample complexity. Moreover, the sample complexity analysis of RHI (quantifying concentration bound and error characterizations) are also new.
>
> vs. [Wang et al., 2024f]: This work studies robust discounted RL with MLMC techniques. Our contributions beyond this work include: (1). The entire average-reward framework: handling the double unknowns $(Q, g^*)$, the non-contractive operator, and the quotient-space formulation --- none of which appear in the discounted setting (and our approach to solve them is also different from [Wang et al., 2023d]).
>  (2). The K-order MLMC estimator: [Wang et al., 2024f] uses truncated MLMC with $\mathcal{O}(1/N)$ bias, whereas our K-order construction achieves $\mathcal{O}(f(K)/N_0^{K+1})$ bias, which is the key to achieving the near-optimal $\varepsilon^{-(2+o(1))}$ rate instead of $\varepsilon^{-3}$ (also see the comparison in Section 5.1.1).
>
> We will make these distinctions more explicit in the revised introduction and related work sections.
>
>
> **W2: Some assumptions are introduced somewhat implicitly.**
>
> Thank you for this valid point. We will improve the presentation by:
>
>  (1). Stating the smoothness Assumption E.2 on $\Psi$ explicitly in the main text (Section 5.1), rather than deferring it entirely to the appendix. We highlight that these conditions are satisfied by our concrete constructions, as we discussed in the paper.
>
> (2). Adding a brief remark after Eq. (12) discussing which uncertainty sets satisfy the moment representation (12) and the required smoothness. Currently this is partially covered in Remark 5.1, but we will expand it to include the smoothness requirements.
>
>
> **Q2: Role of the span parameter.**
>
> Sensitivity: The convergence and $\varepsilon$-optimality results will still hold if we replace $\mathcal{H}$ by any upper bound on it. Such dependence on $\mathcal{H}$ is inevitable as $\mathcal{H}$ can be viewed as the effective horizon of the robust average reward, and it represents the hardness of the problem.
>
> On the other hand, our parameter-free variant (Algorithm 2 in Appendix G) removes such an **explicit** dependence on $\mathcal{H}$; instead, we use a doubling trick, which requires \emph{no knowledge of $H$}, and automatically terminates when sufficient iterations have been performed. Thus our PF-RHI is fully implementable without any prior knowledge in practice. However, as we mentioned above, although PF-RHI does not explicitly depends on $\mathcal{H}$, the resulting sample complexity still inherently depends on $\mathcal{H}$.
>
> **Q3 \& Q4: Generality of the MLMC estimator and extension to other sets.**
> Our K-order MLMC framework applies to any uncertainty set whose robust support function can be expressed (or approximated) as eq (12) with smooth duality objective function (Assumption E.2). In our paper, we studied KL divergence and $\chi^2$ divergence, which satisfy these requirements. Beyond them, for uncertainty set defined through $f$-divergence, there exists a duality form of the corresponding support function (Ghosh et al., 2025), and the duality function is smooth for a broad range of $f$-divergence, like R\'enyi divergence, Cressie-Read family, etc. For these models, our technique is extendable.
>
> On the other hand, our technique may not be directly applicable for Wasserstein uncertainty sets, as its support function involves another optimization over couplings, rather than reweighting as in $f$-divergence set. However, we could still use the standard MLMC technique (Xu et al., 2025 a;b) to estimate the support increments, and derive (a higher) sample complexity with our RHI. It will be our future interest to reduce the bias and achieve tighter complexity.
>
>
> **Limitation.** We appreciate the suggestion. The generative model is the most fundamental setting for RL theory, allowing us to develop the information-theoretic complexity. The extension to online setting is also feasible but requires additional efforts (please also kindly see our response to W2, Q3 of Reviewer AJbB).

---

> > ### Author Rebuttal · Reviewer_mjJf · 2026-04-04
> >
> > Thanks for the authors' response. My concerns have been addressed. I will keep my score.

---

> > > ### Author Response · Authors · 2026-04-04
> > >
> > > Dear reviewer, thank you for taking the time to read our rebuttal and for your continued positive assessment of our work. We deeply appreciate your constructive feedback and your engagement throughout the review process.

---

### Official Review · Reviewer_AJbB · 2026-03-12

**Soundness:** 3
**Presentation:** 2
**Significance:** 2
**Originality:** 2
**Overall Recommendation:** 4
**Confidence:** 2

**Summary:**

This paper studies robust average-reward RL in the generative-model, model-free setting. A core technical issue is that theBellman operator in this case is generally non-contractive. The paper proposes Robust Halpern Iteration (RHI) as a direct approach to this difficulty. Another problem is the optimization of solving the robust Bellman equation while not knowing the gain. To deal with this issue, a key underlying assumption is assumption 3.1, which assumes that for any policy and transition in the set of transitions the induced MDP is irreducible.

Another challenge is the non-linearity of the bellman operator. To deal with this issue,  the authors assume a black-box oracle that can estimate the worst-case from nominal data.

The authors then derive a sample complexity bound to reach $(\epsilon,\delta)$-PAC guarantees. Lastly, the paper instantiates that oracle  KL and chi2 uncertainty sets.

**Compliance With Llm Reviewing Policy:**

Affirmed.

**Final Justification:**

After carefully re-reading the paper, the rebuttal, and the other reviewers' assessments, I am willing to increase my score to weak accept. The paper makes a clear theoretical advance, and derives a model-free algorithm for robust average-reward RL with finite-sample guarantees, and the K-order MLMC construction   is a genuine technical contribution. The Halpern iteration in quotient space elegantly sidesteps the non-contractive operator and unknown gain simultaneously.

However, my concerns are only partially resolved. Assumption 3.1 remains strong and the paper still lacks concrete sufficient conditions or worked examples where it holds naturally (the rebuttal's pointer to "small radius" is informal and not developed). The $o(1)$ exponent, while vanishing asymptotically, yields complexity closer to $\epsilon^{-3}$ for practical regimes (moderate $\epsilon$) . Ultimately I believe the contribution is solid and fits well into the literature.

**Key Questions For Authors:**

- How essential is Assumption 3.1 in practice? the guarantee must hold uniformly over kernels in the uncertainty set and over policies, this can be a strong requirement. Is there a clean example class where the assumption is natural?

- Can you provide at least minimal empirical evidence?

- is it possible to move from a generative setting to an online one?

- What is the computational cost per oracle call?

**Limitations:**

No, the paper should more explicitly discuss the strengths of the assumptions and the lack of empirical results.

**Strengths And Weaknesses:**

-	The  first part of the paper is generally well structured. The motivation is clear, and the high-level narrative is easy to follow. However, the presentation becomes more dense in the later sections. In particular, the discussion of the black-box estimator and the oracle construction is difficult to follow, and some key technical material is deferred to the appendix in a way that weakens the readability of the main paper.
-	The problem is meaningful, the authors study average-reward robust RL and provide PAC guarantees.
-	The authors provide original ideas in solving some problems, like Halpern iteration to deal with the non-contractive issue. Also the K-order MLMC seems to be a novel contribution. However, I am not an expert in this specific area of RL,  and I cannot assess the novelty w.r.t. prior work.
-	Results rely on a strong generative-model assumption, which limits practical relevance. Furthermore, assumption 3.1 is quite strong in practice (or every policy and every admissible kernel the induced chain is irreducible)
-	While the authors claim the method is model-free, it relies on a strong simulator/oracle access.

---

> ### Author Rebuttal · Authors · 2026-03-26
>
> We thank the reviewer for the detailed review. We provide a response as below.
>
> **W1. Presentation.**
> We briefly explain it here and will improve our presentation.
>
> For the oracle construction, the challenge is that the robust operator is nonlinear in the nominal distribution. The key insight is that for divergence uncertainty sets, its robust operator can be solved via duality as eq (12), and it suffices to estimate the duality objective $\Psi(\theta(h))$.
>
> But a naive plug-in estimator $\hat{G}\_N (h) = \Psi(\hat{\theta\_N}(h))$ using $N$ samples has a bias $\mathcal{O}(1/N)$, which leads to large complexity. We thus designed our K-order MLMC to reduce this bias: for each layer $l\in [K]$, we use $N_l$ samples to form the plug-in estimators $\hat{G}\_{N_l}$, and take a weighted combination $\sum_{l=0}^K w_l^{(K)} \hat{G}_{N_l}(h)$. These weights $w_l^{(K)}$ are specifically chosen to cancel the highest order bias term from each layer, and the resulting residual bias is $\tilde{O}(K! / N_0^{K+1})$ (Prop E.7). Balancing $K \approx \sqrt{\log(1/\varepsilon)}$ yields a lower bias and our finite complexity.
>
>
> **W2. Generative model/model-free.** (please also see Q3) We first highlight that the generative model is the most fundamental setting in theoretical studies, as it bypasses the exploration challenge in RL and provides a clean setting for information-theoretic studies of statistical efficiency. Our results hence establish the most fundamental information-theoretic complexity of model-free robust average-reward RL. Generative model is also a reasonable abstraction for many simulation-based applications (robotics simulators, game engines), where one can query any state-action pair.
>
> On the other hand, in RL literature, `model-based' means the algorithm explicitly constructs/estimates the full transition model and then solves a planning problem (it is irrelevant to generative models). Our algorithm is model-free as it directly updates Q-values via R-SAMPLE.
>
> **Q1. Assumption 3.1.** The irreducible assumption can be satisfied by many examples. For instance, if the nominal kernel is irreducible, and the radius is relatively small, then the assumption holds (Xu et al., 2025a;b).
>
> We clarify that our K-order MLMC does not require Ass 3.1, and only the RHI part does. Average reward depends on the underlying chain structures, and assumptions are inevitably. Ass 3.1 is the most standard and widely-studied setting (whereas studies under more general settings are extremely limited). Moreover, as in Remark 3.3, our results directly extend to more general settings with bounded bias spans, including unichain (Wang et al., 2023c,d) and potentially weakly communicating ones.
>
> **Q2: Empirical evidence.** We apologize for misunderstanding, and refer the reviewer to our empirical results in Appendix H.
> Fig 1 shows that our K-order MLMC estimator's bias decreases significantly faster than baseline. Fig 2 demonstrates the convergence of RHI. These experiments confirm our theoretical results. We will also develop larger experiments.
>
> **Q3: Online setting.** We believe online extension is feasible, and key components of our framework could be reused. A natural extension is to embed our methods within an epoch-based online learning framework such as UCRL2 (Jaksch et al., 2010). The agent will build (visiting-number dependent) confidence sets $\Phi_k(s,a)$ around empirical transitions using concentration ineq, such that the true nominal kernel $P^a_s \in \Phi_k(s,a)$, and do an optimistic robust planning to execute $\arg\max_\pi \max_{p\in \Phi}\min_{q\in \mathcal{P}(p)} g^\pi_q$ to collect new data, where $\mathcal{P}(p)$ is the uncertainty set centered at $p$. Since $\max_{p\in \Phi}\min_{q\in \mathcal{P}(p)} g^\pi_q \geq g^\pi_{\mathcal{P}}$ and $\Phi$ shrinks with more samples, it will balance exploration and exploitation and reduce online regret. In this step, we can reuse our RHI as the solver, and our K-order MLMC to construct tighter estimation.
>
> However, online setting introduces additional difficulties beyond our current scope: (1) samples from trajectories are correlated, requiring mixing-time-dependent concentration arguments; (2) estimating the worst-case performance from trajectory-wise nominal data can be exponentially inefficient, as the worst-case states may not be reachable under nominal kernel (Lu et. al., 2024; Ghosh et al., 2025).
>
> We thus view this extension as an important yet challenging direction.
>
> **Q4: Computational cost.**
> In each call, for each $(s,a)$-pair,  it solves a duality optimization for $K+1$ times, and solves a $K$-variable linear equation system in Line 328 (with $O(K^3)$ complexity). The duality optimization can be solved polynomially, e.g., the computational complexity under $\chi^2$ is $O(S\log(S))$ (Iyengar, 2005). Thus the complexity per call is $\tilde{O}(SA(K^3+K\text{poly}(S)))$, and since we set $K \approx \sqrt{\log(1/\varepsilon)}$, it is polynomial in $S, A, \epsilon$.

---

> > ### Author Rebuttal · Reviewer_AJbB · 2026-04-03
> >
> > I thank the authors for their time and their rebuttal.
> >
> > - After re-reading the paper and appendix,  the submission does contain empirical evidence in Appendix H, although it is limited and easy to miss. However, I note that these experiments validate internal theoretical consistency  rather than practical utility. I encourage the authors to include at least one problem where the robust average-reward criterion is the natural objective.
> > - On the oracle construction and presentation: the rebuttal's explanation of the K-order MLMC is clearer than the main text's treatment.  I appreciate the clarification that the K-order MLMC does not itself require Assumption 3.1
> > - On the sample complexity: doesn't the $o(1)$ term yield a complexity that, for practical values of $\epsilon$, is much closer to $\epsilon^{-3}$ if $o(1)=c/\sqrt{\log(1/\epsilon)}?
> > - On the online extension: the UCRL2 idea is reasonable
> > - That said, my main concern remains Assumption 3.1. As written, it requires irreducibility for every policy and every admissible kernel, which is a strong condition. The rebuttal gives some intuition for when it may hold, but that justification is not developed in the paper itself.

---

> > > ### Author Response · Authors · 2026-04-03
> > >
> > > We appreciate the reviewer's careful response. We promise to include all the discussed modifications to our paper. We also sincerely request you to consider adjusting your evaluation score of our work.
> > >
> > > **Assumption.** We sincerely appreciate your suggestions. As we mentioned in the response, due to the hardness of robust average reward, such a 'global' condition (for all kernels and policies) is standard and inevitable for convergent algorithm design and complexity analysis in current studies. And even for this setting, the finite sample complexity is still limited understand. **We will include our discussions and justifications to the paper.**
> > >
> > > **Sample complexity.**
> > > We thank the reviewer for this clarifying question. In the asymptotic regime, the exponent is closer to **$-2$**.
> > > To see why, we can evaluate the limit of the exponent of $\varepsilon$ as the error tolerance $\varepsilon$ approaches $0$:
> > > $$\lim_{\varepsilon \to 0^+} \left( -2 - \frac{c}{\sqrt{\log(1/\varepsilon)}} \right).$$
> > >
> > > As $\varepsilon \to 0^+$, the term $1/\varepsilon$ approaches $+\infty$, which means the denominator $\sqrt{\log(1/\varepsilon)}$ also grows to $+\infty$. Consequently, the lower-order penalty term $\frac{c}{\sqrt{\log(1/\varepsilon)}}$ vanishes to $0$.
> > >
> > > Therefore, the limit of the exponent is exactly $-2$ instead of $-3$.
> > >
> > >
> > > **Experiments.**
> > > In our experiment part, we validate the convergence of RHI in Sec H.2, where we evaluate the robust average reward as the objective function. We will add more experiments with larger and more practical settings to the paper.

---

### Official Review · Reviewer_VvSP · 2026-03-16

**Soundness:** 3
**Presentation:** 4
**Significance:** 3
**Originality:** 3
**Overall Recommendation:** 5
**Confidence:** 4

**Summary:**

This paper studies robust RL under the average-reward criterion, i.e. average-reward MDPs (AMDPs). Robust Halpern Iteration (RHI), a novel algorithm, is proposed to achieve a sample complexity of $\widetilde{O}\left( \frac{SA\mathcal{H}^2}{\varepsilon^{2+o(1)}} \right)$. It first points out the challenges of solving robust AMDPs, and then proposes the RHI algorithm to settle these challenges by leveraging Halpern iterations to find the stationary point of the non-expansion Bellman operator. It also shows that the requirements on the R-SAMPLE oracle can actually be fulfilled by a $K$-order MLMC design, which is able to handle KL-divergence and $\chi^2$ uncertainty sets.

**Compliance With Llm Reviewing Policy:**

Affirmed.

**Key Questions For Authors:**

1. The order of $\varepsilon$ in the sample complexity bound is $\varepsilon^{-(2+o(1))}$, which seems to be close enough to the SOTA $\varepsilon^{-2}$ order, yet eliminating the $o(1)$ term seems to require exponentially many calls of the R-SAMPLE oracle when $\varepsilon$ is small (as indicated by the choice of $C_{\mathrm{rs}}$ in Theorems 5.2 and 5.3, which could be prohibitively high. The issue seems to be resolved in the contamination model briefly covered in Appendix F, where the $C_{\mathrm{rs}}$ disappears. Is this dependency on $C_{\mathrm{rs}}$ purely a proof artifact, or is it something deeply rooted in the uncertainty set structure? Do you have any ideas on how to remove the annoying $o(1) exponent?
2. Is the RHI algorithm (or its variants) generalizable to a wider range of uncertainty set structures, e.g. $s$-rectangular/Wasserstein? Further, is it possible to generalize it to deal with non-tabular MDPs/function approximations? If such ideas, however vague they might be, do exist in your mind, what can we expect about the sample complexity in those cases?
3. Section D.3.1 is somehow empty but does not quite interrupt the flow. I wonder if something is really missing there.

**Limitations:**

It could be better if a brief discussion on limitations could be appended to the conclusion sectio.

**Strengths And Weaknesses:**

**Soundness**: This is a mostly theoretical paper. Proofs of all claims are included in the appendix, and the proofs are examined to be (largely) correct by the reviewer. The experimental evaluation, on the other hand, is quite limited and preliminary, which is understandable due to theoretical nature of the paper, but could be improved to provide empirical evidence of the algorithm's practical performance.

**Presentation**: The paper is very carefully written in a structured and reader-friendly way. Motivations and discussions are sufficiently provided to help readers follow the flow and make sense of the results. Section 4.1 is especially appreciated in that it clearly explains what challenges we need to overcome in designing the RHI algorithm. It is also satisfying to see that the assumed black-box sampling oracle can actually be constructed to satisfy the assumptions needed.

**Significance**: The research problem is of great theoretical interest. The results fit well into the literature and achieve SOTA sample complexity, as summarized in Table 1. In particular, the results do not rely on the aperiodicity or uni-chain assumption required by similar results, which constitute an advancement in analyzing techniques.

**Originality**: The method and the analysis method both look new to me.

---

> ### Author Rebuttal · Authors · 2026-03-26
>
> We sincerely thank the reviewer for the positive assessment of our work and for recognizing the contributions in algorithmic design, analysis, and presentation. We address each question below.
>
> **Q1: The $o(1)$ term.** We first clarify that when $\varepsilon$ is small, this $o(1)$ term does not require exponentially many samples, instead, it will be much smaller than any polynomial of $\varepsilon$. In our results, this term is $\exp(\sqrt{\log(\varepsilon^{-1})})$, and we have $\lim_{\varepsilon\to 0} \frac{\exp(\sqrt{\log(\varepsilon^{-1})})}{\varepsilon^{-p}}=0$ for any $p>0$. Hence, this term will only introduce a sub-polynomial ordered additional complexity.
>
> This term arises specifically from the bias-variance tradeoff inherent in estimating nonlinear functionals of the transition kernel (e.g., the KL or $\chi^2$ support functions), rather than from the RHI algorithm. Specifically, for KL and $\chi^2$ sets, the robust support function $\sigma_{s,a}(h)$ is nonlinear of the nominal distribution. Any existing estimator of such a functional incurs a bias of order $\mathcal{O}(1/N)$ (as we discussed in Sec 5). Our $K$-order MLMC reduces this to $\mathcal{O}(f(K)/N_0^{K+1})$, but the factorial growth $f(K) = \tilde{\mathcal{O}}(K!)$ prevents us from taking $K \to \infty$ freely, and balancing these terms yields the $o(1)$ residual.
>
> Thus, this term is due to the uncertainty set structure. As the reviewer correctly observed, for uncertainty sets where $\sigma_{s,a}$ is linear (contamination and $l_p$-norm models), standard Monte Carlo is unbiased, and the $o(1)$ term vanishes (Theorem F.2).
>
>
> One promising direction to remove it is to develop variance-reduced or debiased estimators for the support function that avoid the factorial growth in the bias expansion. For instance, techniques like variance reduction [1] tailored to the specific structure of KL/chi-squared duality might reduce the bias more efficiently (without the compromise of $f(K)$). We leave this as an important open question.
>
>
> **Q2: Is RHI generalizable?**
> Thank you for this forward-looking question. We discuss all aspects below.
>
> Other uncertainty structures: Our RHI framework is modular: the outer Halpern iteration (Algorithm 1) is agnostic to the uncertainty set---it only requires an R-SAMPLE oracle satisfying Assumption 4.3. Therefore, extending RHI to other uncertainty sets reduces to constructing a suitable oracle. Beyond the models we studied, any $f$-divergence set where the conjugate $f^*$ is smooth leads to a smooth dual objective, and our technique is extendable. This includes a broader range of $f$-divergence, like R\'enyi divergence, Cressie-Read family, etc [2]. However, we also clarify that our technique may not be directly applicable for Wasserstein sets, where the support involves another optimization over couplings rather than reweighting. We could instead use the standard truncated MLMC for the Wasserstein sets  (Xu et al., 2025a;b), which will imply a convergent algorithm with a (higher) sample complexity, together with our RHI analysis.
>
>
>
> For $s$-rectangular sets, the key challenge is that the worst-case kernel couples across actions at each state, making the robust Bellman operator harder to decompose. Extending our quotient-space formulation to this setting is nontrivial but conceptually feasible if we can develop the robust optimality equation for the $s$-rectangular sets. This could be done by utilizing the robust discounted Bellman equation for $s$-rectangular sets [3] and the discount-vanishing limit technique (Wang et al., 2023c).
>
>
> Function approximation: Extending RHI to non-tabular settings (e.g., linear function approximation) is also an exciting direction. The main obstacles are: (1) the quotient-space embedding relies on the finite-dimensional structure of the Q-table, and (2) the span semi-norm analysis would need to be replaced by appropriate function-class-dependent norms. We believe combining our framework with recent advances in non-robust average-reward RL with function approximation could be a fruitful research direction.
>
> We will add a discussion of these extensions in the conclusion of the revised paper.
>
> **Q3: Section D.3.1.**
> Thank you for catching this. This section head should be removed.
>
> **Limitation part.**
> We thank the reviewer for this suggestion. We will add a dedicated paragraph in the conclusion discussing limitations, including: the generative model assumption versus more practical online/offline settings,  the $o(1)$ gap for nonlinear uncertainty sets, and the restriction to tabular settings.
>
> [1] Wang, Shengbo, et al. Sample complexity of variance-reduced distributionally robust q-learning. JMLR 2024
>
> [2] Ghosh, Debamita, et al. Provably near-optimal distributionally robust reinforcement learning in online settings. AAAI 2026
>
> [3] Li, Zhenghao, et al. Near-Optimal Sample Complexities of Divergence-based S-rectangular Distributionally Robust Reinforcement Learning. AISTATS 2026

---

> > ### Author Rebuttal · Reviewer_VvSP · 2026-03-31
> >
> > Thank you for the responses. I'll keep my positive evaluation of this paper.

---

> > > ### Author Response · Authors · 2026-03-31
> > >
> > > Dear reviewer, thank you for taking the time to read our rebuttal and for your continued positive assessment of our work. We deeply appreciate your constructive feedback and your engagement throughout the review process.

---

### Decision · Program_Chairs · 2026-04-30

**Decision:**

Accept (regular)

**Comment:**

**Summary:** This paper studies model-free robust reinforcement learning under the average-reward criterion and proposes Robust Halpern Iteration (RHI), a new algorithm for robust average-reward MDPs in the generative-model setting. A central technical challenge is that the robust Bellman operator is generally non-contractive and involves an unknown gain term; the paper addresses this through a quotient-space reformulation and Halpern iteration. To estimate the robust Bellman operator from nominal samples, the paper introduces a black-box oracle abstraction and instantiates it using a higher-order multi-level Monte Carlo estimator, leading to near-optimal finite-sample guarantees for KL- and chi-squared-based uncertainty sets. Overall, the paper makes a strong theoretical contribution to robust average-reward RL.

**Meta Review:** The reviewers were uniformly positive about the paper. Reviewer VvSP viewed both the algorithmic design and the analysis as novel, and highlighted the significance of obtaining state-of-the-art sample complexity without relying on stronger assumptions such as aperiodicity or unichain conditions. Reviewer mjJf also regarded the quotient-space formulation and higher-order MLMC estimator as technically interesting and meaningful, and found the finite-sample guarantees in the robust average-reward setting to be a substantive advance over recent related work. Reviewer AJbB likewise recognized original ideas in the use of Halpern iteration and the K-order MLMC construction, and after discussion acknowledged that the paper provides a clear theoretical advance and fits well into the literature. Taken together, the reviews support acceptance of the paper as a solid and impactful theoretical contribution.

There are, however, a few minor points that should be improved in the final version. First, as pointed by Reviewer AJbB, Assumption 3.1 is a strong structural condition, requiring irreducibility uniformly over all policies and admissible kernels. While such assumptions are understandable in a theory paper, the final paper would benefit from a clearer and more concrete discussion of when this assumption is natural, for example, by giving intuitive sufficient conditions or worked examples, and by clarifying more explicitly how the argument might extend to weaker settings such as weakly communicating problems. Second, the empirical evidence, while acceptable for a theory paper, is limited and easy to overlook, and the final version would benefit from making these experiments more visible and from briefly emphasizing that they mainly validate the theoretical mechanisms rather than demonstrate broad practical superiority.

Overall, I recommend acceptance.